# GRAPH NEURAL NETWORKS EXTRAPOLATE OUT-OF-DISTRIBUTION FOR SHORTEST PATHS

## ABSTRACT

Neural networks (NNs), despite their success and wide adoption, still struggle to extrapolate out-of-distribution (OOD), i.e., to inputs that are not well-represented by their training dataset. Addressing the OOD generalization gap is crucial when models are deployed in environments significantly different from the training set, such as applying Graph Neural Networks (GNNs) trained on small graphs to large, real-world graphs. One promising approach for achieving robust OOD generalization is the framework of *neural algorithmic alignment*, which incorporates ideas from classical algorithms by designing neural architectures that resemble specific algorithmic paradigms (e.g. dynamic programming). The hope is that trained models of this form would have superior OOD capabilities, in much the same way that classical algorithms work for all instances. We employ sparsity regularization as a tool for analyzing the role of algorithmic alignment in achieving OOD generalization, focusing on graph neural networks (GNNs) applied to the canonical **shortest path** problem. We prove that GNNs, trained to minimize a sparsity-regularized loss over a small set of shortest path instances, are **guaranteed to extrapolate** to arbitrary shortest-path problems, including instances of any size. In fact, if a GNN minimizes this loss within an error of $\epsilon$, it computes shortest path distances up to $O(\epsilon)$ on instances. Our empirical results support our theory by showing that NNs trained by gradient descent are able to minimize this loss and extrapolate in practice.

## 1 INTRODUCTION

Neural networks (NNs) have demonstrated remarkable versatility across domains, yet a persistent and critical challenge remains in their ability to generalize to out-of-distribution (OOD) inputs, i.e., inputs that differ distributionally from their training data. This challenge is pervasive in machine learning and arises whenever a model is applied to situations that are not represented in the training data. For instance, a medical diagnosis model trained on North American patients may struggle to generalize when applied to patients in the UK due to differences in underlying population distributions. This issue has motivated entire subfields, such as distribution shift, transfer learning, and domain adaptation (10; 34; 23).

Graphs, in particular, highlight this challenge as they can vary dramatically in size, connectivity, and topological features. Graph neural networks (GNNs) (29; 11) have seen tremendous development in the past decade (46; 37), and have been broadly applied to a wide range of domains, from social network analysis (9; 4) and molecular property prediction (7; 38; 36) to combinatorial optimization (1; 15). However, these applications often involve scenarios where the graphs encountered in practice are significantly larger, more complex, or structurally distinct from those in training. The case of **size generalization**, where we hope to generalize to graphs larger than seen in training, is an especially severe case of the OOD generalization problem as the graphs belong to distinct spaces, making the training and test distributions disjoint.

Empirical evidence shows that a powerful route to OOD generalization is *algorithmic alignment*—designing a model's architecture to match a target algorithmic framework (41; 2). Such alignment biases the network to finding solutions that resemble algorithms, and thus, inherits properties of algorithms like size independence that can aid OOD generalization. Despite many empirical successes, theoretical guarantees for this approach remain extremely limited.

The primary challenge to OOD guarantees is the very expressivity that makes neural networks successful: highly expressive models can fit the training distribution while realizing hypotheses

that fail off-distribution. (See (44) for an example with GNNs, and interesting discussion on the challenges of size generalization.) Consequently, capacity-based generalization bounds are ill-suited to distribution shift and to size generalization. Existing positive results typically assume a tightly controlled relation between train and test. For example, they embed graphs of different sizes into a shared limit space (graphons) and bound their distance (26; 17; 16; 21), or impose explicit discrepancy bounds between the two distributions (23; 22)—thereby narrowing applicability. We instead ask, can OOD generalization guarantees be achieved **without any assumptions on the data**? We answer this question in the affirmative by taking an approach that focuses on analyzing a model's inductive bias. In particular, our introduction of explicit regularization makes the effect of algorithmic alignment on inductive bias explicit and analyzable, rendering OOD guarantees tractable.

> By combining algorithmic alignment and sparsity (which lend strong inductive biases), we demonstrate that it is possible to train GNNs that provably overcome OOD generalization challenges for the canonical task of computing shortest paths. In particular, we show that training a GNN on just a few well selected small graphs can yield a model that generalizes provably well to arbitrarily large graphs, marking the first result of this kind.

Message-passing graph neural networks are popular architectures for handling data in the form of graphs (cf. surveys (11; 47)). At a high level, they operate by assigning each node $v$ an ***embedding***—say, a vector $h_v \in \mathbb{R}^d$—and then iteratively updating these embeddings until they contain a solution to the problem at hand. During each step, each node updates its embedding based on the embeddings of its neighbors and the weights of the edges connecting them. More precisely, letting $h_v^{(\ell)}$ denote the embedding after $\ell$ update steps,

$$h_v^{(\ell)} = f^{\text{up}}\left(h_v^{(\ell-1)}, f^{\text{combine}}\left(\{h_u^{(\ell-1)} \oplus w_{uv} : u \in \mathcal{N}(v)\}\right)\right), \tag{1}$$

where $\oplus$ denotes concatenation, $w_{uv}$ is the weight of the edge between $u$ and $v$, the set $\mathcal{N}(v)$ is the neighborhood of $v$, and where $f^{\text{combine}}$ and $f^{\text{up}}$ are functions realized by feedforward neural nets. In this way, after $\ell$ update steps, each node's embedding can incorporate information from other nodes up to $\ell$ hops away (see Figure 1). Since we focus only on message-passing graph neural networks in this work, we refer to them simply as GNNs henceforth. As this model applies to graphs with any number of nodes, a key question is: when and how do GNNs perform well on inputs of varying size?

Neural algorithmic alignment is a well-studied framework aiming to design neural architectures that align structurally with specific algorithmic paradigms for the purpose of improving the OOD generalization abilities of a NN. For instance, an astonishingly vast range of practical algorithms are based on ***dynamic programming***, an algorithmic strategy that exploits ***self-reducibility*** in problems: that is, expressing the solution to the problem in terms of solutions to smaller problems of the same type. Shortest paths admit such a decomposition: if the shortest path from $s$ to $t$ goes through node $u$, then it consists of the shortest path from $s$ to $u$, followed by the shortest path from $u$ to $t$, two smaller subproblems. Interestingly, dynamic programs appear to be well-aligned with graph neural networks (39; 3).

The Bellman-Ford (BF) algorithm for shortest-path computations is the canonical example in the algorithmic alignment literature, highlighting the connection between dynamic programming and message-passing GNNs. In each iteration $k$ of BF, the shortest path distances from each node to the source that are achievable with at most $k$ steps are computed using the shortest path distances achievable with at most $(k-1)$ steps. For a specific node $v$, the distance $d_v^{(k)}$ is updated as

$$d_v^{(k)} = \min\left\{d_u^{(k-1)} + w_{(u,v)} : u \in \mathcal{N}(v)\right\}, \tag{2}$$

where $w_{(u,v)}$ is the weight of the edge connecting $u$ to $v$. This iterative update process closely mirrors the message-passing mechanism in GNNs, where node features are updated layer by layer based on aggregated information from their neighbors.

**Contribution.** Our work provides theoretical guarantees and empirical validation of out-of-distribution generalization, and marks a significant advancement in understanding the benefits of neural algorithmic alignment. While many prior studies have highlighted the expressivity of NNs, the structural similarity of GNNs and classical algorithmic control flows, and their capacity to mimic algorithmic behavior, they typically fall short of providing rigorous guarantees on generalization,

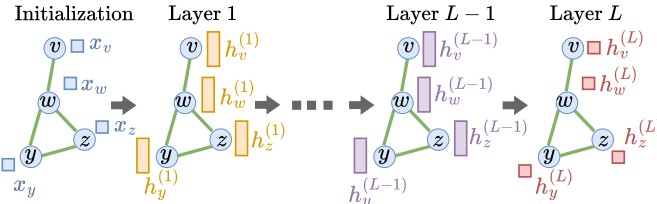

Figure 1: Diagram showing $L$ GNN layers. The node features are represented by rectangles. At each layer, node features are updated according to neighboring node features and the weights of the adjoining edges.

particularly in the OOD setting. In contrast, we show that, by using sparsity regularized training regime, GNNs aligned with the BF algorithm provably extrapolate to arbitrary graphs, regardless of size or structure.

We use sparsity to elucidate the connection between GNNs and the Bellman-Ford (BF) algorithm and demonstrate how it provides guaranteed OOD generalization. In particular, we show that if a GNN minimizes a sparsity-regularized loss over a particular small set of shortest-path instances, then the GNN exactly implements the BF algorithm and hence works on arbitrary graphs, regardless of size. Furthermore, if the GNN minimizes the loss up to some error, then it generalizes with at worst proportional error. In this sense, a small sparsity-regularized loss over a specific set instances serves as a certificate guaranteeing the model's ability to generalize to arbitrary graphs. Furthermore, we empirically validate that gradient-based optimization indeed finds these BF-aligned solutions, highlighting the practical viability of leveraging algorithmic alignment for enhanced generalization. *This work moves algorithmic alignment beyond intuitive analogies or expressivity-based arguments. Moreover, our results highlight the unique potential of algorithmic alignment to bridge data-driven and rule-based paradigms, offering a principled framework for tackling generalization challenges in NNs.* We believe that our approach combining regularization with algorithmic alignment to analyze OOD generalization will prove useful in other settings. Indeed, subsequent work of (5) has used a similar approach to get OOD generalization guarantees for GNNs in solving the heat equation.

## 1.1 RELATED WORK

**Neural algorithmic alignment.** Data-dependent approaches to solving combinatorial optimization problems have surged in the past few years (1) with GNNs among the most popular architectures used. Early work on GNNs for algorithmic tasks was primarily empirical (15; 13) or focused on representational results (28; 19). The idea of *neural algorithmic alignment* emerged as a conceptual framework for designing suitable GNNs, by selecting architectures that could readily capture classical algorithms for similar tasks. This framework has sample complexity benefits (39) and promising empirical results (30; 8; 40). It has also gained traction as a way of understanding the theoretical properties of a given model in terms of its behavior for simple algorithmic tasks (such as BF shortest paths or dynamic programming as a whole) (31; 33; 32; 6). Our work is the first to establish, both theoretically and empirically, that a NN will converge to the correct parameters which implement a specific algorithm.

**Size generalization.** Size generalization of graph neural networks has been studied empirically, in a variety of settings including classical algorithmic tasks (31), physics simulations (27), and efficient numerical solvers (20). There is also work on generalization properties of infinite-width GNNs (the so-called *neural tangent kernel* regime); for the simple problem of finding max degree in a graph, (41) show that graph neural networks in the NTK regime with max readout can generalize to out-of-distribution graphs. Complementary graphon approaches use graph limits combined with continuity of GNNs to understand size generalization (26; 17; 16; 21). Beyond size generalization in GNNs, there is a parallel literature on Transformers that analyzes extrapolation to longer sequences, often called *length generalization* (instead of size generalization). Current work on length generalization asks whether models trained on short sequences of simple arithmetic problems such as addition and modular addition can correctly solve longer problems of the same type (45; 12; 18; 14). Within this line, RASP and C-RASP model attention as discrete programs, yielding logic equivalences and depth hierarchies that attempt to explain when such programs extrapolate (35; 45; 43; 42). Although

these works offer empirically grounded frameworks for size generalization, they provide no formal guarantees. In contrast, we give conditions for provable size generalization.

## 2 EXTRAPOLATION GUARANTEES

### 2.1 MODEL

In graph neural networks, each node (and possibly edge) is associated with a vector, and in each layer of processing, these vectors are updated based on the vectors of neighboring nodes and adjacent edges. An ***attributed*** undirected graph is of the form $G = (V, E, X_\mathrm{v}, X_\mathrm{e})$, where $X_\mathrm{e} = \{x_e : e \in E\}$ are the edge embeddings and $X_\mathrm{v} = \{x_v : v \in V\}$ are the node embeddings. In our case, the edge embeddings will simply be fixed nonnegative edge weights, $x_{(u,v)} = w_{uv}$, with self-loops set to zero, $x_{(u,u)} = 0$. The initial node embeddings $X_\mathrm{v}$ encode the problem input, while the final node embeddings $h_v^{(L)}$ contain the computed shortest path distances. For instances of shortest path problems, we take $x_v = 0$ if $v$ is the source node and use $x_v = \beta$ to indicate nodes with

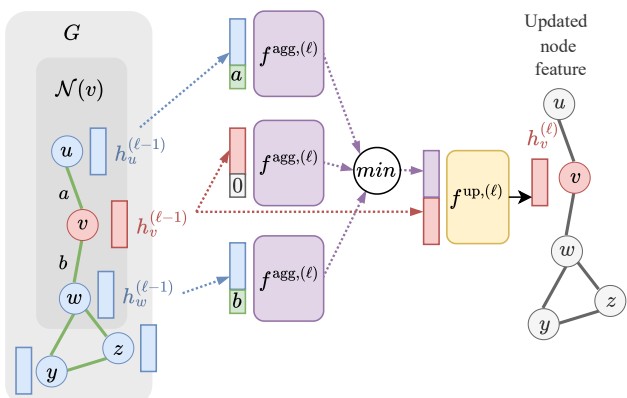

Figure 2: Visual representation of the $\ell$-th layer of a MinAgg GNN operating on a graph $G$, where $f^{\mathrm{agg},(\ell)}$ is the aggregation MLP and $f^{\mathrm{up},(\ell)}$ is the update MLP. Only nodes in the neighborhood $\mathcal{N}(v)$ of $v$ are used in the update, so the output at $v$ is independent of $x$ and $y$.

infinite distance to the source, where $\beta$ is some number greater than the sum of edge weights. The space of graphs we consider is then

$$\mathscr{G} = \left\{ G = (V, E, X_\mathrm{v}, X_\mathrm{e}) : \sum_{e \in E} x_e < \beta \right\}.$$

The embedding of node $v$ at step $\ell$ is denoted $h_v^{(\ell)}$ and follows the update rule in Eq. 1 above. (When referring to specific graphs we use $h_v^{(\ell)}(G)$ and $x_{(u,v)}(G)$.)

The $f^{\mathrm{up},(\ell)}$ function is an MLP that takes two vectors as input: the current embedding of node $v$, and a vector representing the aggregated information from $v$'s neighbors. It outputs the new embedding of $v$. Here $\mathcal{N}(v)$ denotes the neighbors of node $v$, and we take them to include $v$ itself. The $f^{\mathrm{combine},(\ell)}$ function combines the embeddings of $v$'s neighbors, and the edge weights, into a single vector. A common choice is to apply some MLP $f^{\mathrm{agg},(\ell)}$ to each (neighbor, edge weight) pair and to then take the sum, or max, or min, of these $|\mathcal{N}(v)|$ values. We adopt the min. This design choice aligns the network with the structure of the BF algorithm, while still representing a broad and expressive class of GNNs.

**Definition 2.1.** *An L-layer Min-Aggregation Graph Neural Network (MinAgg GNN) with d-dimensional hidden layers is a map $\mathcal{A}_\theta : \mathscr{G} \to \mathscr{G}$ which is computed by layer-wise node-updates (for all $\ell \in [L]$) defined as*

$$h_v^{(\ell)} = f^{\mathrm{up},(\ell)}\left( \min_{u \in \mathcal{N}(v)} \{f^{\mathrm{agg},(\ell)}(h_u^{(\ell-1)} \oplus x_{(u,v)})\} \oplus h_v^{(\ell-1)} \right) \tag{3}$$

*where $f^{\mathrm{agg},(\ell)} : \mathbb{R}^{d_{\ell-1}+1} \to \mathbb{R}^d$ and $f^{\mathrm{up},(\ell)} : \mathbb{R}^{d+d_{\ell-1}} \to \mathbb{R}^{d_\ell}$ are L-layer ReLU MLPs, and $d_0 = d_K = 1$. Given an input $G = (V, E, X_\mathrm{v}, X_\mathrm{e})$ the initialization is $h_u^{(0)} = x_u$.*

For simplicity, we assume that $d_\ell = d$ for $L > \ell > 0$. This assumption is made to reduce the number of hyperparameters needed in the analysis, but is made without loss of generality – all of our results hold with general $d_\ell$. Furthermore, the choice to make all MLP's have $L$ layers is also made for simplicity of presentation (again, without loss of generality). Let $\Gamma$ be a map which

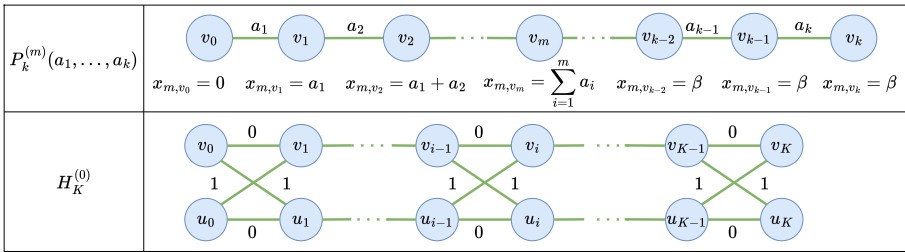

Figure 3: Graphs used in the training sets $\mathscr{H}_{\text{small}}$ and $\mathscr{G}_K$.

implements a single step of the BF algorithm. If $G = (V, E, X_e, X_v)$ is an attributed graph, then $\Gamma(G) = (V, E, X_e, X_v')$ such that for any $v \in V$,

$$x_v' = \min\{x_u + x_{(u,v)} : u \in \mathcal{N}(v) \cup \{v\}\}.$$

We aim to train a GNN to learn $K$ iterations of $\Gamma$, which we denote by $\Gamma^K$.

## 2.2 TOY EXAMPLE

We begin with a toy example that introduces the main ideas. Suppose we look at perhaps the simplest possible GNN that is capable of computing shortest paths. It has updates of the form

$$h_v^{(1)} = \sigma(w_2 \min_{u \in \mathcal{N}(v)} \{\sigma(W_1(x_u \oplus x_{(v,u)} + b_1))\} + b_2) \tag{4}$$

Notice that there are just five parameters in this model: $b_1, b_2, W_{11}, W_{12},$ and $w_2$. The BF algorithm can be simulated by this GNN, as $b_1 = b_2 = 0$ and $W_{11} = W_{12} = w_2$ yields Eq. 2. Interestingly, there are many parameter choices that implement the BF update: all that is needed is that $w_2 W_{11} = w_2 W_{12} = 1$ and $w_2 b_1 + b_2 = 0$. Now, let's consider training this model using a small collection of eight graphs, each a path consisting of just one or two edges. Specifically, let $\mathscr{H}_{\text{small}} = \mathscr{H}_0 \cup \mathscr{H}_1$ with

$$\mathscr{H}_0 = \{P_1^{(0)}(a_i) : i \in \{1, \ldots, 4\}\} \quad \text{and} \quad \mathscr{H}_1 = \{P_2^{(1)}(a_i, 0) : i \in \{5, \ldots, 8\}\} \tag{5}$$

where $P_k^{(m)}$ denotes a path graph as defined in Fig. 3. The labeled training set is then $\mathcal{H}_{\text{small}} = \{(G, \Gamma(G)) : G \in \mathscr{H}_{\text{small}}\}$.

**Theorem 2.2.** *Let $0 < \epsilon < 1$. If $\forall G \in \mathscr{H}_{\text{small}}$ and $\forall u \in V(G)$, a 1-layer GNN $\mathcal{A}_\theta$ with update given by Eq. 4 computes a node feature satisfying $|h_u^{(1)}(G) - x_u(\Gamma(G))| < \frac{\epsilon}{20}$, then for any $G' \in \mathscr{G}$ and $v \in V(G')$*

$$(1 - \epsilon)x_v(\Gamma(G')) - \epsilon \le h_v^{(1)}(G') \le (1 + \epsilon)x_v(\Gamma(G')) + \epsilon.$$

This theorem shows that if the GNN in Eq. 4 achieves low loss on $\mathscr{H}_{\text{small}}$ then it must implement the $\Gamma$ operator (a BF step) up to proportionally small error.

*Proof Sketch.* Recall that $\sigma(\cdot)$ is the ReLU activation function, which effectively divides the input space into two halfspaces. This means that the output of the model on any of the input graphs is one of just 4 possible linear functions of the input. The number of input graphs is enough to cover all these cases, so if there is small error on $\mathcal{H}_{\text{small}}$, the model must simplify to

$$h_v^{(1)} = w_2(\min_{u \in \mathcal{N}(v)} W_1(x_u \oplus x_{(v,u)} + b_1) + b_2) \tag{6}$$

for most training instances. It is now straightforward to show small error is only achieved if $w_2 W_{11}$ and $w_2 W_{12}$ are close to 1 and $w_2 b_1 + b_2$ is close to zero. These conditions guarantee that the BF algorithm is approximately identified. $\square$

## 2.3 MAIN RESULT

Now we move to our main result. This time, we consider a full MinAgg GNN as given by Def. 2.1. To train this model, we again use a small number of simple graphs. The training set contains

$$\mathscr{G}_K = \mathscr{G}_{\text{scale},K} \cup \{P_1^{(0)}(1), P_2^{(1)}(1,0), H_K^{(0)}\} \tag{7}$$

where $\mathscr{G}_{\text{scale},K}$ contains all path graphs of the form $P_{K+1}^{(1)}(a,0,\ldots,0,b,0,\ldots,0)$ for $(a,b) \in \{0,1,\ldots,2K\} \times \{0,2K+1\}\}$ ($b$ is the weight of the $k$th edge). The training instance $H_K^{(0)}$, is shown in Fig. 3. The labeled set is $\mathcal{G}_K = \{(G, \Gamma^K(G)) : G \in \mathscr{G}_K\}$.

For each graph in the training set $G \in \mathscr{G}_{\text{train}}$, we compute the loss only over the set of nodes reachable from the source $V^*(G)$ (the total number of reachable nodes is $|\mathscr{G}_{\text{train}}|^*$). The regularized loss we use $\mathscr{L}_{\text{reg}}$ is

$$\mathcal{L}_{\text{reg}}(\mathcal{G}_{\text{train}}, \mathcal{A}_\theta) = \mathcal{L}_{\text{MAE}}(\mathcal{G}_{\text{train}}, \mathcal{A}_\theta) + \eta\|\theta\|_0, \tag{8}$$

where $\mathcal{L}_{\text{MAE}}$ is

$$\frac{1}{|\mathscr{G}_{\text{train}}|^*} \sum_{G \in \mathscr{G}_{\text{train}}} \sum_{v \in V^*(G)} |x_v(\Gamma^K(G)) - h_v^K(G)|.$$

**Theorem 2.3.** *Consider a training set $\mathscr{G}_{\text{train}}$ with $M$ total reachable nodes and $\mathscr{G}_K \subset \mathscr{G}_{\text{train}}$. For $L \geq K > 0$, if an $L$-layer MinAgg GNN $\mathcal{A}_\theta$ with $m$-layer MLPs achieves a loss $\mathcal{L}_{\text{reg}}(\mathcal{G}_{\text{train}}, \mathcal{A}_\theta)$ within $\epsilon$ of its global minimum, where $0 < \epsilon < \eta < \frac{1}{2M(mL+mK+K)}$, then on any $G \in \mathscr{G}$ the features computed by the MinAgg GNN satisfy*

$$(1 - M\epsilon)x_v(\Gamma^K(G)) \leq h_v^{(L)}(G) \leq (1 + M\epsilon)x_v(\Gamma^K(G))$$

*for all $v \in V(G)$.*

This theorem shows that low regularized loss implies that an $L$ layer MinAgg GNN correctly implements $\Gamma^K$ (i.e., $K$-steps of BF), where the error in implementing this operator is proportional to the distance of the loss from optimal. We later show in experiments that this low loss can be achieved via $L_1$-regularized gradient descent. Here we allow for the training set $\mathscr{G}_{\text{train}}$ to be larger than $\mathscr{G}_K$. However, these additional training examples dilute the training signal from $\mathscr{G}_K$, and so the strongest bounds are given if $\mathscr{G}_{\text{train}} = \mathscr{G}_K$.

*Proof Sketch.*

1. **Implementing BF:** $mL + mK + K$ non-zero parameters are sufficient for the MinAgg GNN to perfectly implement $K$ steps of the BF algorithm.
2. **Sparsity Constraints:** Next, we show that small loss on $\mathscr{G}_K$ necessitates at least $mL + mK + K$ non-zero parameters. Specifically, we show the following.
   - High accuracy on $P_1^{(0)}(1)$ can only be achieved if each layer of $f^{\text{up},(\ell)}$ has at least one non-zero entry. *This requires $mL$ non-zero parameters.*
   - High accuracy on $H_K^{(0)}$ requires $K$ layers where $f^{\text{agg},(\ell)}$ depends on both the node and edge components of its input. This means that each layer of $f^{\text{agg},(\ell)}$ has at least one non-zero entry, and the first layer of $f^{\text{agg},(\ell)}$ has two non-zero entries. *This requires $mK + K$ non-zero parameters.*

   We use this fact to derive that the minimum value of $\mathcal{L}_{\text{reg}}$ is $\eta(mL + mK + K)$. The BF implementation reaches the minimum value of $\mathcal{L}_{\text{reg}}$ since it achieves perfect accuracy with $mL + mK + K$ non-zero parameters.
3. **Simplifying the Updates:** Using the above sparsity structure, we can simplify the MinAgg GNN updates to an equivalent update where the intermediate dimensions are always 1 and there are $K$ updates instead of the previous $m$ updates: $\overline{h}_v^{(k)} = \mu^{(k)} \min_{u \in \mathcal{N}(v)} \left\{ \overline{h}_v^{(k-1)} + \nu^{(k)} x_{(u,v)} \right\}$ where $\mu^{(k)}, \nu^{(k)}, \overline{h}_v^{(k)} \in \mathbb{R}$.
4. **Parameter Constraints and Approximation:** If $\mathcal{L}_{\text{reg}}$ is within $\epsilon$ of its minimum, the parameters $\mu^{(k)}, \nu^{(k)}$ must be constrained to avoid poor training accuracy on certain graphs. These constraints ensure node features approximate BF's intermediate values, and compiling these errors completes the proof.

$\square$

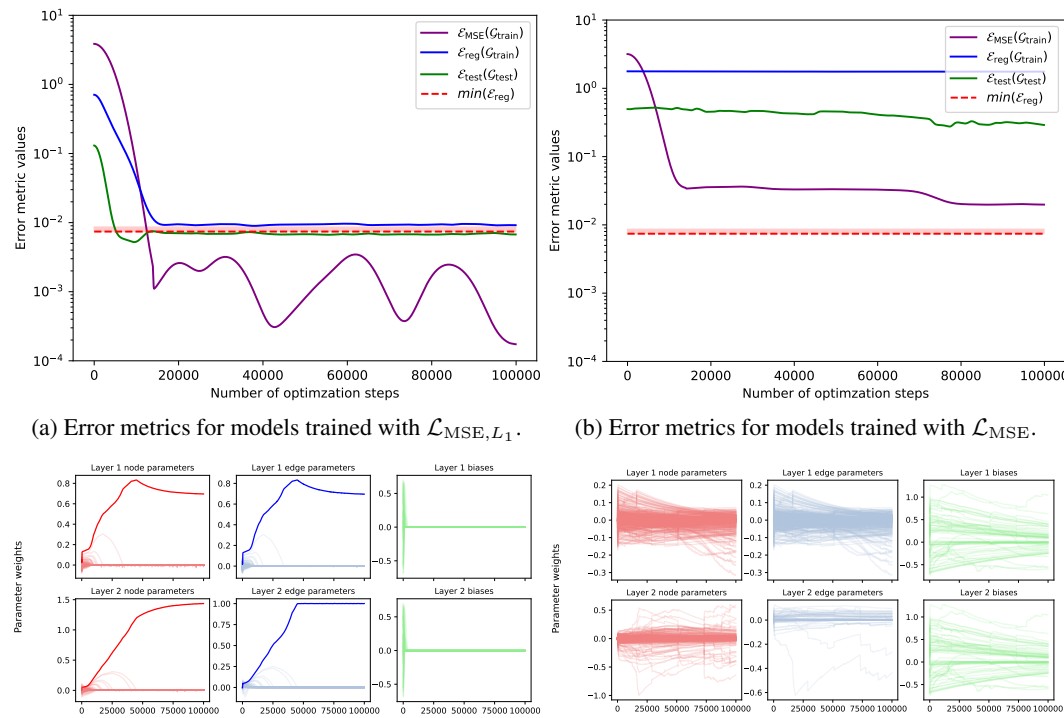

(a) Error metrics for models trained with $\mathcal{L}_{\text{MSE},L_1}$.

(b) Error metrics for models trained with $\mathcal{L}_{\text{MSE}}$.

(c) Model parameters summaries for model trained with $\mathcal{L}_{\text{MSE},L_1}$.

(d) Model parameters summaries for model trained with $\mathcal{L}_{\text{MSE}}$.

Figure 4: Performance metrics and parameter updates for a two-layer MinAgg GNN trained on a two steps of the BF algorithm. The dotted line in (a) and (b) is the global minimum of Eq. (8). In (a) and (b), we track the change in the train loss, test loss, and $\mathcal{L}_{\text{reg}}$ over each optimization step for the models trained with $\mathcal{L}_{\text{MSE},L_1}$ and $\mathcal{L}_{\text{MSE}}$. The final test loss for the model trained with $\mathcal{L}_{\text{MSE},L_1}$ is 0.006 while the final test loss for the model trained with $\mathcal{L}_{\text{MSE}}$ is 0.288. (c) and (d) show changes in model parameters over each optimization step with and without $L_1$ regularization, respectively. Each curve has been smoothed with a truncated Gaussian filter with $\sigma = 20$.

## 3 EXPERIMENTS

Across all configurations, we use a common training set: the constructed set from our theory (Theorem 2.3) augmented with a small number of additional graphs to aid optimization. We then vary the OOD test sets, evaluating extrapolation on unseen graphs spanning different sizes and topologies (cycles, complete graphs, and Erdős–Rényi with $p = 0.5$).

Our main theoretical results (Theorems 2.2 and 2.3) state that a trained model with a sufficiently low $L_0$-regularized loss approximates the BF procedure. We now empirically show how to find such a low-loss trained model by applying gradient descent to a $L_1$-*regularized loss* $\mathcal{L}_{\text{MSE},L_1}$ given by

$$\underbrace{\frac{1}{|\mathscr{G}_{\text{train}}|^*} \sum_{G \in \mathscr{G}_{\text{train}}} \sum_{v \in V^*(G)} (x_v(\Gamma^K(G)) - h_v^K(G))^2}_{\mathcal{L}_{\text{MSE}}} + \|\theta\|_1. \qquad (9)$$

This training loss $\mathcal{L}_{\text{MSE},L_1}$ is a practical proxy for the $L_0$ regularized loss $\mathcal{L}_{\text{reg}}$. To see the effect of sparsity regularization ($L_1$-term), we also train a comparison model using the *unregularized loss* $\mathcal{L}_{\text{MSE}}$ (bracketed terms in Eq. (9)). We show that models trained with $\mathcal{L}_{\text{MSE},L_1}$ find sparse and generalizable solutions for BF; while models trained without sparsity regularization (with $\mathcal{L}_{\text{MSE}}$) have worse generalization.

**Additional setup.** We verify our theoretical results empirically using a two-layer MinAgg GNN trained on two steps of BF. Specifically, we show that converging to a low value of $\mathcal{L}_{\text{reg}}$ indicates

| # of nodes | Single | | Iterated | |
|---|---|---|---|---|
| | No $L_1$-reg. | With $L_1$-reg. | No $L_1$-reg. | With $L_1$-reg. |
| 100 | $0.0202 \pm 0.0055$ | $0.0014 \pm 0.0002$ | $0.0617 \pm 0.0111$ | $0.0036 \pm 0.0002$ |
| 500 | $0.0242 \pm 0.0231$ | $0.0015 \pm 0.0002$ | $0.0881 \pm 0.0080$ | $0.0036 \pm 0.0002$ |
| 1K | $0.0183 \pm 0.0066$ | $0.0035 \pm 0.0005$ | $0.0951 \pm 0.0127$ | $0.0092 \pm 0.0030$ |

Table 1: Measuring $\mathcal{E}_{\text{test}}$ as the number of nodes per graph increases. We test models trained with $\mathcal{L}_{\text{MSE},L_1}$ and models trained with $\mathcal{L}_{\text{MSE}}$. For each model, we examine $\mathcal{E}_{\text{test}}$ (first two columns): for two steps of BF (a single forward pass of each model) and (last two columns): for six steps of BF (where each model is iterated three times). Each test set consists of Erdös–Rényi graphs generated with the corresponding sizes listed with $p$ such that the expected degree $np = 5$. For both models, there is little variation in $\mathcal{E}_{\text{test}}$ as the graph size increases. However, for the iterated version of each model, $\mathcal{E}_{\text{test}}$ for the model trained with $\mathcal{L}_{\text{MSE},L_1}$ remains accurate, while the unregularized model shows a significantly larger test error when iterated 3 times (i.e, comparing third column with first column).

better performance – particularly in improving generalization to larger test graphs. We additionally show that with $L_1$ regularization, the trained model parameters approximately implement a sparse BF step. In our experiments, we configure the MinAgg GNN with two layers and 64 hidden units in both the aggregation and update functions. The first layer has an output dimension of eight, while the second layer outputs a single value. In the supplement, we present additional results evaluating the performance of several other model configurations on one and two steps of BF. To evaluate trained models, we use the following three error metrics:

1. **Empirical training error** ($\mathcal{E}_{\text{MSE}}$): This error $\mathcal{E}_{\text{MSE}}$ is the same as $\mathcal{L}_{\text{MSE}}$ and tracks the model's accuracy on the training set. $\mathscr{G}_{\text{train}}$ consists of $\mathscr{G}_K$ where $K = 2$ as well as four three-node path graphs initialized at step zero of BF and four five-node path graphs initialized at step two of BF. We include these extra graphs to provide examples for the initial and final two steps of the BF algorithm. Empirically, we observe that this expanded training set eases model convergence.

2. **Test error** ($\mathcal{E}_{\text{test}}$): We compute the average multiplicative error of the model predictions compared to the ground-truth BF output over a test set $\mathscr{G}_{\text{test}}$:

$$\mathcal{E}_{\text{test}}(\mathscr{G}_{\text{test}}) = \frac{1}{|\mathscr{G}_{\text{test}}|} \sum_{G \in \mathscr{G}_{\text{test}}} \sum_{v \in V(G)} \left| 1 - \frac{x_v(\Gamma^K(G))}{h_v^K(G)} \right|.$$

$\mathscr{G}_{\text{test}}$ consists 200 total graphs. In order to test the generalization ability of each model, we construct $\mathscr{G}_{\text{test}}$ from 3-cycles, 4-cycles, complete graphs (with up to 200 nodes), and Erdös-Rényi graphs generated using $p = 0.5$.

3. $L_0$-**regularized error** ($\mathcal{E}_{\text{reg}}$): This metric, which is $\mathcal{L}_{\text{reg}}$ (see Eq. 8) evaluated on $\mathscr{G}_{\text{train}}$, shows how the model's performance satisfies the conditions of Theorem 2.3.

Furthermore, we also track a summary of the model parameters per epoch. For a detailed discussion of the model parameter summary see the supplement. In brief, at each layer, we track biases, the parameters which scale the node features, and the parameters which scale the edge features. For the sparse implementation of two-steps of BF, the node and edge parameter updates both have the same single non-zero positive value $a$ in the first layer. In the second layer, the node and edge parameter updates both have a single non-zero positive value but the edge parameter update converges to 1 while the node parameter update converges to $1/a$.

**Results.** Fig. 4 shows the results of training on two steps of BF. Here (a) and (b) show $\mathcal{L}_{\text{MSE},L_1}$ (i.e, the model trained with regularized loss) achieves a low value of $\mathcal{L}_{\text{reg}}$ and a correspondingly a low test error, $\mathcal{L}_{\text{test}}$ (in the supplement, we show that this small value of $\mathcal{L}_{\text{reg}}$ satisfies the conditions of Theorem 2.3). In contrast, the model trained with $\mathcal{L}_{\text{MSE}}$ (i.e., regularized loss) has significantly higher $\mathcal{L}_{\text{reg}}$ and $\mathcal{L}_{\text{test}}$. Although both models achieve values of $\mathcal{E}_{\text{MSE}}$ below 0.10, we see that the train error does not necessarily indicate if a sparse implementation of Bellman-Ford has been learned. As such, Fig. 4 (a) and (b) experimentally validates Theorem 2.3 by demonstrating that low values of $\mathcal{L}_{\text{reg}}$ yield better generalization on test graphs with different sizes and topologies from the train graphs. In Fig. 4 (c) and (d), we elucidate the effect of $L_1$ regularization with regards to achieving

low values of $\mathcal{L}_{\text{reg}}$ and show that that the model trained with $\mathcal{L}_{\text{MSE},L_1}$ indeed approximates a sparse implementation of BF. To further illustrate that achieving low values of $\mathcal{L}_{\text{reg}}$ implies that the BF GNN model will learn to implement Bellman-Ford, we provide parameter heatmaps for models trained with $L_1$ regularization and without in Fig. 5. We observe that the model trained with $L_1$ regularization implements exactly the parameters for Bellman-Ford suggested from our theory.

In Table 1, we further assess the generalization ability of the $L_1$-regularized model on sparse Erdös-Renyí graphs of increasing sizes as compared to the unregularized model. Interestingly, when we use the trained 2-step MinAgg GNN as a primitive module and iterate it 3 times (to estimate $2 \times 3 = 6$ BF steps), the test error of our $L_1$-regularized model *does not accumulate* while the error by the un-regularized model increases by roughly a factor of 3. Again, $L_1$ regularization improves generalization. By iteratively applying the trained 2-step BF multiple times, we obtain an neural model to approximate general shortest paths with guarantees.

## 4 DISCUSSION

We show that algorithmic alignment can fundamentally enhance out-of-distribution (OOD) generalization. By training GNNs with a sparsity-regularized loss on a small set of shortest-path instances, we obtain models that correctly implement the BF algorithm. This result provides a theoretical guarantee that the learned network can generalize OOD to graphs of sizes and structures beyond those encountered during training. This is one of the first results where a neural model, when trained to sufficiently low loss, can **guarantee OOD size generalization** for a non-linear algorithm.

A key challenge in machine learning research lies in comparing neural networks with similar expressivity but different generalization behaviors. This challenge centers on understanding the inductive biases that guide models toward particular solutions. Sparsity regularization creates a setting where the influence of architecture choice on inductive bias is clear and easy to analyze. Indeed, our work demonstrates how this regularization interacts with the GNN architecture to bias the GNN toward implementing the

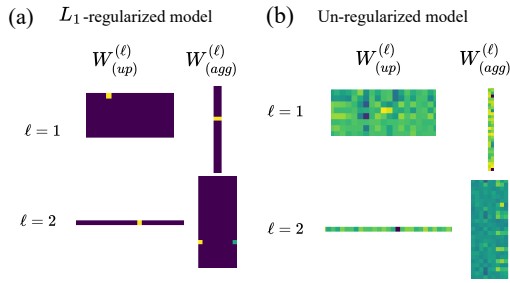

Figure 5: Example of the parameter heatmaps for both the $L_1$-regularized model and the un-regularized model. Our Bellman-Ford model update per node feature is defined as $\sigma(W_{(up)}^{(\ell)} \min\{\sigma(W_{(agg)}^{(\ell)}(x_u + x_{(u,v)} + b_{(agg)}^{(\ell)}) : u \in \mathcal{N}(v)\} + b_{(up)}^{(\ell)}$. Notice that our $L_1$ regularized model only has 6 non-zero parameters (exactly as suggested by our theoretical results) while the un-regularized model does not.

Bellman-Ford algorithm. By making inductive biases explicit and quantifiable, we gain insight into why algorithmic alignment is effective at promoting neural networks to generalize beyond their training distribution, a critical capability for real-world applications.

**Extensions.** As our BF-aligned GNN correctly implements $K$ steps, we can extend this capability by recurrently iterating the network. (See the last column of Table 1.) This approach allows the network to solve shortest-path problems that require more than $K$ iterations. Such scalability enables generalization to shortest-path computations that require arbitrary computational costs.

Furthermore, the ability to learn a single algorithmic step is valuable in broader contexts of neural algorithmic reasoning. This modular design means that the MinAgg GNN can serve as a subroutine within more complex neural architectures that aim to solve higher-level tasks. For instance, in neural combinatorial optimization or graph-based decision-making tasks, shortest-path computations are often just one component of a larger process. By ensuring the network reliably implements each step of the BF algorithm, we create a reusable building block that can be integrated into more sophisticated models. This supports the goal of developing NNs that can reason algorithmically, enabling them to solve increasingly complex problems through the composition of learned algorithmic steps.

**Conclusion.** Our work opens an exciting new direction for research by raising the question of when low training loss can serve as a guarantee for out-of-distribution generalization in other tasks or architectures. While our results focus on the BF algorithm and message-passing GNNs, they

suggest the potential for similar guarantees in other instances of alignment, such as different (dynamic programming based) graph algorithms, sequence-to-sequence tasks, or architectures like transformers and recurrent neural networks. Investigating the structural and algorithmic properties that enable such guarantees could provide a unified framework for designing NNs that generalize reliably across diverse tasks and input domains.

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

## A  DEFINITIONS AND NOTATION

We begin with definitions and notations we utilize in our proof. To ensure the supplementary material is easy to navigate and reader-friendly, we reiterate some definitions from the main text. We take $[n] = \{1, 2, \ldots, n\}$ and use $x \oplus y$ to denote the concatenation of the vectors $x$ and $y$. The neighborhood of a node $v$ is denoted $\mathcal{N}(v)$ and we use the convention $v \in \mathcal{N}(v)$. When referring to the $i$th component of $x$ we write $x_i$ or $[x]_i$.

Given a source node $s \in V$, let $\mathrm{d}^{(t)}(s, v)$ denote the length of the $t$-step shortest path from $s$ to $v$. If no such path exists, $\mathrm{d}^{(t)}(s, v) = \beta$ and $\beta$ is some large number. We define a single $t$-step Bellman-Ford instance to be a attributed graph $G^{(t)} = (V, E, X_{\mathrm{v}}, X_{\mathrm{e}})$ where $X_{\mathrm{v}} = \{x_v = \mathrm{d}^{(t)}(s, v) : v \in V\}$ for some $s \in V$. For every 0-step Bellman-Ford instance $G^{(0)} = (V, E, X_{\mathrm{v}}, X_{\mathrm{e}})$, $x_s = 0$ for the source node $s \in V$ and $x_u = \beta$ for all other nodes $u \in V$. Throughout this manuscript, $t$-step BF instances are always denoted by a superscript $(t)$. Recall that all edge weights considered in this manuscript are non-negative.

Let $\Gamma$ be a map which implements a single step of the BF algorithm. If $G = (V, E, X_{\mathrm{e}}, X_{\mathrm{v}})$ is an attributed graph, then $\Gamma(G) = (V, E, X_{\mathrm{e}}, X_{\mathrm{v}}')$ such that for any $v \in V$,

$$x_v' = \min\{x_u + x_{(u,v)} : u \in \mathcal{N}(v)\}.$$

Let $\Gamma^K$ be $K$ iterations of $\Gamma$. Note that applying $\Gamma^K$ to a 0-step Bellman-Ford instance $G^{(0)}$ yields the $K$-step shortest path from $s$ to $v$, i.e., $\Gamma^K(G^{(0)}) = G^{(K)}$. Although we restrict our training set to BF instances, our extrapolation guarantees show that the MinAgg GNN approximates the operator $\Gamma^K$ on any graph in

$$\mathscr{G} = \left\{ G = (V, E, X_{\mathrm{v}}, X_{\mathrm{e}}) : \sum_{e \in E} x_e < \beta \right\}.$$

Define a length-$k$ path graph instance as $P_k^{(t)}(a_1, \ldots, a_k) = (V, E, X_v, X_e)$ where $V = \{v_0, v_1, \ldots, v_k\}$ and $E = \{(v_{i-1}, v_i) \mid i \in \{1, \ldots, k\}\}$. Let $x_{(v_{i-1}, v_i)} = a_i$, $x_{v_0} = 0$ (i.e. the source node is $s = v_0$) and $x_{v_i} = d^{(t)}(s, v_i)$ for $i > 0$.

**Definition A.1.** *An $L$-layer MinAgg GNN with $d$-dimensional hidden layers is a map $\mathcal{A}_\theta : \mathscr{G} \to \mathscr{G}$ which is computed by layer-wise node-updates (for all $\ell \in [L]$) defined as*

$$h_v^{(\ell)} = f^{\mathrm{up},(\ell)}\Big( \min_{u \in \mathcal{N}(v)} \{f^{\mathrm{agg},(\ell)}(h_u^{(\ell-1)} \oplus x_{(u,v)})\} \oplus h_v^{(\ell-1)} \Big) \tag{10}$$

*where $f^{\mathrm{agg},(\ell)} : \mathbb{R}^{d_{\ell-1}+1} \to \mathbb{R}^d$ and $f^{\mathrm{up},(\ell)} : \mathbb{R}^{d+d_{\ell-1}} \to \mathbb{R}^{d_\ell}$ are $L$-layer ReLU MLPs, and $d_0 = d_K = 1$. Given an input $G = (V, E, X_v, X_e)$ the initialization is $h_u^{(0)} = x_u$. The MinAgg GNN $\mathcal{A}_\theta$ has output $\mathcal{A}_\theta(G) = (V, E, X_v' = \{h_v^{(\ell)} : v \in V\}, X_e)$. A **simple** $L$-layer MinAgg GNN instead uses the layer-wise update*

$$h_v^{(\ell)} = f^{\mathrm{up},(\ell)}\Big( \min_{u \in \mathcal{N}(v)} \{f^{\mathrm{agg},(\ell)}(h_u^{(\ell-1)} \oplus x_{(u,v)})\} \Big). \tag{11}$$

We refer to the first $d_{\ell-1}$ components of the domain of $f^{\mathrm{agg},(\ell)}$ as its node component, and we refer to the last component of the domain of $f^{\mathrm{agg},(\ell)}$ as its edge component.

A $K$-step training set $\mathcal{G}_{\mathrm{train}}$ is a set of tuples where for each element $(G^{(t)}, \Gamma^K(G^{(t)})) \in G_{\mathrm{train}}$ the graph $G^{(t)}$ is a $t$-step BF instance. For a graph $G = (V, E, X_v, X_e)$ let $V^*(G) = \{v \in V : x_v \neq \beta\}$ be the set of reachable nodes and let $|\mathcal{G}_{\mathrm{train}}|^* = \sum_{G^{(t)} \in \mathcal{G}_{\mathrm{train}}} |V^*(G^{(t)})|$ be the total number of reachable nodes in the training set. For each graph $G$ we consider the training loss over the subset of vertices $V^*(G)$ because the choice of the feature at unreachable nodes $\beta$ is arbitrary and so should not be included when providing supervision for shortest path problems.

**Definition A.2.** *An $m$-layer ReLU MLP is a function $f_\theta : \mathbb{R}^{d_0} \to \mathbb{R}^{d_m}$ parameterized by $\theta = \{W_j : W_j \in \mathbb{R}^{d_j \times d_{j-1}}, j \in [m]\} \cup \{b_j : b_j \in \mathbb{R}^{d_j}, j \in [m]\}$ where for all $j \in [m]$,*

$$x^{(0)} = x,$$
$$x^{(j)} = \sigma(W_j x^{(j-1)} + b_j),$$

*and $f_\theta(x) = x^{(m)}$. Here, $\sigma$ is the rectified linear unit (ReLU) activation function.*

The MinAgg GNN is parameterized by the set of weights

$$\theta = \bigcup_{\ell=1}^{L} (\theta^{\mathrm{up},(\ell)} \cup \theta^{\mathrm{agg},(\ell)}),$$

where $\theta^{\mathrm{up},(\ell)}$ and $\theta^{\mathrm{agg},(\ell)}$ denote the parameters of the update and aggregation MLPs at layer $\ell$, respectively.

We also formalize the definition of path graph instances. A 0-step path graph instance $P_k^{(0)}(a_1, \ldots, a_k)$ consists of a graph $(V, E, X_v, X_e)$ where the vertex set is $V = \{v_0, v_1, \ldots, v_k\}$, the edge set is $E = \{(v_{i-1}, v_i) : i \in \{1, \ldots, k\}\}$, and the edge weights are defined as $x_{(v_{i-1}, v_i)} = a_i$ for $i \in \{1, \ldots, k\}$. The node features are initialized as $x_{v_0} = 0$ for the source node $v_0$, while all other nodes $v_i$ for $i > 0$ are initialized with $x_{v_i} = \beta$, representing an unreachable state.

## B WARM-UP: SINGLE LAYER GNNS IMPLEMENT ONE STEP OF BF

We start with the simple setting of a single layer GNN with shallow and narrow MLP components. This example provides key insights on why a perfectly (or almost perfectly) trained model can generalize. We analyze the general case of a multilayer GNN with wide and deep MLPs in Sec. C. Although the general case is more sophisticated technically, the approach follows similar intuitions. In particular, we later show that sparsity regularization can be used to reduce the analysis of GNN with wide and deep MLPs trained on a single BF step to the simple model analyzed in this section.

We start by by proving Theorem B.1, which shows how perfect accuracy on $\mathcal{H}_{\mathrm{small}}$ requires certain restrictions on parameters of the simple MinAgg GNN. Next, in Corollary B.2 we show that such restrictions guarantee the parameters implement the BF algorithm. Finally, we extend this analysis to

evaluate how MinAgg GNNs approximately minimizing the training loss perform on arbitrary graphs in Theorem 2.2

Suppose we have a *simple* 1-layer Bellman-Ford GNN, $\mathcal{A}_\theta$, where $f^{\text{up},(0)} : \mathbb{R} \to \mathbb{R}$ and $f^{\text{agg},(0)} : \mathbb{R}^2 \to \mathbb{R}$ are single layer MLPs. To be explicit:

$$h_u^{(1)} = \sigma(w_2 \min\{\sigma(W_1(x_v \oplus x_{(u,v)} + b_1)) : v \in \mathcal{N}(u) \cup \{u\}\} + b_2), \tag{12}$$

where $\sigma$ is ReLU, $W_1 \in \mathbb{R}^{1 \times 2}$, and $w_2, b_1, b_2 \in \mathbb{R}$.

We consider the training set

$$\mathcal{H}_{\text{small}} = \{(P_1^{(0)}(a_i), P_1^{(1)}(a_i)) : i \in \{1, \dots, 4\}\} \cup \{(P_2^{(1)}(a_i, 0), P_2^{(2)}(a_i, 0)) : i \in \{5, \dots, 8\}\}. \tag{13}$$

For concreteness we take $a_i = 2i$, and we utilize these specific choices of edge weights in the proof of Theorem 2.2. However, any choice of $a_i$ satisfying $a_i \neq a_j$ if $i \neq j$ and $a_i > 0$ is sufficient for the other results in this section.

**Theorem B.1.** *If,* $\forall (H^{(t)}, \Gamma(H^{(t)})) \in \mathcal{H}_{\text{small}}$,

$$\mathcal{A}_\theta(H^{(t)}) = \Gamma(H^{(t)}),$$

*i.e. the computed node features are* $h_u^{(1)}(H^{(t)}) = x_u(\Gamma(H^{(t)}))$ *for all* $u \in V(H^{(t)})$, *then* $w_2 W_1 = \mathbb{1}$ *and* $w_2 b_1 + b_2 = 0$.

*Proof.* First, note that from the definition of $P_1^{(0)}(a_i)$, the source node is $s = v_0$ so $x_{v_0} = 0$, and $x_{v_1}(P_1^{(0)}(a_i)) = \beta$. Additionally, given the training example $(P_1^{(0)}(a_i), P_1^{(1)}(a_i)) \in \mathcal{H}_{\text{small}}$, recall that $P_1^{(0)}(a_i)$ is the input to $\mathcal{A}_\theta$ (1-layer Bellman-Ford GNN). By the definition of $\mathcal{A}_\theta$, the computed node feature for $v_1 \in V(P_1^{(0)}(a_i))$ is

$$h_{v_1}^{(1)} = \sigma(w_2 \min\{\sigma(W_1(x_{v_1} \oplus x_{(v_1,v_1)}) + b_1), \sigma(W_1(x_{v_0} \oplus x_{(v_0,v_1)}) + b_1)\} + b_2)$$
$$= \sigma(w_2 \min\{\sigma(W_{11}\beta + b_1), \sigma(W_{12}a_i + b_1)\} + b_2)$$

where $\sigma$ is the ReLU activation function. Since

$$\mathcal{A}_\theta(P_1^{(0)}(a_1)) = P_1^{(1)}(a_1)$$
$$\vdots$$
$$\mathcal{A}_\theta(P_1^{(0)}(a_4)) = P_1^{(1)}(a_4),$$

for each $v_1 \in V(P_1^{(1)}(a_i))$, $x_{v_1}(P_1^{(1)}(a_i)) = a_i$ so $h_{v_1}^{(1)}(P_1^{(0)}(a_i)) = a_i$. Therefore,

$$a_1 = \sigma(w_2 \min\{\sigma(W_{11}\beta + b_1), \sigma(W_{12}a_1 + b_1)\} + b_2)$$
$$\vdots$$
$$a_4 = \sigma(w_2 \min\{\sigma(W_{11}\beta + b_1), \sigma(W_{12}a_4 + b_1)\} + b_2).$$

Suppose $\sigma(W_{11}\beta + b_1) = \min\{\sigma(W_{11}\beta + b_1), \sigma(W_{12}a_i + b_1)\}$ and $\sigma(W_{11}\beta + b_1) = \min\{\sigma(W_{11}\beta + b_1), \sigma(W_{12}a_j + b_1)\}$ for $i \neq j$. Then $a_i = a_j$ when $i \neq j$ which is a contradiction. Therefore, there can be at most one $i$ for which $a_i = \sigma(w_2\sigma(W_{11}\beta + b_1) + b_2)$. WLOG, assume that

$$a_i = \sigma(w_2\sigma(W_{12}a_i + b_1) + b_2)$$

where $i \in [3]$. Since $a_i > 0$ and $\sigma$ is the ReLU function, we have that $a_i = w_2\sigma(W_{12}a_i + b_1) + b_2$ for $i \in [3]$. Suppose $W_{12}a_i + b_1 \leq 0$ and $W_{12}a_j + b_1 \leq 0$ for $i, j \in [3]$ where $i \neq j$. Then $a_i = a_j = b_2$ which is a contradiction. WLOG, assume that $W_{12}a_i + b_1 > 0$ for $i \in [2]$. Then, we get the following system of linear equations

$$a_1 = w_2 W_{12}a_1 + w_2 b_1 + b_2$$
$$a_2 = w_2 W_{12}a_2 + w_2 b_1 + b_2.$$

These linear equations are only satisfied when $w_2 W_{12} = 1$ and $w_2 b_1 + b_2 = 0$.

Now, consider $\{(P_2^{(1)}(a_i, 0), P_2^{(2)}(a_i, 0)) : a_i \in \mathbb{R}^+, i \in \{5, \ldots, 8\}, a_i \neq a_j\}$. From the definition of $P_2^{(1)}(a_i, 0)$, we know that $s = v_0$, $x_{v_0}(P_2^{(1)}(a_i, 0)) = 0$, $x_{v_1}(P_2^{(1)}(a_i, 0)) = a_i$, $x_{v_2}(P_2^{(1)}(a_i, 0)) = \beta$, $x_{(v_0, v_1)} = a_i$, and $x_{(v_1, v_2)} = 0$. Since $\mathcal{A}_\theta(P_2^{(1)}(a_i, 0)) = P_2^{(2)}(a_i, 0)$, the computed node feature for $v_2 \in V(P_2^{(1)}(a_i, 0))$ is

$$h_{v_2}^{(1)} = a_i = \sigma(w_2 \min\{\sigma(W_1(x_{v_2} \oplus x_{(v_2, v_2)}) + b_1), \sigma(W_1(x_{v_1} \oplus x_{(v_1, v_2)}) + b_1)\} + b_2)$$
$$= \sigma(w_2 \min\{\sigma(W_{11}\beta + b_1), \sigma(W_{11}a_i + b_1)\} + b_2)$$

for $i \in \{5, \ldots, 8\}$. Similar to above, we have that $\sigma(W_{11}\beta + b_1) = \min\{\sigma(W_{11}\beta + b_1), \sigma(W_{11}a_i + b_1)\}$ can only occur for one $i \in \{5, \ldots, 8\}$. Again, WLOG we can assume that $a_8 = \sigma(w_2\sigma(W_{11}\beta + b_1) + b_2)$ and $a_i = \sigma(w_2\sigma(W_{11}a_i + b_1) + b_2)$ for $i \in \{5, 6, 7\}$. Then, using a similar system of linear equations as above, we get that $w_2 W_{11} = 1$. $\qquad \square$

**Corollary B.2.** *Let $\mathcal{A}_\theta$ be a simple 1-layer Bellman-Ford GNN, as given in Eq.* (12)*. If $\mathcal{A}_\theta(H^{(t)}) = \Gamma(H^{(t)})$ for all $(H^{(t)}, \Gamma(H^{(t)})) \in \mathcal{H}_{\text{small}}$, then for any $G \in \mathcal{G}$ the MinAgg GNN outputs $A_\theta(G) = \Gamma(G)$ which means for any $v \in V(G)$*

$$h_v^{(1)} = \min\{x_u + x_{(u,v)} : u \in \mathcal{N}(v)\}.$$

*Proof.* If $\mathcal{A}_\theta(H^{(t)}) = \Gamma(H^{(t)})$ for all $(H^{(t)}, \Gamma(H^{(t)})) \in \mathcal{H}_{\text{small}}$ then, by Theorem B.1, we know that $w_2 W_1 = \mathbb{1}$ and $w_2 b_1 + b_2 = 0$. First, suppose $w_2 < 0$. Since $w_2 W_1 = \mathbb{1}$, we know that $W_{11} = W_{12}$ and $W_{11}, W_{12} < 0$. Consider $(P_1^{(0)}(a_i), P_1^{(1)}(a_i)) \in \mathcal{H}_{\text{small}}$. Recall that $a_i > 0$. For any $i \in \{1, \ldots, 4\}$, we have that $v_1 \in V(P_1^{(0)}(a_i))$ gets the computed node feature

$$h_{v_1}^{(1)} = \sigma(w_2 \min\{\sigma(W_{11}x_s + W_{12}x_{(s,v_1)} + b_1), \sigma(W_{11}x_{v_1} + W_{12}x_{(v_1,v_1)} + b_1)\} + b_2)$$
$$= \sigma(w_2 \min\{\sigma(W_{11}a_i + b_1), \sigma(W_{11}\beta + b_1)\} + b_2)$$

Since $0 \leq a_i \ll \beta$, $W_{11}\beta + b_1 \leq W_{11}a_i + b_1$ so

$$\min\{\sigma(W_{11}a_i + b_1), \sigma(W_{11}\beta + b_1)\} = \sigma(W_{11}\beta + b_1)$$

Then

$$h_{v_1}^{(1)} = \sigma(w_2\sigma(W_{11}\beta + b_1) + b_2)$$

for $v_1 \in P_1^{(0)}(a_i)$ for any $i \in \{1, \ldots, 4\}$. However, this is a contradiction because $\mathcal{A}_\theta(P_1^{(0)}(a_i)) = P_1^{(1)}(a_i)$ for $i \in \{1, \ldots, 4\}$ and $a_i \neq a_j$ for $i \neq j$. Therefore, $w_2 > 0$ so $W_{11}, W_{12} > 0$.

Suppose $w_2 b_1 < 0$. Because $w_2 > 0$, $b_1 < 0$. Additionally, since $w_2 b_1 < 0$ and we know that $w_2 b_1 + b_2 = 0$, we have that $b_2 > 0$. Then consider $(P_1^{(0)}(a_1), P_1^{(1)}(a_1)) = (P_1^{(0)}(0), P_1^{(1)}(0)) \in \mathcal{H}_{\text{small}}$. Then, $v_1 \in V(P_1^{(0)}(0))$ gets the updated node feature

$$h_{v_1}^{(1)} = \sigma(w_2 \min\{\sigma(b_1), \sigma(W_{11}\beta + b_1)\} + b_2) = b_2.$$

This is contradiction because $\mathcal{A}_\theta(P_1^{(0)}(a_1)) = P_1^{(1)}(a_1)$ which means that the computed node feature $h_{v_1}^{(1)}$ should be $a_1 = 0$.

Now, given $G^{(m)} \in \mathcal{G}$, then given $v \in V(G^{(m)})$, the updated node feature for $v \in V(\mathcal{A}_\theta(G^{(m)}))$ is

$$h_v^{(1)} = \sigma(w_2 \min\{\sigma(W_{11}x_u + W_{12}x_{(v,u)} + b_1) : u \in \mathcal{N}(v)\} + b_2)$$
$$= \sigma(\min\{w_2\sigma(W_{11}(x_u + x_{(v,u)}) + b_1) : u \in \mathcal{N}(v)\} + b_2)$$
$$= \sigma(\min\{\sigma(w_2 W_{11}(x_u + x_{(v,u)}) + w_2 b_1) : u \in \mathcal{N}(v)\} + b_2) \text{ since } w_2 > 0$$
$$= \sigma(\min\{\sigma(w_2 W_{11}(x_u + x_{(v,u)}) + w_2 b_1) + b_2 : u \in \mathcal{N}(v)\})$$
$$= \sigma(\min\{\sigma(x_u + x_{(v,u)} + w_2 b_1) + b_2 : u \in \mathcal{N}(v)\})$$
$$= \sigma(\min\{x_u + x_{(v,u)} + w_2 b_1 + b_2 : u \in \mathcal{N}(v)\}) \text{ since } x_u + x_{(v,u)} + w_2 b_1 \geq 0$$
$$= \sigma(\min\{x_u + x_{(v,u)} : u \in \mathcal{N}(v)\})$$
$$= \min\{x_u + x_{(v,u)} : u \in \mathcal{N}(v)\}.$$

$\qquad \square$

**Lemma B.3.** *Consider two points $(x_1, y_1), (x_2, y_2) \in \mathbb{R}^2$ such that $|x_1| < D$ and $|x_2 - x_1| > 2$ and an affine function $f(x) = ax + b$. Suppose $|f(x_1) - y_1| < \epsilon$ and $|f(x_2) - y_2| < \epsilon$. If $a_0 = \frac{y_2 - y_1}{x_2 - x_1}$ and $b_0 = y_1 - a_0 x_1$ are the slope and y-intercept of a line passing through $(x_1, y_1)$ and $(x_2, y_2)$ then $|a_0 - a| < \epsilon$ and $|b_0 - b| < 2(1 + D)\epsilon$.*

*Proof.* First, $a = \frac{f(x_2) - f(x_1)}{x_2 - x_1}$ implies

$$
\begin{aligned}
|a_0 - a| &= \left| \frac{y_2 - y_1}{x_2 - x_1} - \frac{f(x_2) - f(x_1)}{x_2 - x_1} \right| \\
&= \frac{1}{|x_2 - x_1|} |(y_2 - f(x_2)) - (y_1 - f(x_1))| \\
&\leq \frac{1}{|x_2 - x_1|} (|y_2 - f(x_2)| + |(y_1 - f(x_1))|) \\
&\leq \frac{2\epsilon}{|x_2 - x_1|} \\
&\leq \epsilon.
\end{aligned}
$$

Now, since $b = f(x_1) - ax_1$ we have

$$
\begin{aligned}
|b_0 - b| &= |y_1 - a_0 x_1 - (f(x_1) - ax_1)| \\
&= |(y_1 - f(x_1)) - x_1(a_0 - a)| \\
&\leq |y_1 - f(x_1)| + |x||a_0 - a| \\
&< (1 + D)\epsilon.
\end{aligned}
$$

$\square$

We now restate Theorem 2.2 with additional details and provide a proof.

**Theorem B.4.** *Let $0 < \epsilon < 1$. If $\forall (H^{(t)}, \Gamma(H^{(t)})) \in \mathcal{H}_{\text{small}}$, a MinAgg GNN $\mathcal{A}_\theta$ that, for $u \in V(G^{(t)})$, computes a node feature satisfying $|h_u^{(1)}(G^{(t)}) - x_u(\Gamma(G^{(t)}))| < \frac{\epsilon}{20}$. Then*

*(i) $\|w_2 W_1 - \mathbb{1}\|_1 < \epsilon$ and $|w_2 b_1 + b_2| < 20\epsilon$*

*(ii) $w_2, W_{11}, W_{12} \geq 0$*

*(iii) For $G \in \mathcal{G}$ and $v \in V(G)$*

$$
(1 - \epsilon)x_v(G) - \epsilon \leq h_v^{(1)}(G) \leq (1 + \epsilon)x_v(G) + \epsilon
$$

*Proof.* (i) We first show part (i) i.e. if $|h_u^{(t)}(G^{(t)}) - x_u(\Gamma(G^{(t)}))| < \frac{\epsilon}{20}$, for any $(G^{(t)}, \Gamma(G^{(t)})) \in \mathcal{H}_{\text{small}}$, then $\|w_2 W_1 - 1\| < \epsilon$ and $|w_2 b_1 + b_2| < 20\epsilon$. Let $\epsilon_0 = \frac{\epsilon}{20}$. Given the definition of $\mathcal{A}_\theta$, the computed node feature for $v_1 \in V(P_1^{(0)}(a_i))$ for $i \in \{1, \ldots, 4\}$ is

$$
h_{v_1}^{(1)} = \sigma[w_2 \min\{\sigma(W_{11}\beta + b_1), \sigma(W_{12}a_i + b_1)\}]
$$

Since $|h_{v_1}^{(1)}(P_1^{(0)}(a_i)) - x_{v_1}(P_1^{(1)}(a_i))| < \epsilon_0$,

$$
|\sigma(w_2 \min(\sigma(W_{11}\beta + b_1), \sigma(W_{12}a_1 + b_1)) + b_2) - a_1| < \epsilon_0
$$

$$
\vdots
$$

$$
|\sigma(w_2 \min(\sigma(W_{11}\beta + b_1), \sigma(W_{12}a_4 + b_1)) + b_2) - a_4| < \epsilon_0
$$

Suppose $\sigma(W_{11}\beta + b_1) = \min\{\sigma(W_{11}\beta + b_1), \sigma(W_{12}a_i + b_1)\}$ and $\sigma(W_{11}\beta + b_1) = \min\{\sigma(W_{11}\beta + b_1), \sigma(W_{12}a_j + b_1)\}$ for $i \neq j$. Then $|\sigma(w_2\sigma(W_{11}\beta + b_1) + b_2) - a_i| < \epsilon_0$ and $|\sigma(w_2\sigma(W_{11}\beta + b_1) + b_2) - a_j| < \epsilon_0$ so $|a_i - a_j| < 2\epsilon_0 < 2$. This is a contradiction because for any $i \neq j$, $|a_i - a_j| \geq 2$.

Suppose that $\sigma(W_{11}\beta + b_1) = \min\{\sigma(W_{11}\beta + b_1), \sigma(W_{12}a_{i_1} + b_1)\}$ and $\sigma(W_{11}\beta + b_1) = \min\{\sigma(W_{11}\beta + b_1), \sigma(W_{12}a_{i_2} + b_1)\}$ for $i_1, i_2 \in \{1, 2, 3, 4\}$ for $i_1 \neq i_2$. This implies that $|a_{i_1} - a_{i_2}| < 2$ which is a contradiction. Thus, w.l.o.g. we can assume that for

$i \in \{1,2,3\}$, $\sigma(W_{12}a_i + b_1) = \min\{\sigma(W_{11}\beta + b_1), \sigma(W_{12}a_i + b_1)\}$. Additionally, suppose $W_{12}a_i + b_1 < 0$ and $W_{12}a_j + b_1 < 0$ for $i \neq j$ and $i, j \in \{1,2,3\}$. Then, $h_{v_1}(P_1^{(0)}(a_i)) = \sigma(b_2)$ and $h_{v_1}(P_1^{(0)}(a_j)) = \sigma(b_2)$ so $|\sigma(b_2) - a_i| < \epsilon_0$ and $|\sigma(b_2) - a_j| < \epsilon_0$. From here, we get that $|a_i - a_j| < 2$, which is a contradiction. Therefore, we assume that $\sigma(W_{12}a_i + b_1) = \min\{\sigma(W_{11}\beta + b_1), \sigma(W_{12}a_{i_1} + b_1)\}$ and $W_{12}a_i + b_1 > 0$ for $i = \{1,2\}$.

Then we have

$$|(a_1 w_2 W_{12} + w_2 b_1 + b_2) - a_1| \leq \epsilon_0$$

and

$$|(a_2 w_2 W_{12} + w_2 b_1 + b_2) - a_2| \leq \epsilon_0.$$

Note that $f(a) = a w_2 W_{12} + w_2 b_1 + b_2$ is an affine function with slope $m = w_2 W_{12}$ and intercept $w_2 b_1 + b_2$. As $|a_1|, \ldots, |a_4| < 9$ and $|a_i - a_j| > 2$, by Lemma B.3,

$$|w_2 W_{11} - 1| < \epsilon_0 < \frac{\epsilon}{2}$$

$$|w_2 b_1 + b_2| < 2 \cdot (1 + 9)\epsilon_0 = 20\epsilon_0 = \epsilon.$$

A parallel method using $\mathcal{H}_{\text{small}}$ with $a_i = 2i$ for $i \in \{5,6,7,8\}$ follows for bounding $|w_2 W_{11} - 1| < \frac{\epsilon}{2}$.

(ii) Now, we turn our attention to part (ii) and show that $w_2, W_{11}, W_{12} \geq 0$. Suppose $w_2 < 0$, which implies $W_{11}, W_{12} < 0$ as otherwise $w_2 W_{11}, w_2 W_{12} < 0$ and $|w_2 W_1 - \mathbb{1}| > \epsilon$ (recall $0 < \epsilon < 1$). The computed node feature for $v_1 \in V(P_1^{(0)}(a_i))$ is then

$$h_{v_1}^{(1)} = \sigma(w_2 \min\{\sigma(W_{11}x_s + W_{12}x_{(s,v_1)} + b_1), \sigma(W_{11}x_{v_1} + W_{12}x_{(v_1,v_1)} + b_1)\} + b_2)$$
$$= \sigma(w_2 \min\{\sigma(W_{11}a_i + b_1), \sigma(W_{11}\beta + b_1)\} + b_2)$$
$$= \sigma(w_2 \sigma(W_{11}\beta + b_1)\} + b_2).$$

Note that the above inequality follows from the fact that $W_{11}a_i + b_1 > W_{11}\beta + b_1$ since $a_i < \beta$ and $W_{11} < 0$. For $v_1 \in V(P_1^{(0)}(a_i))$ for all $i \in \{1,2,3,4\}$, $h_{v_1}^{(1)} = \sigma(w_2 \sigma(W_{11}\beta + b_1)\} + b_2)$. However, this is a contradiction, since $|a_i - a_j| \geq 2$ for all $i \neq j$ where $i, j \in \{1, \ldots, 4\}$. Thus, $w_2, W_{11}, W_{12} \geq 0$.

(iii) We will now show that given any $G \in \mathcal{G}$ and $v \in V(G)$, a neural network with the weights given in part (i) will approximately yield a single step of Bellman-Ford i.e.

$$(1 - \epsilon)h_v^{(1)}(G) - \epsilon \leq h_v^{(1)}(G) \leq (1 + \epsilon)x_v(G) + \epsilon.$$

Since $\|w_2 W_1 - \mathbb{1}\|_1 < \epsilon$ from part (i), we know that $|w_2 W_{11} - 1| < \epsilon$ and $|w_2 W_{12} - 1| < \epsilon$. Additionally, we know that for any $G \in \mathcal{G}$ and $v \in V(G)$,

$$h_v^{(1)}(G) = \sigma(w_2 \min\{\sigma(W_{11}x_u + W_{12}x_{(u,v)} + b_1) : u \in \mathcal{N}(v)\} + b_2)$$

From (ii), we know that $W_{11} \geq 0$, $W_{12} \geq 0$, and $W_{11}x_u + W_{12}x_{(u,v)} + b_1 > 0$ so the ReLU activation function $\sigma$ can be removed from the aggregation MLP i.e.,

$$h_v^{(1)}(G) = \sigma(w_2 \min\{W_{11}x_u + W_{12}x_{(u,v)} + b_1 : u \in \mathcal{N}(v)\} + b_2).$$

Suppose

$$u' = \text{argmin}_{u \in \mathcal{N}(v)}\{W_{11}x_u + W_{12}x_{(u,v)} + b_1\}$$

and

$$u^* = \text{argmin}_{u \in \mathcal{N}(v)}\{x_u + x_{(u,v)}\}.$$

Note that $x_v(\Gamma(G)) = x_{u^*} + x_{(u^*,v)}$. Then,

$$h_v^{(1)}(G) = \sigma(w_2(W_{11}x_{u'} + W_{12}x_{(u',v)} + b_1) + b_2)$$
$$\leq \sigma(w_2 W_{11}x_{u^*} + w_2 W_{12}x_{(u*,v)} + w_2 b_1 + b_2)$$
$$\leq \sigma((1 + \epsilon)(x_{u^*} + x_{(u^*,v)}) + w_2 b_1 + b_2)$$

Note that if $w_2 b_1 + b_2 \leq 0$, then

$$h_v^{(1)}(G) \leq \sigma((1+\epsilon)(x_{u^*} + x_{(u^*,v)}) + w_2 b_1 + b_2) \leq \sigma((1+\epsilon)(x_{u^*} + x_{(u^*,v)})) = (1+\epsilon)(x_{u^*} + x_{(u^*,v)}).$$

If $w_2 b_1 + b_2 > 0$, then

$$\begin{aligned}
h_v^{(1)}(G) &\leq \sigma((1+\epsilon)(x_{u^*} + x_{(u^*,v)}) + w_2 b_1 + b_2) \\
&\leq (1+\epsilon)(x_{u^*} + x_{(u^*,v)}) + \epsilon \\
&= (1+\epsilon)x_v(\Gamma(G)) + \epsilon.
\end{aligned}$$

In both cases, $h_v^{(1)}(G) \leq (1+\epsilon)x_v(\Gamma(G)) + \epsilon$.

Now, we consider the lower bound and show that $(1-\epsilon)x_v(\Gamma(G)) - \epsilon < h_v^{(1)}(G)$. By the definition of $u^*$,

$$x_{u^*} + x_{u^*,v} \leq x_{u'} + x_{(u',v)}$$

Note that because $0 < \epsilon < 1$, we have $0 < 1 - \epsilon < 1$ and $\frac{1}{1-\epsilon} > 1$. We will consider two cases: when $w_2 b_1 + b_2 \geq 0$ and when $w_2 b_1 + b_2 < 0$. Let $w_2 b_1 + b_2 > 0$. Then

$$\begin{aligned}
x_{u^*} + x_{u^*,v} &\leq x_{u'} + x_{(u',v)} + w_2 b_1 + b_2 \\
&\leq \left(\frac{1-\epsilon}{1-\epsilon}\right)x_{u'} + \left(\frac{1-\epsilon}{1-\epsilon}\right)x_{(u',v)} + \left(\frac{1-\epsilon}{1-\epsilon}\right)(w_2 b_1 + b_2) \\
&\leq \left(\frac{1}{1-\epsilon}\right)\cdot(1-\epsilon)x_{u'} + \left(\frac{1}{1-\epsilon}\right)\cdot(1-\epsilon)x_{(u',v)} + \left(\frac{1}{1-\epsilon}\right)(w_2 b_1 + b_2) \\
&\leq \left(\frac{1}{1-\epsilon}\right)\left((1-\epsilon)x_{u'} + (1-\epsilon)x_{(u',v)} + w_2 b_1 + b_2\right) \\
&\leq \left(\frac{1}{1-\epsilon}\right)\cdot(w_2 W_{11} x_{u'} + w_2 W_{12} x_{(u',v)} + w_2 b_1 + b_2) \\
&= \left(\frac{1}{1-\epsilon}\right)\cdot h_v(G)
\end{aligned}$$

Therefore,

$$(1-\epsilon)(x_{u^*} + x_{u^*,v}) = (1-\epsilon)x_v(\Gamma(G)) \leq h_v(G).$$

Let $w_1 b_1 + b_2 < 0$. We know that

$$w_1 W_{11} x_{u'} + w_2 W_{12} x_{(u',v)} + w_2 b_1 + b_2 \leq w_1 W_{11} x_{u'} + w_1 W_{12} x_{(u',v)} + w_2 b_1 + b_2 + \epsilon$$

Since $|w_1 b_1 + b_2| < \epsilon$, $w_1 b_1 + b_2 + \epsilon > 0$. Therefore,

$$\begin{aligned}
x_{u^*} + x_{u^*,v} &\leq x_{u'} + x_{(u',v)} + w_2 b_1 + b_2 + \epsilon \\
&\leq \left(\frac{1-\epsilon}{1-\epsilon}\right)x_{u'} + \left(\frac{1-\epsilon}{1-\epsilon}\right)x_{(u',v)} + \left(\frac{1-\epsilon}{1-\epsilon}\right)((w_2 b_1 + b_2) + \epsilon) \\
&\leq \left(\frac{1-\epsilon}{1-\epsilon}\right)x_{u'} + \left(\frac{1-\epsilon}{1-\epsilon}\right)x_{(u',v)} + \left(\frac{1}{1-\epsilon}\right)((w_2 b_1 + b_2) + \epsilon) \\
&\leq \left(\frac{1}{1-\epsilon}\right)(h_v(G) + \epsilon)
\end{aligned}$$

Thus,

$$(1-\epsilon)(x_{u^*} + x_{u^*,v}) - \epsilon \leq h_v(G)$$

$\square$

## C  SPARSITY REGULARIZED DEEP GNNS IMPLEMENT BF

In this section we analyze GNNs that are large both in their number of layers and the size of their respective MLPs. The key to showing these complex GNNs implement the BF algorithm is the introduction of sparsity regularization to the loss. With this type of regularization we can show any solution that approximates the global minimum must have only a few non-zero parameters.

Furthermore, any GNN with so few non-zero parameters can solve shortest path problems only via the BF algorithm. In short, although the model is over-parameterized, solutions approximating the global minimum are not.

Our overarching approach is as follows. We first give an implementation of BF by GNN with a small number of non-zero parameters $S$. Next, we show that, on our constructed training set, any GNN with less than $S$ non-zero parameters has large error. This allows us to conclude that the global minimum of the sparsity regularized loss must have exactly $S$ non-zero parameters. This sparsity allows us to simplify the MinAgg GNN update to include only a few parameters. We then derive approximations to these parameters which show the MinAgg GNN must be implementing BF algorithm, up to some scaling factor.

A key strategy in this section is to track the dependencies of the functions $f^{\mathrm{agg},(\ell)}$ and $f^{\mathrm{up},(\ell)}$ on their components. In particular, we say a function $f$ depends on a component or set of components if it is not constant over these components. Note that inputs to these functions are always non-negative (they are always proceeded by a ReLU), so by constant we mean constant over all non-negative values. The precise definitions are as follows.

**Definition C.1.** *For $\ell \in [L]$ the function $f^{\mathrm{agg},(\ell)}$* **depends on its node component** *iff it is not constant over its first $d_{\ell-1}$ components, i.e., there exits $x, y \in \mathbb{R}_{\geq 0}^{d_{\ell-1}+1}$ with $x \neq y$ and $x_{d_{\ell-1}+1} = y_{d_{\ell-1}+1}$ such that*

$$f^{\mathrm{agg},(\ell)}(x) \neq f^{\mathrm{agg},(\ell)}(y).$$

*The function $f^{\mathrm{agg},(\ell)}$* **depends on its edge component** *iff it is not constant over its edge component (the $(d_{\ell-1}+1)$th component). That is, there exits $x, y \in \mathbb{R}_{\geq 0}^{d_{\ell-1}+1}$ with $x \neq y$ and $x_i = y_i$ for $i \in \{1, \ldots, d_{\ell-1}\}$ such that*

$$f^{\mathrm{agg},(\ell)}(x) \neq f^{\mathrm{agg},(\ell)}(y).$$

**Definition C.2.** *For $\ell \in [L]$ the function $f^{\mathrm{up},(\ell)}$* **depends on its aggregation component** *iff it is not constant over its first $d$ components, i.e., there exits $x, y \in \mathbb{R}_{\geq 0}^{d+d_{\ell-1}}$ with $x \neq y$ and $x_i = y_i$ for $i \in \{d+1, \ldots, d_{\ell-1}\}$ such that*

$$f^{\mathrm{up},(\ell)}(x) \neq f^{\mathrm{up},(\ell)}(y).$$

*The function $f^{\mathrm{up},(\ell)}$* **depends on its skip component** *iff it is not constant over its last $d_{\ell-1}$ components, i.e., there exits $x, y \in \mathbb{R}_{\geq 0}^{d+d_{\ell-1}}$ with $x \neq y$ and $x_i = y_i$ for $i \in \{1, \ldots, d\}$ such that*

$$f^{\mathrm{up},(\ell)}(x) \neq f^{\mathrm{up},(\ell)}(y).$$

Our approach proceeds by showing that requisite dependencies can only be achieved if there is some minimal number of non-zero entries in $\theta$.

## C.1 IMPLEMENTING BF

We begin by showing there is a choice of parameters that makes the MinAgg GNN implement $K$ steps of the BF algorithm.

**Lemma C.3.** *Let $L \geq K > 0$. For an $L$-layer MinAgg GNN $\mathcal{A}_\theta$ with $m$-layer update and aggregation MLPs and parameters $\theta$, there is an assignment of $\theta$ with $mL + mK + K$ non-zero values such that $\mathcal{A}_\theta$ implements $K$ steps of the BF algorithm, i.e., for any $G \in \mathcal{G}$*

$$\mathcal{A}_\theta(G) = \Gamma^K(G).$$

*Proof.* We proceed by assigning parameters to $\mathcal{A}_\theta$ such that $\mathcal{A}_\theta$ simulates $K$ steps of Bellman-Ford, i.e., for any $G \in \mathcal{G}$, $\mathcal{A}_\theta(G) = \Gamma^K(G)$. For $\ell \in [L]$, let $f^{\mathrm{agg},(\ell)} : \mathbb{R}^{d_{\ell-1}+1} \to \mathbb{R}^d$ and $f^{\mathrm{up},(\ell)} : \mathbb{R}^{d+d_{\ell-1}} \to \mathbb{R}^d$ be the $m$-layer update and aggregation MLPs respectively. Note that $d_0 = d_L = 1$, and for $\ell \in \{1, \ldots, L-1\}$ the hidden layer dimension is $d_\ell = d$, for some arbitrary $d \geq 1$. The parameters for $f^{\mathrm{agg},(\ell)}$ and $f^{\mathrm{up},(\ell)}$ are $\{(W_j^{\mathrm{agg},(\ell)}, b_j^{\mathrm{agg},(\ell)})\}_{j\in[m]}$ and $\{(W_j^{\mathrm{up},(\ell)}, b_j^{\mathrm{up},(\ell)})\}_{j\in[m]}$,

respectively, where for $j \in [m]$,

$$W_j^{\mathrm{agg},(\ell)} \in \mathbb{R}^{d_j^{\mathrm{agg},(\ell)} \times d_{j-1}^{\mathrm{agg},(\ell)}}$$

$$b_j^{\mathrm{agg},(\ell)} \in \mathbb{R}^{d_j^{\mathrm{agg},(\ell)}}$$

$$W_j^{\mathrm{up},(\ell)} \in \mathbb{R}^{d_j^{\mathrm{up},(\ell)} \times d_{j-1}^{\mathrm{up}(\ell)}}$$

$$b_j^{\mathrm{up},(\ell)} \in \mathbb{R}^{d_j^{\mathrm{up},(\ell)}}.$$

The dimension of these parameters are

$$d_j^{\mathrm{agg},(\ell)} = \begin{cases} d_{\ell-1} + 1 \text{ if } j = 0 \\ d \text{ otherwise} \end{cases}$$

$$d_j^{\mathrm{up},(\ell)} = \begin{cases} d_{\ell-1} + d \text{ if } j = 0 \\ d_\ell \text{ if } j = m \\ d \text{ otherwise} \end{cases}.$$

Now we give values of these parameters that make $\mathcal{A}_\theta$ implement the BF algorithm. Let $b_j^{\mathrm{up},(\ell)} = \mathbf{0}$ and $b_j^{\mathrm{agg},(\ell)} = \mathbf{0}$ for all $\ell \in [L]$ and $j \in [m]$. Set

$$W_1^{\mathrm{agg},(1)} = \begin{pmatrix} 1 & 1 \\ \vdots & \vdots \\ 0 & 0 \end{pmatrix}$$

and for $\ell \in \{2, \ldots, K\}$

$$W_1^{\mathrm{agg},(\ell)} = \begin{pmatrix} 1 & 0 & \ldots & 0 & 1 \\ 0 & 0 & \ldots & 0 & 0 \\ \vdots & \vdots & & & \vdots \\ 0 & 0 & \ldots & & 0 \end{pmatrix}. \tag{14}$$

This choice makes $W_1^{\mathrm{agg},(\ell)}$ sum the edge weight and first component of the node feature into the first component of the resulting vector. That is, $[W_1^{\mathrm{agg},(\ell)}(h_v^{(\ell-1)} \oplus x_{(v,u)})]_1 = [h_v^{(\ell-1)}]_1 + x_{(v,u)}$. Next, set

$$W_j^{\mathrm{agg},(\ell)} = \begin{pmatrix} 1 & 0 & \ldots & 0 \\ 0 & \ddots & & \vdots \\ \vdots & & & \\ 0 & \ldots & & 0 \end{pmatrix} \quad \text{for } \ell \in [K] \text{ and } j \in \{2, \ldots, m\},$$

$$W_j^{\mathrm{up},(\ell)} = \begin{pmatrix} 1 & 0 & \ldots & 0 \\ 0 & \ddots & & \vdots \\ \vdots & & & \\ 0 & \ldots & & 0 \end{pmatrix} \quad \text{for } \ell \in [K] \text{ and } j \in [m].$$

Finally, for $\ell \in \{K+1, \ldots, L\}$, let

$$W_j^{\mathrm{agg},(\ell)} = \mathbf{0} \quad \text{for } j \in [m]$$

$$W_1^{\mathrm{up},(\ell)} = \begin{pmatrix} 0 & 0 & \ldots & 1 \\ 0 & \ddots & & \vdots \\ \vdots & & & \\ 0 & \ldots & & 0 \end{pmatrix}$$

$$W_j^{\mathrm{up},(\ell)} = \begin{pmatrix} 1 & 0 & \ldots & 0 \\ 0 & \ddots & & \vdots \\ \vdots & & & \\ 0 & \ldots & & 0 \end{pmatrix} \quad \text{for } j \in \{2, \ldots, m\}.$$

Given the above assignments of $W_j^{\text{agg},(\ell)}$ and $W_j^{\text{up},(\ell)}$, the final $L - K$ layers implement the identity on the first component of the node feature, i.e., $[h_v^{(\ell)}]_1 = [h_v^{(\ell-1)}]_1$ for $\ell \in \{K+1, \ldots, L\}$. Since edge weights are always non-negative and there are no negative parameters in the above, the ReLU activations can be ignored. Then, for all $v \in V$ and for $\ell \leq K$, we get

$$[h_v^{(\ell)}]_1 = \min\{[h_u^{(\ell-1)}]_1 + x_{(u,v)} \mid u \in \mathcal{N}(v)\}$$

which is the BF algorithm update. This implies, by the correctness of the BF algorithm, that $[h_v^{(K)}]_1$ is the $K$-step shortest path distance. The last $L - K$ layers of the GNN implement the identity function so $h_v^{(L)} = [h_v^{(L)}]_1 = [h_v^{(K)}]_1$ is also the $K$-step shortest path distances. Perfect accuracy is then achieved on all $K$-step shortest path instances. $\square$

We later show that the requirement $L \geq K$ is indeed necessary (Corollary C.13).

## C.2 TRAINING SET

Our training set is comprised of multiple parts, which we describe in this subsection. The first set of training instances is used to regulate how the MinAgg GNN scales features throughout computation.

**Definition C.4.** *For $k \in [K]$, define $\mathscr{H}_{k,K}$ as*

$$\mathscr{H}_{k,K} = \{P_{K+1}^{(1)}(a, 0, \ldots, 0, b, 0, \ldots, 0) : (a, b) \in \{0, 1, \ldots, 2K\} \times \{0, 2K+1\}\}$$

*where $P_{K+1}^{(1)}(a, 0, \ldots, 0, b, 0, \ldots, 0)$ is the attributed $K$-edge path graph with weight $a$ for the first edge, weight $b$ for the $(k+1)$th edge, and weight zero for all other edges.*

Next, we define graph that is used to show the MinAgg GNN must have at least $K$ steps that depend on both edge weights and neighboring node features. If these conditions are not met, then the MinAgg GNN is not expressive enough to compute the shortest path distances in this graph.

**Definition C.5.** *Let $H^{(0),K}$ be a 0-step BF instance with $2K + 2$ vertices*

$$V = \{v_0, v_1, \ldots v_K\} \cup \{u_0, u_1, \ldots, u_K\},$$

*edges*

$$E = \{(v_{i-1}, v_i) \mid i \in [K]\} \cup \{(u_{i-1}, u_i) \mid i \in [K]\} \cup \{\{(u_{i-1}, v_i) \mid i \in [K]\} \cup \{(v_{i-1}, u_i) \mid i \in [K]\},$$

*edge features $X_{\text{e}}$ given by*

$$x_{(w,q)} = \begin{cases} 1 \text{ if } (w,q) = (u_{k-1}, v_k) \text{ or } (w,q) = (v_{k-1}, u_k) \text{ for } k \in [K] \\ 0 \text{ otherwise} \end{cases},$$

*and initial node features $X_{\text{v}}$ given by*

$$x_w = \begin{cases} 0 \text{ if } w = v_0 \\ \beta \text{ otherwise} \end{cases}.$$

We also write $H_K^{(K)} = \Gamma^K(H_K^{(0)})$.

The complete training set also includes $P_1^{(0)}(1), P_2^{(1)}(1, 0)$.

**Definition C.6.** *For $K > 1$, we let*

$$\mathscr{G}_{scale,K} = \cup_{K \geq k > 1} \mathscr{H}_{k,K}$$
$$\mathscr{G}_K = \mathscr{G}_{scale,K} \cup \{P_1^{(0)}(1), P_2^{(1)}(1, 0), H_K^{(0)}\}$$
$$\mathcal{G}_K = \{(G^{(t)}, \Gamma(G^{(t)})) : G^{(t)} \in \mathscr{G}_K\}.$$

Note the distinction here between $\mathscr{G}_K$ which is a set of graphs and $\mathcal{G}_K$, which contains pairs of graphs (an input graph and a target graph).

## C.3 SPARSITY STRUCTURE

Here we show that the training set $\mathcal{G}_K$ requires a MinAgg GNN to have a minimal sparsity to achieve good performance. Furthermore, this non-zero parameters must follow a particular structure.

**Definition C.7.** *An **isomorphism** between two attributed graphs $G = (V, E, X_v, X_e)$ and $G' = (V', E', X'_v, X'_e)$ is a bijection $\phi : V \to V'$ satisfying*

$$(u, v) \in E \text{ if and only if } (\phi(u), \phi(v)) \in E'$$

*and*

$$x_v = x'_{\phi(v)} \quad \forall v \in V$$
$$x_{(v,u)} = x'_{(\phi(v),\phi(u))} \quad \forall (v, u) \in E.$$

**Fact C.8.** *Suppose $\phi$ is an isomorphism between two attributed graphs $G = (V, E, X_v, X_e)$ and $G' = (V', E', X'_v, X'_e)$. Then $\phi$ is also an isomorphism between $A_\theta(G)$ and $A_\theta(G')$.*

**Definition C.9.** *For an $L$-layer MinAgg GNN $\mathcal{A}_\theta$, we say that a layer $\ell \in [L]$ is **message passing** if the aggregation function $f^{\mathrm{agg},(\ell)}$ depends on its node component. A **edge-dependent message passing layer** is a message passing layer for which $f^{\mathrm{agg},(\ell)}$ also depends on its edge component. A layer is **stationary** if it is not message passing*

The update at each node can only depend on neighboring node features through the node component of the aggregation function. Thus, for stationary layers, each updated node feature only depends on the previous value of the node feature and the value of adjacent edges.

**Fact C.10.** *Consider a MinAgg GNN $\mathcal{A}_\theta$ such that its $\ell$th layer is stationary. Then, taking $\theta$ to be fixed, the feature $h_v^{(\ell)}$ is only a function of $h_v^{(\ell-1)}$ and $x_{(u,v)}$ for $u \in \mathcal{N}(v)$.*

The importance of message passing layers is that these are the only layers for which an updated node depends on the previous features of its neighboring nodes. This is made precise by the following statement.

**Claim C.11.** *Consider a MinAgg GNN $A_\theta$ acting on a graph $G = (V, E, X_v, X_e)$ and consider two nodes $v, w \in V$ such that $v$ is $j$ steps away from $w$. Suppose for some $\ell \in [L]$ there are $k$ layers in $[\ell]$ that are message passing with $k < j$. Then the feature $h_v^{(\ell)}(G)$ is independent of $x_w(G)$. That is, for any graph $G' = (V, E, X_v, X'_e)$ that differs from $G$ only in the feature $x_w(G')$, we have $h_v^{(\ell)}(G) = h_v^{(\ell)}(G')$. Similarly, for any edge $(u, w) \in E$ if both $u$ and $w$ are $j$ steps away from $v$ then $h_v^{(\ell)}(G)$ is independent of $x_{(u,w)}(G)$.*

*Proof.* We proceed by induction so assume the statement holds for $j - 1$. Note the base case of $j = 1$ is immediate from Fact C.10 as this implies no message passing has occurred. Suppose $v$ is $j$ steps away from $w$. Let $\ell'$ be the largest $\ell$ in $\{1, \ldots, \ell\}$ that is message passing. By definition of the MinAgg GNN,

$$h_v^{(\ell')} = f^{\mathrm{up},(\ell')}\Big(\min\{f^{\mathrm{agg},(\ell')}(h_u^{(\ell'-1)} \oplus x_{(u,v)}) : u \in \mathcal{N}(v)\}\Big). \tag{15}$$

Every node in $\mathcal{N}(v)$ is at least $j - 1$ steps from $w$ and there are at most $k - 1$ message passing steps in $[\ell' - 1]$. Invoking the inductive hypothesis for $j - 1$ yields that $h_u^{(\ell'-1)}$ for $u \in \mathcal{N}(v)$ is independent of $x_u(G)$. Thus, since every variable in the expression for $h_v^{(\ell')}$ is independent of $x_u(G)$, the feature $h_v^{(\ell')}$ is independent as well. Finally, if $h_v^{(\ell')}$ is independent of $x_u(G)$ then by Fact C.10, $h_v^{(\ell)}$ is also independent of $x_u(G)$ since all the layers between $\ell'$ and $\ell$ are stationary. (If $\ell$ is message passing then $\ell' = \ell$.)

A parallel argument can be used to show independence of $h_v^{(\ell)}$ from $x_{(u,w)}$ in the case that $v$ is $j$ steps away from both $u$ and $w$ $\qquad\square$

**Lemma C.12.** *An $L$-layer MinAgg GNN $\mathcal{A}_\theta$ satisfies*

$$h_{v_K}^{(L)}(H_K^{(0)}) \neq h_{u_K}^{(L)}(H_K^{(0)})$$

*only if it has at least $K$ edge-dependent message passing layers.*

*Proof.* We proceed by proving claim $(*)$, regarding the output of the MinAgg GNN on $H_K^{(0)}$. This claim is shown through two cases. Note that since in this proof only the graph $H_K^{(0)}$ is considered, we suppress notation referring to the graph for simplicity.

Claim $(*)$ Let $k' \in [K]$ and $\ell \in [L]$. If for all $k \geq k'$,

$$h_{v_k}^{(\ell-1)} = h_{u_k}^{(\ell-1)},$$

and $f^{\mathrm{agg},(\ell)}$ is not edge-dependent message passing, then for all $k \geq k'$,

$$h_{v_k}^{(\ell)} = h_{u_k}^{(\ell)}.$$

If $f^{\mathrm{agg},(\ell)}$ is not edge-dependent message passing, then it either does not depend on its node component or does not depend on its edge component.

Case I: $f^{\mathrm{agg},(\ell)}$ **does not depend on its node component.** For $k \geq k'$,

$$h_{v_k}^{(\ell)} = f^{\mathrm{up},(\ell)}\Big( \min\{f^{\mathrm{agg},(\ell)}(h_w^{(\ell-1)} \oplus x_{(w,v_k)}) : w \in \{v_{k-1}, u_{k-1}, v_k, v_{k+1}, u_{k+1}\}\} \oplus h_{v_k}^{(\ell-1)}\Big)$$

$$= f^{\mathrm{up},(\ell)}\Big( \min\{f^{\mathrm{agg},(\ell)}(\cdot \oplus 0), f^{\mathrm{agg},(\ell)}(\cdot \oplus 1)\} \oplus h_{v_k}^{(\ell-1)}\Big)$$

where a center dot $\cdot$ is used to indicate the lack of dependence on the node feature. (Any value can replace $\cdot$ and the expression does not change since $f^{\mathrm{agg},(\ell)}$ is constant over its node component.) The key here is that since $f^{\mathrm{agg},(\ell)}$ does not depend on its node component, only the set of edge weights incident on $v_k$, which is $\{0, 1\}$, effects the value of

$$\min\{f^{\mathrm{agg},(\ell)}(h_w^{(\ell-1)} \oplus x_{(w,v_k)}) : w \in \{v_{k-1}, u_{k-1}, v_k, v_{k+1}, u_{k+1}\}\}.$$

Similarly, for $u_k$ we have

$$h_{u_k}^{(\ell)} = f^{\mathrm{up},(\ell)}\Big( \min\{f^{\mathrm{agg},(\ell)}(h_w^{(\ell-1)} \oplus x_{(w,u_k)}) : w \in \{v_{k-1}, u_{k-1}, u_k, v_{k+1}, u_{k+1}\}\} \oplus h_{u_k}^{(\ell-1)}\Big)$$

$$= f^{\mathrm{up},(\ell)}\Big( \min\{f^{\mathrm{agg},(\ell)}(\cdot \oplus 0), f^{\mathrm{agg},(\ell)}(\cdot \oplus 1)\} \oplus h_{u_k}^{(\ell-1)}\Big)$$

and since $h_{v_k}^{(\ell-1)} = h_{v_k}^{(\ell-1)}$, we get $h_{u_k}^{(\ell)} = h_{v_k}^{(\ell)}$.

Case II: $f^{\mathrm{agg},(\ell)}$ **does not depend on its edge component.**
For $k \geq k'$,

$$h_{v_k}^{(\ell)} = f^{\mathrm{up},(\ell)}\Big( \min\{f^{\mathrm{agg},(\ell)}(h_w^{(\ell-1)} \oplus x_{(w,v_k)}) : w \in \{v_{k-1}, u_{k-1}, v_k, v_{k+1}, u_{k+1}\}\} \oplus h_{v_k}^{(\ell-1)}\Big)$$

$$= f^{\mathrm{up},(\ell)}\Big( \min\{f^{\mathrm{agg},(\ell)}(h_w^{(\ell-1)} \oplus \cdot) : w \in \{v_{k-1}, u_{k-1}, v_k, v_{k+1}, u_{k+1}\}\} \oplus h_{v_k}^{(\ell-1)}\Big)$$

and

$$h_{u_k}^{(\ell)} = f^{\mathrm{up},(\ell)}\Big( \min\{f^{\mathrm{agg},(\ell)}(h_w^{(\ell-1)} \oplus x_{(w,u_k)}) : w \in \{v_{k-1}, u_{k-1}, u_k, v_{k+1}, u_{k+1}\}\} \oplus h_{u_k}^{(\ell-1)}\Big)$$

$$= f^{\mathrm{up},(\ell)}\Big( \min\{f^{\mathrm{agg},(\ell)}(h_w^{(\ell-1)} \oplus \cdot) : w \in \{v_{k-1}, u_{k-1}, u_k, v_{k+1}, u_{k+1}\}\} \oplus h_{u_k}^{(\ell-1)}\Big).$$

However, since $h_{v_k}^{(\ell-1)} = h_{u_k}^{(\ell-1)}$, the minimums in the expressions for $h_{v_k}^{(\ell)}$ and $h_{u_k}^{(\ell)}$ are taken over the same set and so $h_{u_k}^{(\ell)} = h_{v_k}^{(\ell)}$.

Now that we have proved claim $(*)$ we proceed by induction to show that for any $\ell' \in [L]$ if there are $k'$ edge-dependent message passing layers in $[\ell']$ then $h_{u_k}^{(\ell')} = h_{v_k}^{(\ell')}$ for all $k > k'$. This holds for the base case of $k' = 0$ since at initialization $h_{v_k}^{(0)} = h_{u_k}^{(0)}$ for all $k \in [K]$, and $(*)$ yields that $h_{v_k}^{(\ell')} = h_{u_k}^{(\ell')}$ since no layer in $[\ell']$ is edge-dependent message passing. Now suppose the inductive hypothesis holds for $k' - 1$ edge-dependent message passing layers. That is, suppose that for any $\ell' \in [L]$ if there are $k' - 1$ edge-dependent message passing layers in $[\ell']$ then $h_{u_k}^{(\ell')} = h_{v_k}^{(\ell')}$ for all $k > k' - 1$. Now suppose for some $\ell' \in [L]$ there are $k'$ edge-dependent message passing layers in

$[\ell']$, and let $\ell^*$ be the last such layer. Then $h_{v_k}^{(\ell^*-1)} = h_{u_k}^{(\ell^*-1)}$ for all $k > k' - 1$ by the inductive hypothesis. F or all $k > k'$,

$$h_{v_k}^{(\ell^*)} = f^{\mathrm{up},(\ell^*)}\Big( \min\{f^{\mathrm{agg},(\ell^*)}(h_w^{(\ell^*-1)} \oplus x_{(w,v_k)}) : w \in \{v_{k-1}, u_{k-1}, v_k, v_{k+1}, u_{k+1}\}\} \oplus h_{v_k}^{(\ell^*-1)}\Big)$$

and

$$h_{u_k}^{(\ell^*)} = f^{\mathrm{up},(\ell^*)}\Big( \min\{f^{\mathrm{agg},(\ell^*)}(h_w^{(\ell^*-1)} \oplus x_{(w,u_k)}) : w \in \{v_{k-1}, u_{k-1}, u_k, v_{k+1}, u_{k+1}\}\} \oplus h_{u_k}^{(\ell^*-1)}\Big).$$

Recall that $x_{(w,w)}$ is always zero for any node $w$. The minimums in these expressions are then taken over the sets

$$S_v = \{h_{v_{k-1}}^{(\ell^*-1)} \oplus x_{(v_{k-1},v_k)}, h_{u_{k-1}}^{(\ell^*-1)} \oplus x_{(u_{k-1},v_k)}, h_{v_k}^{(\ell^*-1)} \oplus 0, h_{v_{k+1}}^{(\ell^*-1)} \oplus x_{(v_{k+1},v_k)}, h_{u_{k+1}}^{(\ell^*-1)} \oplus x_{(u_{k+1},v_k)}\}$$

and

$$S_u = \{h_{v_{k-1}}^{(\ell^*-1)} \oplus x_{(v_{k-1},u_k)}, h_{u_{k-1}}^{(\ell^*-1)} \oplus x_{(u_{k-1},u_k)}, h_{u_k}^{(\ell^*-1)} \oplus 0, h_{v_{k+1}}^{(\ell^*-1)} \oplus x_{(v_{k+1},u_k)}, h_{u_{k+1}}^{(\ell^*-1)} \oplus x_{(u_{k+1},u_k)}\}$$

respectively. Note, by the inductive hypothesis, $h_{v_{k-1}}^{(\ell^*-1)} = h_{u_{k-1}}^{(\ell^*-1)}$, $h_{v_k}^{(\ell^*-1)} = h_{u_k}^{(\ell^*-1)}$, and $h_{v_{k+1}}^{(\ell^*-1)} = h_{u_{k+1}}^{(\ell^*-1)}$, since $k - 1 > k' - 1$. Furthermore, for all $i \in [K]$, we have $x_{(v_{i-1},u_i)} = x_{(u_{i-1},v_i)} = 1$ and $x_{(v_{i-1},v_i)} = x_{(u_{i-1},u_i)} = 0$. Thus,

$$S_v = S_u = \{h_{u_{k-1}}^{(\ell^*-1)} \oplus 0, h_{u_{k-1}}^{(\ell^*-1)} \oplus 1, h_{u_k}^{(\ell^*-1)} \oplus 0, h_{u_{k+1}}^{(\ell^*-1)} \oplus 0, h_{u_{k+1}}^{(\ell^*-1)} \oplus 1\}.$$

so $h_{u_k}^{(\ell^*)} = h_{v_k}^{(\ell^*)}$ for $k > k'$. Finally, the remaining layers $\ell \in \{\ell^* + 1, \dots, \ell'\}$ are not edge-dependent message passing, so claim $(*)$ gives that $h_{u_k}^{(\ell)} = h_{v_k}^{(\ell)}$ is maintained for $k > k'$, completing the induction.

Taking the inductive hypothesis that we just proved with $\ell' = L$ and $k' < K$ gives that if there are are less than $K$ edge-dependent message passing layers then $h_{u_K}^{(L)} = h_{v_K}^{(L)}$. □

Note $x_{v_K}(H_K^{(K)}) \neq x_{u_K}(H_K^{(K)})$, so any MinAgg GNN with less than $K$ message passing layers can not achieve perfect accuracy on $H_K^{(0)}$. In particular, since a MinAgg GNN with less than $K$ layers must also have less than $K$ message passing layers, this lemma demonstrates that $K$ layers are indeed necessary for a MinAgg GNN to have perfect accuracy on $K$-step shortest path problems.

**Corollary C.13.** *Any MinAgg GNN with less than $K$ layers has non-zero error on the training pair* $(H_K^{(0)}, H_K^{(K)})$.

**Lemma C.14.** *An $L$-layer GNN $\mathcal{A}_\theta$ has output satisfying*

$$h_{v_0}^{(L)}(P_1^{(0)}(1)) \neq h_{v_1}^{(L)}(P_1^{(0)}(1)) \tag{16}$$

*only if for all $\ell \in [L]$ the function $f^{\mathrm{up},(\ell)}$ is not constant.*

*Proof.* Suppose $f^{\mathrm{up},(\ell)}$ is constant for some $\ell \in [L]$. Then the node features after the $\ell$th layer are identical: $h_{v_0}^{(\ell)} = h_{v_1}^{(\ell)}$. Consider the attributed graph at this layer $G^{(\ell)} = (V, E, X_{\mathrm{v}}^{(\ell)}, X_{\mathrm{e}})$ with node features $x_v = h_v^{(\ell)}$. This graph has

$$\phi : v_0 \mapsto v_1$$
$$v_1 \mapsto v_0$$

as an automorphism. By Fact C.8, $A_\theta(G)$ must also have $\phi$ as an automorphism. Thus, since isomorphisms respect node features (Def. C.7), $h_{v_0}^{(L)} = h_{\phi(v_0)}^{(L)} = h_{v_1}^{(L)}$.

□

We are now ready to show that any MinAgg GNN achieving small loss on a training set that includes $P_1^{(0)}(1)$ and $H_K^{(0)}$ must have a specific sparsity structure.

**Lemma C.15.** *For $K \geq 1$, let $\mathcal{G}_{\text{train}}$ be a set containing pairs of training instances $(G^{(t)}, \Gamma^K(G^{(t)}))$ where $\mathcal{G}_{\text{train}}$ contains $M$ total reachable nodes and*

$$\{(H_K^{(0)}, H_K^{(K)}), (P_1^{(0)}(1), P_1^{(K)}(1))\} \subset \mathcal{G}_{\text{train}}.$$

*For $L \geq K > 0$, consider an $L$-layer MinAgg GNN $\mathcal{A}_\theta$ with $m$-layer MLPs and parameters $\theta$. Given regularization coefficient $0 < \eta < \frac{1}{2M(mL+mK+K)}$ and error $0 \leq \epsilon < \eta$, then the loss*

$$\mathcal{L}_{\text{reg}} = \mathcal{L}_{\text{MAE}}(G_{\text{train}}, \mathcal{A}_\theta) + \eta\|\Theta\|_0 \tag{17}$$

*has a minimum value of $\eta(mL + mK + K)$ and if the loss achieved by $A_\theta$ is within $\epsilon$ of this minimum then $A_\theta$ has exactly $mL + mK + K$ non-zero parameters, $K$ message passing layers, and for all layers $f^{\text{up},(\ell)}$ is non-constant. In particular, for all $j \in [m]$ and $\ell \in [L]$*

$$\|W_j^{\text{up},(\ell)}\|_0 = 1,$$

*and $b_j^{\text{up},(\ell)} = b_j^{\text{agg},(\ell)} = 0$. For each of the $K$ message passing layers $\ell_k \in [L]$,*

$$\|W_1^{\text{agg},(\ell)}\|_0 = 2$$
$$\|W_j^{\text{agg},(\ell)}\|_0 = 1 \quad \text{for } m \geq j > 1$$

*where the two non-zero entries in $W_1^{\text{agg},(\ell)}$ share the same row.*

*Proof.* First, the BF implementation given by Lemma $C.3$ shows there is a choice of parameters that achieves perfect accuracy with $mL + mK + K$ non-zero values. This implementation achieves a loss of $\mathcal{L}_{\text{reg}} = \eta(mL + mK + K)$. Thus, $A_\theta$ can not have more than $mL + mK + K$ non-zero values as this would mean a loss at least $\eta > \epsilon$ greater than the loss achieved by the BF implementation. We now derive the sparsity structure that any MinAgg GNN achieving a loss no less than $\eta(mL + mK + K) + \epsilon$ must achieve. We show at the end of the proof that $\eta(mL + mK + K)$ is indeed the global minimum of the loss.

Note that $|x_{v_0}(P_1^{(K)}(1)) - x_{v_1}(P_1^{(K)}(1))| = 1$ and $|x_{v_K}(H_K^{(K)}) - x_{u_K}(H_K^{(K)})| = 1$. If $h_{v_0}^{(L)}(P_1^{(0)}(1)) = h_{v_1}^{(L)}(P_1^{(0)}(1))$ then

$$\mathcal{L}_{\text{reg}} \geq \mathcal{L}_{\text{MAE}}(\mathcal{G}_{\text{train}}, A_\theta)$$
$$\geq \frac{|x_{v_0}(P_1^{(K)}(1)) - h_{v_0}^{(L)}(P_1^{(0)}(1))| + |x_{v_1}(P_1^{(K)}(1)) - h_{v_1}^{(L)}(P_1^{(0)}(1))|}{M}$$
$$\geq \frac{1}{M}$$
$$\geq \eta(mL + mK + K) + \epsilon,$$

where the last inequality follows from $\epsilon < \frac{1}{2M(mL+mK+K)}$ and $\eta(mL + mK + K) < \frac{1}{2M}$. We can then conclude that any MinAgg GNN with loss less than or equal to $\eta(mL + mK + K) + \epsilon$ must satisfy $h_{v_k}^{(L)}(H_K^{(0)}) \neq h_{u_k}^{(L)}(H_K^{(0)})$ and $h_{v_0}^{(L)}(P_1^{(0)}(1)) \neq h_{v_1}^{(L)}(P_1^{(0)}(1))$ so by lemmas C.12 and C.14, any such MinAgg GNN has at least $K$ edge-dependent message passing layers and for all $\ell \in [L]$ the update function $f^{\text{up},(\ell)}$ is nonconstant.

Next, we analyze the sparsity required to achieve $K$ edge-dependent message passing layers and nonconstant $f^{\text{up},(\ell)}$.

- If $f^{\text{up},(\ell)}$ is non constant, then for all $j \in [m]$ and $\ell \in [L]$

$$\|W_j^{\text{up},(\ell)}\|_0 \geq 1 \tag{18}$$

  so there must be at least $mL$ non-zero entries coming from $f^{\text{up},(\ell)}$.

- If $\ell$ is a edge-dependent message passing layer then $f^{\text{agg},(\ell)}$ is nonconstant and so for all $j \in [m]$ and $\ell \in [L]$

$$\|W_j^{\text{agg},(\ell)}\|_0 \geq 1. \tag{19}$$

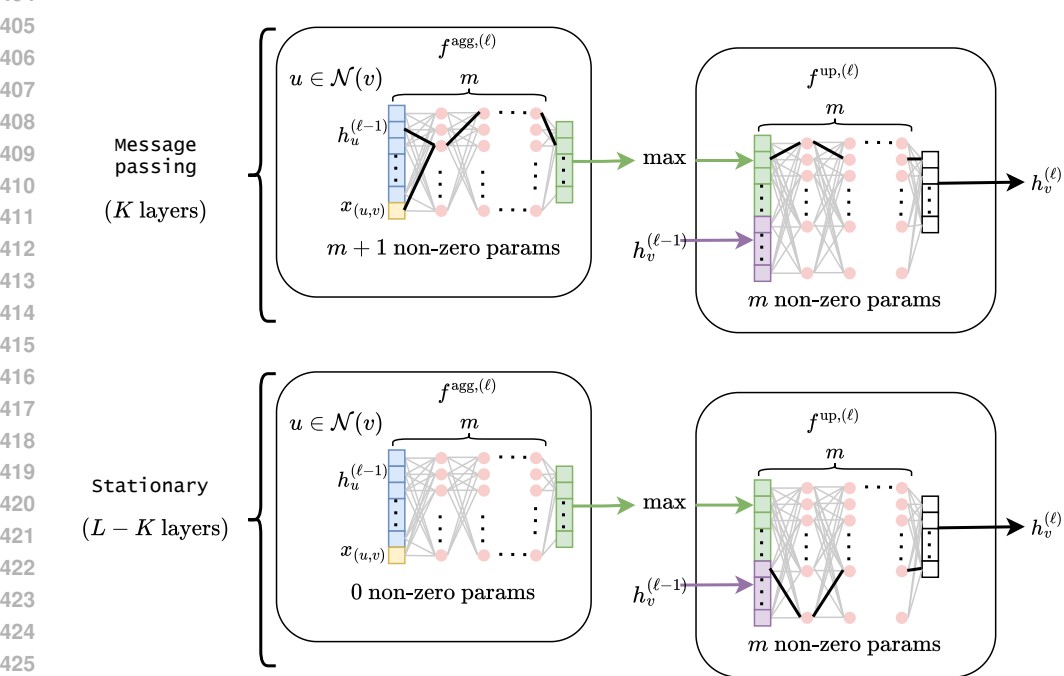

Figure 6: A diagram showing an example of a MinAgg GNN with the sparsity structure given by Lemma C.15. Bold black connections in the neural network indicate non zero parameters, while grey lines indicate zero parameters.

Also, since $f^{\mathrm{agg},(\ell)}$ is edge-dependent message passing it depends on both its node and its edge component so $W_1^{\mathrm{agg},(\ell)}$ must have two columns that have non-zero entries, giving

$$\|W_1^{\mathrm{agg},(\ell)}\|_0 \geq 2. \tag{20}$$

Thus, the total number of non-zero entries coming from $f^{\mathrm{agg},(\ell)}$ is at least $mK + K$ since there are at least $K$ edge-dependent message passing layers.

Combining the contributions from $f^{\mathrm{up},(\ell)}$ and $f^{\mathrm{agg},(\ell)}$ we have that $A_\theta$ has at least $mL + mK + K$ non-zero parameters. However, at the start of the proof we showed that $mL + mK + K$ is also an upper bound on the number of non-zero parameters, so there must be exactly $mL + mK + K$ non-zero parameters. Additionally, we can see that there can not be more than $K$ message passing layers as this would require additional non-zero parameters. Furthermore, if only $mL + mK + K$ parameters are non-zero then inequalities (18) - (20) must be tight and, also, $b_j^{\mathrm{up},(\ell)} = b_j^{\mathrm{agg},(\ell)} = 0$. Finally, we remark that the non-zero entries in $W_1^{\mathrm{agg},(\ell)}$ must share a row since $W_2^{\mathrm{agg},(\ell)}$ has only one non-zero entry. Thus, if the non-zero entries in $W_1^{\mathrm{agg},(\ell)}$ do not share a row, either edge dependence or node dependence is lost.

Furthermore, any MinAgg GNN that achieves a loss less than or equal to $\eta(mL + mK + K) + \epsilon$ must have exactly $mL + mK + K$ non-zero parameters. This implies that $\eta(mL + mK + K)$ is the minimum of the loss.

$\square$

We can now limit our analysis to MinAgg GNNs with exactly $K$ message passing layers. For these MinAgg GNNs let the $k$th message passing layer be $\ell_k \in [L]$ where $k \in [K]$.

## C.4 BOUNDING GNN EXPRESSIVITY

The following section aims to bound the expressiveness of $\mathcal{A}_\theta$ under certain conditions that are more challenging to analyze. The benefit is that with sufficiently many training examples we can restrict

our analysis to more straightforward cases, as for must training instances, these challenging to analyze conditions do not arise.

**Lemma C.16.** *Consider $P_{K+1}^{(1)}(x, a_2, \ldots, a_{K+1})$ where $a_2, \ldots, a_{K+1}$ are taken to be fixed and view the computed node feature $h_{v_K}^{(L)} : \mathbb{R} \to \mathbb{R}$ solely as a function of the first edge weight $x$, which is variable. Consider a MinAgg GNN $\mathcal{A}_\theta$ with exactly $K$ message passing layers. If for the $k$th message passing layer $\ell_k \in [L]$ there is a region $D_k \subset \mathbb{R}$ such that the feature $h_{v_{k+1}}^{(\ell_k)}(x)$ is constant on $D_k$, i.e., takes the same value for all $x \in D_k$, then $h_{v_{K+1}}^{(\ell_K)}(x)$ is also constant over $D_k$.*

*Proof.* We proceed by induction, so consider $k' \in [K]$ with $k' \geq k$ and assume that $h_{v_{k'}}^{(\ell_{k'-1})}(x)$ is constant over $D_k$. We aim to show $h_{v_{k'+1}}^{(\ell_{k'})}(x)$ is constant over $D_k$. This is true for the base case of $k' = k$ by the assumption in the theorem statement that $h_{v_{k+1}}^{(\ell_k)}(x)$ is constant on $D_k$. Now we prove the general case, so take $k' > k$. Since,

$$h_{v_{k'+1}}^{(\ell_{k'})} = f^{\text{up},(\ell_{k'})}\Big( \min\{f^{\text{agg},(\ell_{k'})}(h_u^{(\ell_{k'-1})} \oplus x_{(u,v)}) : u \in \{v_{k'}, v_{k'+1}, v_{k'+2}\}\} \oplus h_{v_{k'+1}}^{(\ell_{k'-1})}\Big), \quad (21)$$

$h_{v_{k'+1}}^{(\ell_{k'})}(x)$ is a function solely of $h_{v_{k'+1}}^{(\ell_{k'-1})}(x)$, $h_{v_{k'+2}}^{(\ell_{k'-1})}(x)$, and, $h_{v_{k'}}^{(\ell_{k'-1})}(x)$. (All edge weights other than that of $(v_0, v_1)$ are taken to be constant, and $k' > k \geq 1$ so there is no dependence on $x_{(v_0,v_1)}$ in the expression.) The feature $h_{v_{k'}}^{(\ell_{k'-1})}(x)$ is constant over $D_k$ by the inductive hypothesis. Furthermore, $h_{v_{k'}}^{(\ell_{k'-1})}(x)$ is constant over $D_k$ by Fact C.10 since all layers in $\{\ell_{k'-1}+1, \ldots, \ell_{k'-1}\}$ must be stationary. Note that while the stationary layers may change the node feature at $v_{k'}$ so that $h_{v_{k'}}^{(\ell_{k'-1})}(x) \neq h_{v_{k'}}^{(\ell_{k'-1})}(x)$, these two features $h_{v_{k'}}^{(\ell_{k'-1})}(x)$ and $h_{v_{k'}}^{(\ell_{k'-1})}$ are still both constant functions of $x$ on $D_k$. In $P_{K+1}^{(1)}(x, a_2, \ldots, a_{K+1})$ only the first edge weight $x_{(v_0,v_1)}$ and the second node feature $x_{v_1}$ depend on $x$. Thus, by Claim C.11, if $j \in [K]$ with $j > k'$ then $h_{v_{j+1}}^{(\ell_{k'})}(x)$ is constant across all $x \geq 0$ as $v_{j+1}$ is more than $k'$ steps from $v_1$. It then follows that the features $h_{v_{k'+1}}^{(\ell_{k'-1})}(x)$ and $h_{v_{k'+2}}^{(\ell_{k'-1})}(x)$ are both constant. Also, $h_{v_{k'+1}}^{(\ell_{k'-1})}(x)$ and $h_{v_{k'+2}}^{(\ell_{k'-1})}(x)$ are both constant since the layers in $\{\ell_{k'-1}+1, \ldots, \ell_{k'}-1\}$ must be stationary. We can then conclude $h_{v_{k'+1}}^{(\ell_{k'})}(x)$ is constant on $D_k$, since it depends only on variables which are constant on $D_k$, completing the inductive argument. Since we have proved $h_{v_{k'+1}}^{(\ell_{k'})}(x)$ is constant on $D_k$ for all $k' \geq k$, we have that $h_{v_{K+1}}^{(\ell_K)}(x)$ is constant. Finally, since $h_{v_{K+1}}^{(m)}(x)$ only depends on $h_{v_K}^{(\ell_K)}(x)$, as all layers following $\ell_k$ are stationary, we have that $h_{v_K}^{(m)}(x)$ is constant on $D_k$

$\square$

**Definition C.17.** *A GNN $\mathcal{A}_\theta$ has 1-dimensional aggregation if for all message passing steps $\ell_k \in [L]$*

$$\|W_0^{\text{up},(\ell_k)}\|_0 = 1.$$

One dimensional aggregation implies that the output of $f^{\text{up},(\ell)}$ only depends on one component of the vector produced by min. Furthermore, the value of this one component is determined by which vector in the set

$$\{f^{\text{agg},(\ell_k)}(h_u^{(\ell_k)}, x_{(u,v)}) \mid u \in \mathcal{N}(v)\} \quad (22)$$

is minimal in that component.

**Definition C.18.** *Consider $P_{K+1}^{(1)}(a_1, a_2, \ldots, a_{K+1})$ and a MinAgg GNN $\mathcal{A}_\theta$ with exactly $K$ message passing layers and 1-dimensional aggregation. We say that $\mathcal{A}_\theta$ is path derived on $P_{K+1}^{(1)}(a_1, a_2, \ldots, a_{K+1})$ if for the $k$th message passing step $\ell_k$,*

$$h_{v_{k+1}}^{(\ell_k)} := f^{\text{up},(\ell_k)}\Big( \min\{f^{\text{agg},(\ell_k)}(h_u^{(\ell_k-1)}, x_{(u,v_{k+1})}) \mid u \in \{v_{k+1}\} \cup N(v_{k+1})\} \oplus h_{v_{k+1}}^{(\ell_k-1)}\Big)$$

$$= f^{\text{up},(\ell_k)}\Big( f^{\text{agg},(\ell_k)}(h_{v_k}^{(\ell_k-1)}, x_{(v_k,v_{k+1})}) \oplus h_{v_{k+1}}^{(\ell_k-1)}\Big).$$

**Corollary C.19.** *Consider $P_{K+1}^{(1)}(x, a_2, \ldots, a_{K+1})$ where $a_2, \ldots, a_{K+1}$ are taken to be fixed and view the computed node feature $h_{v_{K+1}}^{(L)} : \mathbb{R} \to \mathbb{R}$ solely as a function of the first edge weight $x$, which*

*is variable. Consider a MinAgg GNN $\mathcal{A}_\theta$ with exactly $K$ message passing layers and 1-dimensional aggregation. If $D \subseteq \mathbb{R}$ contains all $x$ such that $\mathcal{A}_\theta$ is not path derived on $P_{K+1}^{(1)}(x, a_2, \ldots, a_{K+1})$ and*

$$Y_{\mathrm{pd}} = \{h_{v_{K+1}}^{(L)}(x) : x \in D\} \tag{23}$$

*then $|Y_{\mathrm{pd}}| \leq K$.*

*Proof.* For $k \in [K]$, consider the subset $D_k \subseteq \mathbb{R}$ such that $x \in D_k$ if and only if

$$f^{\mathrm{up},(\ell_k)}\left(\min\{f^{\mathrm{agg},(\ell_k)}(h_u^{(\ell_k-1)}, x_{(u,v_{k+1})}) \mid u \in \{v_{k+1}\} \cup N(v_{k+1})\} \oplus h_{v_{k+1}}^{(\ell_k-1)}\right)$$

$$\neq f^{\mathrm{up},(\ell_k)}\left(f^{\mathrm{agg},(\ell_k)}(h_{v_{k+1}}^{(\ell_k-1)}, x_{(v_k,v_{k+1})}) \oplus h_{v_{k+1}}^{(\ell_k-1)}\right).$$

Since $\mathcal{A}_\theta$ has 1-dimensional aggregation, for $x \in D_k$,

$$h_{v_{k+1}}^{(\ell_k)} = f^{\mathrm{up},(\ell_k)}\left(\min\{f^{\mathrm{agg},(\ell_k)}(h_{v_{k+1}}^{(\ell_k-1)}, x_{(v_{k+1},v_{k+1})}), f^{\mathrm{agg},(\ell_k)}(h_{v_{k+2}}^{(\ell_k-1)}, x_{(v_{k+2},v_{k+1})})\} \oplus h_{v_{k+1}}^{(\ell_k-1)}\right)$$

and we have either

$$h_{v_{k+1}}^{(\ell_k)} = f^{\mathrm{up},(\ell_k)}\left(f^{\mathrm{agg},(\ell_k)}(h_{v_{k+1}}^{(\ell_k-1)}, x_{(v_{k+1},v_{k+1})}) \oplus h_{v_{k+1}}^{(\ell_k-1)}\right)$$

or,

$$h_{v_{k+1}}^{(\ell_k)} = f^{\mathrm{up},(\ell_k)}\left(f^{\mathrm{agg},(\ell_k)}(h_{v_{k+2}}^{(\ell_k-1)}, x_{(v_{k+2},v_{k+1})}) \oplus h_{v_{k+1}}^{(\ell_k-1)}\right).$$

since there is only one component of $f^{\mathrm{agg},(\ell_k)}(h_{v_{k+1}}^{(\ell_k-1)}, x_{(v_{k+1},v_{k+1})})$ and $f^{\mathrm{agg},(\ell_k)}(h_{v_{k+2}}^{(\ell_k-1)}, x_{(v_{k+2},v_{k+1})})$ that is not identically zero.

By Claim C.11, $h_{v_{k+1}}^{(\ell_k-1)}(x)$ and $h_{v_{k+2}}^{(\ell_k-1)}(x)$ are constant functions of $x$ ($v_{k+1}$ and $v_{k+2}$ are both more than $k-1$ steps away from $v_1$ and $(v_0, v_1)$, and at step $\ell_k - 1$ only $k-1$ message passing steps have occurred). Then, $h_{v_{k+1}}^{(\ell_k)}(x)$ must be constant over $x \in D_k$ since it only depends on features that are constant. By Lemma C.16, $h_{v_{K+1}}^{(L)}(x)$ is constant on $D_k$ and must take some value $q_k$. Let $Q = \{q_k \mid k \in [K]\}$ and note $|Q| \leq K$.

Suppose $h_{v_{K+1}}^{(L)}(x)$ is not path derived. Then there exists some $k \in [K]$ such that

$$f^{\mathrm{up},(\ell_k)}\left(\min\{f^{\mathrm{agg},(\ell_k)}(h_u^{(\ell_k-1)}, x_{(u,v_{k+1})}) \mid u \in \{v_{k+1}\} \cup N(v_{k+1})\} \oplus h_{v_{k+1}}^{(\ell_k-1)}\right)$$

$$\neq f^{\mathrm{up},(\ell_k)}\left(f^{\mathrm{agg},(\ell_k)}(h_{v_{k+1}}^{(\ell_k-1)}, x_{(v_k,v_{k+1})}) \oplus h_{v_{k+1}}^{(\ell_k-1)}\right).$$

which implies $h_{v_{K+1}}^{(L)}(x) \in Q$ so $Y_{\mathrm{pd}} \subset Q$ and $|Y_{\mathrm{pd}}| \leq K$. $\qquad\square$

## C.5 GLOBAL MINIMUM IS BF

We are now ready to prove Theorem 2.3, which we restate here with additional details.

**Theorem C.20.** *Let $\mathcal{G}_{\mathrm{train}}$ be a set containing pairs of training instances $(G^{(t)}, \Gamma^K(G^{(t)}))$ where $\mathcal{G}_{\mathrm{train}}$ contains $M$ total reachable nodes and $\mathcal{G}_K \subset \mathcal{G}_{\mathrm{train}}$. For $L \geq K > 0$, consider an $L$-layer MinAgg GNN $\mathcal{A}_\theta$ with $m$-layer MLPs and parameters $\theta$. Given regularization coefficient $0 < \eta < \frac{1}{2M(mL+mK+K)}$ and error $0 \leq \epsilon < \eta$, then the loss*

$$\mathcal{L}_{\mathrm{reg}} = \mathcal{L}_{\mathrm{MAE}}(\mathcal{G}_{\mathrm{train}}, \mathcal{A}_\theta) + \eta\|\Theta\|_0 \tag{24}$$

*has a minimum value of $\eta(mL + mK + K)$ and if the loss achieved by $A_\theta$ is within $\epsilon$ of this minimum then on any $G \in \mathcal{G}$ the features computed by the MinAgg GNN satisfy*

$$(1 - M\epsilon)x_v(\Gamma^K(G)) \leq h_v^{(L)}(G) \leq (1 + M\epsilon)x_v(\Gamma^K(G))$$

*for all $v \in V(G)$.*

*Proof.* Our proof proceeds by first simplifying and reparametrizing the MinAgg GNN update using the sparsity structure derived Section C.3. Next, we prove approximations to these new parameters, where we utilize the results of Section C.4 to restrict analysis to cases where computation follows a simple structure (path-derived cases). We conclude by using these approximations to bound the error the MinAgg GNN achieves on an arbitrary graph. Key to this final argument is showing that the features of the MinAgg GNN approximate the node values in the BF algorithm, up to some scaling factor.

**Simplifying the update.** We next show that the MinAgg GNN under the sparsity constraints of Lemma C.15 can be reformulated into a much simpler framework, where all the information exchanged between message-passing layers is consolidated into a much smaller set of parameters and where node features are always one dimensional. As $\{(H_K^{(0)}, H_K^{(K)}), (P_1^{(0)}(1), P_1^{(K)}(1))\} \subset \mathcal{G}_K \subset \mathcal{G}_{\text{train}}$, Lemma C.15 gives that $A_\theta$ has exactly $mL + mK + K$ non-zero parameters, $K$ edge-dependent message passing layers, and a specific sparsity structure: for all layers $\ell \in [L]$, and $j \in [m]$

$$\|W_j^{\text{up},(\ell)}\|_0 = 1$$

and for each of the $K$ message passing layers $\ell_k \in [L]$

$$\|W_1^{\text{agg},(\ell_k)}\|_0 = 2$$

$$\|W_j^{\text{agg},(\ell_k)}\|_0 = 1 \quad \text{for } j > 1$$

where the two non-zero entries in $W_1^{\text{agg},(\ell_k)}$ must share a row. Furthermore, all bias terms are zero. We now argue that each of these non-zero parameters must also be non-negative. For $\ell \in [L]$, suppose that there is some $W_j^{\text{up},(\ell)}$ with a negative element. Then $\sigma(W_j^{\text{up},(\ell)} x) = 0$, where $x$ is a vector with non-negative entries, is a constant function of $x$. This implies $f^{\text{up},(\ell)}$ is constant (it is the zero function), which contradicts Lemma C.15. Similarly, for any message passing layer $\ell_k \in [L]$ if there is some $W_j^{\text{agg},(\ell_k)}$ with a negative element for $j > 1$ then $\sigma(W_j^{\text{agg},(\ell_k)} x) = 0$ is a constant function of $x$ which gives that $f^{\text{agg},(\ell_k)}$ is constant. Again, we have a contradiction with Lemma C.15, which says that $f^{\text{agg},(\ell_k)}$ must be edge and node dependent.

It remains to show that for messages passing layers $\ell_k \in [L]$ that the non-zero elements in $W_1^{\text{agg},(\ell_k)}$ are non-negative. However, we first simplify the update function given the restrictions we have already derived. By the sparsity of the MinAgg GNN ($\|W_m^{\text{up},(\ell)}\|_0 = 1$), at any step $\ell$, all node features $h_v^{(\ell)}$ have at most 1 non-zero component. We can then simplify the update by reducing the number of dimensions. In particular we get

$$\tilde{h}_v^{(\ell)} = \gamma^{(\ell)} \tilde{h}_v^{(\ell-1)}$$

for stationary layers and

$$\tilde{h}_v^{(\ell)} = \gamma^{(\ell)} \min\{\sigma(\rho^{(\ell)} \tilde{h}_v^{(\ell-1)} + \tau^{(\ell)} x_{(u,v)}) : u \in \mathcal{N}(v)\}$$

for message passing layer where $\gamma^{(\ell)}, \rho^{(\ell)}, \tau^{(\ell)} \in \mathbb{R}$ are new parameters that are functions the initial parameters $\theta$, and $\tilde{h}_v^{(\ell)} \in \mathbb{R}$ is a single-dimensional node feature that is equal to the non-zero value in $h_v^{(\ell)}$ (or zero if there is no such value). We have that $\tilde{h}_v^{(0)} = h_v^{(0)} = x_v^{(0)}$ and $\tilde{h}_v^{(L)} = h_v^{(L)}$ since $h_v^{(0)}, h_v^{(L)} \in \mathbb{R}$. Note that the new parameters do not depend on the input features since they are only dependent on $\theta$.

We now further simplify by combining the stationary layers with their succeeding message passing layer. Let $\ell_0 = 0$. For the $k$th message passing layer $\ell_k \in [L]$

$$\tilde{h}_v^{(\ell_k)} = \gamma^{(\ell_k)} \min\left\{\sigma\left(\rho^{(\ell_k)} \left(\prod_{i \in \{\ell_{k-1}+1, \ldots, \ell_k-1\}} \gamma^{(i)}\right) \tilde{h}_v^{(\ell_k-1)} + \tau^{(\ell_k)} x_{(u,v)}\right) : u \in \mathcal{N}(v)\right\}$$

$$= \gamma^{(\ell_k)} \min\left\{\sigma\left(\alpha^{(\ell_k)} \tilde{h}_v^{(\ell-1)} + \tau^{(\ell_k)} x_{(u,v)}\right) : u \in \mathcal{N}(v)\right\}$$

where $\alpha^{(\ell_k)} = \rho^{(\ell_k)} \left(\prod_{i \in \{\ell_{k-1}+1, \ldots, \ell_k-1\}} \gamma^{(i)}\right)$. Note that if for any $\ell_k \in [L]$ we have $h_v^{(\ell_k)} = 0$, then or all succeeding layers $\ell > \ell_k$, the node feature $h_v^{(\ell)}$ is also zero. This is because the min

aggregation at $h_v^{(\ell_k)}$ always includes $h_v^{(\ell_{k-1})}$, so if $h_v^{(\ell_{k-1})} = 0$ then $h_v^{(\ell_k)} = 0$. We are now ready to show $\alpha^{(\ell_k)} > 0$ and $\tau^{(\ell_k)} > 0$. Suppose $\alpha^{(\ell_k)} \leq 0$ for some $\ell_k \in [L]$ and consider the training instance $(P_2^{(1)}(1,0), P_2^{(K+1)}(1,0))$. The update at $v_2$ is

$$\tilde{h}_{v_2}^{(\ell_k)} = \gamma^{(\ell_k)} \min \left\{ \sigma \left( \alpha^{(\ell)} \tilde{h}_u^{(\ell_{k-1})} \right) : u \in \{v_1, v_2\} \right\}$$
$$= 0$$

since $h_v^{(\ell_{k-1})} \geq 0$ for all $v \in V$. We then also have $\tilde{h}_{v_2}^{(L)} = 0$ and since $x_{v_2}(P_2^{(K+1)}(1,0)) = 1$ the loss is

$$\mathcal{L}_{\text{reg}} \geq \mathcal{L}_{\text{MAE}}(\mathcal{G}_{\text{train}}, \mathcal{A}_\theta)$$
$$> 1/M$$
$$> \eta(mL + mL + K) + \epsilon$$

which is a contradiction.

Now instead suppose $\tau^{(\ell_k)} < 0$ for some $\ell_k \in [L]$ and consider the training instance $(P_1^{(0)}(1), P_1^{(K)}(1))$. Since $h_{v_0}^{(\ell)} = 0$ for all $\ell \in [L]$, the update at $v_1$ is

$$\tilde{h}_{v_1}^{(\ell_k)} = \gamma^{(\ell_k)} \min \left\{ \sigma \left( \tau^{(\ell_k)} x_{(v_0,v_1)} \right), \sigma \left( \alpha^{(\ell)} \tilde{h}_{v_1}^{(\ell_{k-1})} + \tau^{(\ell_k)} x_{(v_1,v_1)} \right) \right\}$$
$$= \gamma^{(\ell_k)} \min \left\{ 0, \sigma \left( \alpha^{(\ell)} \tilde{h}_{v_1}^{(\ell_{k-1})} + \tau(\ell_k) x_{(v_1,v_1)} \right) \right\}$$
$$= 0.$$

We then also have $\tilde{h}_{v_1}^{(L)} = 0$ and since $x_{v_1}(P_1^{(K)}(1)) = 1$ the loss is again greater than $\eta(mL + mK + K) + \epsilon$ which is a contradiction.

We can now remove the final ReLU, since its argument is always non-negative:

$$\tilde{h}_v^{(\ell_k)} = \gamma^{(\ell_k)} \min \left\{ \alpha^{(\ell_k)} \tilde{h}_v^{(\ell_{k-1})} + \tau^{(\ell_k)} x_{(u,v)} : u \in \mathcal{N}(v) \right\}.$$

As another simplification, we re-index to $k \in [K]$ as follows. Let $\overline{h}_v^{(k)} = \tilde{h}_v^{(\ell_k)}$ and $\overline{\gamma}^{(\ell_k)} = \gamma^{(\ell_k)}$ for $k \in \{0, \ldots, K-1\}$. However, to account for the effect of the layers succeeding $\ell_k$ we take $\overline{h}_v^{(K)} = \tilde{h}_v^{(L)} =$ and $\overline{\gamma}^{(K)} = \prod_{i \in \{\ell_K, \ell_K+1, \ldots, L\}} \gamma^{(i)}$. Furthermore, for all $k \in [K]$, let $\overline{\alpha}^{(k)} = \alpha^{(\ell_k)}$ and $\overline{\tau}^{(k)} = \tau^{(\ell_k)}$. Then

$$\overline{h}_v^{(k)} = \overline{\gamma}^{(\ell_k)} \min \left\{ \overline{\alpha}^{(k)} \overline{h}_v^{(k-1)} + \overline{\tau}^{(k)} x_{(u,v)} : u \in \mathcal{N}(v) \right\}.$$

Finally, by letting $\mu^{(k)} = \overline{\gamma}^{(k)} / \overline{\alpha}^{(k)}$ and $\nu^{(k)} = \overline{\tau}^{(k)} / \overline{\alpha}^{(k)}$ we get

$$\overline{h}_v^{(k)} = \mu^{(k)} \min \left\{ \overline{h}_v^{(k-1)} + \nu^{(k)} x_{(u,v)} : u \in \mathcal{N}(v) \right\}. \tag{25}$$

Note that we can factor through the $\min$ here because $\alpha^{(k)} > 0$. For the rest of the proof, we focus on this simplified update, since it has the same output as the MinAgg GNN, i.e., $\overline{h}_v^{(K)} = h_v^{(L)}$.

**Approximating parameters.** We proceed by analyzing the output of the MinAgg GNN on an inputs for which it is path derived. To this end, suppose that $\mathcal{A}_\theta$ is path derived on some input graph $P_{K+1}^{(1)}(a_1, \ldots, a_{K+1})$. Then, by definition of path derived,

$$h_{v_{k+1}}^{(\ell_k)} = f^{\text{up},(\ell_k)} \left( f^{\text{agg},(\ell_k)}(h_{v_k}^{(\ell_{k-1})}, x_{(v_k,v_{k+1})}) \oplus h_{v_{k+1}}^{(\ell_{k-1})} \right)$$

which implies

$$\overline{h}_{v_{k+1}}^{(k)} = \mu^{(k)}(\overline{h}_{v_k}^{(k-1)} + \nu^{(k)} x_{(v_k,v_{k+1})}).$$

Next, we prove, for $k \in \{0, \ldots, K\}$,

$$\overline{h}_{v_{k+1}}^{(k)} = \left( \prod_{k \geq i \geq 1} \mu^{(i)} \right) x_{(v_0,v_1)} + \sum_{s=1}^{k} \nu^{(s)} \left( \prod_{k \geq i \geq s} \mu^{(i)} \right) x_{(v_s,v_{s+1})}$$

by induction, where the products evaluate to 1 if they are indexed over an empty set. For the base case we have

$$\overline{h}_{v_1}^{(0)} = x_{(v_0, v_1)}.$$

Now for $k \in [K]$, suppose

$$\overline{h}_{v_k}^{(k-1)} = \left( \prod_{k-1 \geq i \geq 1} \mu^{(i)} \right) x_{(v_0, v_1)} + \sum_{s=1}^{k-1} \nu^{(s)} \left( \prod_{k-1 \geq i \geq s} \mu^{(i)} \right) x_{(v_s, v_{s+1})}.$$

Then we have

$$\overline{h}_{v_{k+1}}^{(k)} = \mu^{(k)} (\overline{h}_{v_k}^{(k-1)} + \nu^{(k)} x_{(v_k, v_{k+1})})$$

$$= \mu^{(k)} \left( \left( \prod_{k-1 \geq i \geq 1} \mu^{(i)} \right) x_{(v_0, v_1)} + \sum_{s=1}^{k-1} \nu^{(s)} \left( \prod_{k-1 \geq i \geq s} \mu^{(i)} \right) x_{(v_s, v_{s+1})} + \nu^{(k)} x_{(v_k, v_{k+1})} \right)$$

$$= \left( \prod_{k \geq i \geq 1} \mu^{(i)} \right) x_{(v_0, v_1)} + \sum_{s=1}^{k-1} \nu^{(s)} \left( \prod_{k \geq i \geq s} \mu^{(i)} \right) x_{(v_s, v_{s+1})} + \nu^{(k)} \mu^{(k)} x_{(v_k, v_{k+1})}$$

$$= \left( \prod_{k \geq i \geq 1} \mu^{(i)} \right) x_{(v_0, v_1)} + \sum_{s=1}^{k} \nu^{(s)} \left( \prod_{k \geq i \geq s} \mu^{(i)} \right) x_{(v_s, v_{s+1})}.$$

This completes the inductions and yields

$$\overline{h}_{v_{K+1}}^{(K)} = \left( \prod_{K \geq i \geq 1} \mu^{(i)} \right) x_{(v_0, v_1)} + \sum_{s=1}^{K} \nu^{(s)} \left( \prod_{K \geq i \geq s} \mu^{(i)} \right) x_{(v_s, v_{s+1})}$$

Given this expression for the output at $v_{K+1}$ we can now derive bounds on the values of these parameters. However, we must restrict our focus to instances of the training set for which $A_\theta$ is path derived.

For $k \in [K]$, let $\mathscr{H}_{k,K}^0$ contain graphs in $\mathscr{H}_{k,K}$ where the $k+1$ edge has weight 0 and let $\mathscr{H}_{k,K}^1$ contain graphs in $\mathscr{H}_{k,K}$ where the $k+1$ edge has weight $2K+1$. It can be checked that for any $G^{(1)}, G^{(1')} \in \mathscr{H}_{k,K}$,

$$|x_{v_{K+1}}(G^{(K+1)}) - x_{v_{K+1}}(G'^{(K+1)})| \geq 1.$$

Then, it must be that $\overline{h}_{v_{K+1}}^{(K)}(G^{(1)}) \neq \overline{h}_{v_{K+1}}^{(K)}(G'^{(1)})$ as if these output features are equal

$$\mathcal{L}_{\text{reg}} \geq \mathcal{L}_{\text{MAE}}$$

$$\geq \frac{|\overline{h}_{v_{K+1}}^{(K)}(G^{(1)}) - x_{v_{K+1}}(G^{(K+1)})| + |\overline{h}_{v_{K+1}}^{(K)}(G'^{(1)}) - x_{v_{K+1}}(G'^{(K+1)})|}{M}$$

$$\geq \frac{1}{M}$$

$$\geq \eta(mL + mK + K) + \epsilon$$

which violates that $|\mathcal{L}_{\text{reg}} - \eta(mL + mK + K)| < \epsilon$.

Consider the subset $\mathscr{J}_{k,K}^0 \subset \mathscr{H}_{k,K}^0$ containing all graphs $G \in \mathcal{H}_{k,K}^0$ for which $A_\theta$ is not path derived on $G$. By Lemma C.19, if

$$Y_{\text{pd}}^0 = \{\overline{h}_{v_{k+1}}^{(k)}(G) : G \in \mathscr{J}_{k,K}^0\}$$

then $|Y_{\text{pd}}^0| \leq K$ and $|\mathscr{J}_{k,K}| \leq K$. Similarly, if $\mathscr{J}_{k,K}^1 \subset H_{k,K}^1$ contains all graphs $G \in \mathscr{H}_{k,K}^1$ for which $A_\theta$ is not path derived on $G$, and

$$Y_{\text{pd}}^1 = \{\overline{h}_{v_{k+1}}^{(k)}(G) : G \in \mathscr{J}_{k,K}^1\}$$

then $|Y_{\text{pd}}^1| \leq K$ and $|\mathscr{J}_{k,K}| \leq K$.

Using these facts, the pigeon hole principle gives that there must be two graphs $G_0^{(1)} \in \mathscr{H}_{k,K}^0 \setminus \mathscr{J}_{k,K}^0$ and $G_1^{(1)} \in \mathscr{H}_{k,K}^1 \setminus \mathscr{J}_{k,K}^1$ such that

$$x_{(v_0,v_1)}(G_0^{(1)}) = x_{(v_0,v_1)}(G_1^{(1)}),$$

i.e., $G_0^{(1)}$ and $G_1^{(1)}$ only differ in the weight of their $(k+1)$th edge.

Since error less than $\epsilon$ is achieved on $\mathcal{G}_{\text{train}}$,

$$|\overline{h}_{v_{K+1}}^{(K)}(G_0^{(1)}) - x_{v_{K+1}}(G_0^{(K+1)})| \leq M\epsilon$$

and

$$|\overline{h}_{v_{K+1}}^{(K)}(G_1^{(1)}) - x_{v_{K+1}}(G_1^{(K+1)})| \leq M\epsilon.$$

The triangle inequity then gives

$$|\overline{h}_{v_{K+1}}^{(K)}(G_0^{(1)}) - x_{v_{K+1}}(G_0^{(K+1)}) - \overline{h}_{v_{K+1}}^{(K)}(G_1^{(1)}) + x_{v_{K+1}}(G_1^{(K+1)})| \leq 2M\epsilon.$$

Making the substitutions

$$\overline{h}_{v_{K+1}}^{(K)}(G_0^{(1)}) = \left(\prod_{K \geq i \geq 1} \mu^{(i)}\right) x_{(v_0,v_1)}(G_0^{(1)}) + \sum_{s=1}^{K} \nu^{(s)} \left(\prod_{K \geq i \geq s} \mu^{(i)}\right) x_{(v_s,v_{s+1})}(G_0^{(1)})$$

$$\overline{h}_{v_{K+1}}^{(K)}(G_1^{(1)}) = \left(\prod_{K \geq i \geq 1} \mu^{(i)}\right) x_{(v_0,v_1)}(G_1^{(1)}) + \sum_{s=1}^{K} \nu^{(s)} \left(\prod_{K \geq i \geq s} \mu^{(i)}\right) x_{(v_s,v_{s+1})}(G_1^{(1)})$$

$$x_{v_{K+1}}(G_0^{(K+1)}) = x_{(v_0,v_1)}(G_0^{(1)}) + x_{(v_k,v_{k+1})}(G_0^{(1)})$$

$$x_{v_{K+1}}(G_1^{(K+1)}) = x_{(v_0,v_1)}(G_1^{(1)}) + x_{(v_k,v_{k+1})}(G_1^{(1)})$$

and canceling like terms yields

$$|\nu^{(k)} \left(\prod_{K \geq i \geq k} \mu^{(i)}\right) (x_{(v_k,v_{k+1})}(G_0^{(1)}) - x_{(v_k,v_{k+1})}(G_1^{(1)})) - (x_{(v_k,v_{k+1})}(G_0^{(1)}) - x_{(v_k,v_{k+1})}(G_1^{(1)}))| \leq 2M\epsilon.$$

Since $|x_{(v_k,v_{k+1})}(G_0^{(1)}) - x_{(v_k,v_{k+1})}(G_1^{(1)})| = 2K+1 \geq 1$, dividing by this factor gives

$$|\nu^{(k)} \left(\prod_{K \geq i \geq k} \mu^{(i)}\right) - 1| \leq 2M\epsilon.$$

We can rewrite the inequality as

$$\frac{1}{\prod_{K \geq i \geq k+1} \mu^{(i)}} (1 - 2M\epsilon) \leq \nu^{(k)}\mu^{(k)} \leq \frac{1}{\prod_{K \geq i \geq k+1} \mu^{(i)}} (1 + 2M\epsilon). \qquad (26)$$

Next we bound the product $\prod_{K \geq i \geq 1} \mu^{(i)}$ by considering how the MinAgg GNN scales the edge weight $x_{(v_0,v_1)}$. Consider two instances $J^{(1)}, J'^{(1)} \in \mathscr{H}_{k,K}^0 \setminus \mathscr{J}_{k,K}^0$, that is, instances for which $A_\theta$ is path derived (which must exist since $|\mathscr{J}_{k,K}^0| \leq K$ and $|\mathscr{H}_{k,K}^0| = 2K+1$). These instances only differ in the weight of their first edge. As before,

$$|\overline{h}_{v_{K+1}}^{(K)}(J^{(1)}) - x_{v_{K+1}}(J^{(K+1)})| \leq M\epsilon$$

and

$$|\overline{h}_{v_{K+1}}^{(K)}(J'(1)) - x_{v_{K+1}}(J'^{(K+1)})| \leq M\epsilon.$$

which combine to give

$$|\overline{h}_{v_{K+1}}^{(K)}(J^{(1)}) - x_{v_{K+1}}(J^{(K+1)}) - \overline{h}_{v_{K+1}}^{(K)}(J'^{(1)}) + x_{v_{K+1}}(J'^{(K+1)})| \leq 2M\epsilon.$$

Substituting in for these four terms and canceling yields

$$\left| \left( \prod_{K \geq i \geq 1} \mu^{(i)} \right) (x_{(v_0,v_1)}(J^{(1)}) - x_{(v_0,v_1)}(J'^{(1)})) - (x_{(v_0,v_1)}(J^{(1)}) - x_{(v_0,v_1)}(J'^{(1)})) \right| \leq 2M\epsilon.$$

Since $|x_{(v_0,v_1)}(J^{(1)}) - x_{(v_0,v_1)}(J'^{(1)})| \geq 1$ we can rearrange this inequality as

$$\left| \left( \prod_{K \geq i \geq 1} \mu^{(i)} \right) - 1 \right| \leq 2M\epsilon.$$

and

$$(1 - 2M\epsilon) \leq \left( \prod_{K \geq i \geq 1} \mu^{(i)} \right) \leq (1 + 2M\epsilon). \tag{27}$$

**Bounding error on arbitrary graphs.** Consider an arbitrary graph $G \in \mathscr{G}$. We aim to show that the MinAgg GNN outputs on this graph $h_v^{(L)} = \overline{h}_v^{(K)}$ approximate $x_v(\Gamma^K(G))$. Let $r_v^{(0)} = x_v(G)$ and define

$$r_v^{(k)} = \min\{r_u^{(k-1)} + x_{(u,v)} \mid u \in \mathcal{N}(v)\}.$$

meaning $r_v^{(k)}$ are the results of applying $k$ steps of the BF algorithm. In particular, $r_v^{(K)}(G) = x_v(\Gamma^K(G))$

We now show that

$$\overline{h}_v^{(k)} \leq \frac{1 + 2M\epsilon}{\prod_{K \geq i \geq k+1} \mu^{(i)}} r_v^{(k)}$$

by induction. The base case

$$\overline{h}_v^{(0)} = r_v^{(0)} \leq \frac{1 + 2M\epsilon}{\prod_{K \geq i \geq 1} \mu^{(i)}} r_v^{(0)}$$

follows from Eq. 27. Now suppose

$$\overline{h}_v^{(k-1)} \leq \frac{1 + 2M\epsilon}{\prod_{K \geq i \geq k} \mu^{(i)}} r_v^{(k-1)}.$$

Then, using Eq. 26,

$$\overline{h}_v^{(k)} = \mu^{(k)} \min\left\{ \overline{h}_v^{(k-1)} + \nu^{(k)} x_{(u,v)} : u \in \mathcal{N}(v) \right\}$$

$$\leq \mu^{(k)} \min\left\{ (1 + 2M\epsilon) r_v^{(k-1)} \frac{1}{\prod_{K \geq i \geq k} \mu^{(i)}} + \nu^{(k)} x_{(u,v)} : u \in \mathcal{N}(v) \right\}$$

$$\leq \min\left\{ (1 + 2M\epsilon) r_v^{(k-1)} \frac{1}{\prod_{K \geq i \geq k+1} \mu^{(i)}} + \frac{1}{\prod_{K \geq i \geq k+1} \mu^{(i)}} (1 + 2M\epsilon) x_{(u,v)} : u \in \mathcal{N}(v) \right\}$$

$$= \frac{1 + 2M\epsilon}{\prod_{K \geq i \geq k+1} \mu^{(i)}} \min\left\{ r_v^{(k-1)} + x_{(u,v)} : u \in \mathcal{N}(v) \right\}$$

$$= \frac{1 + 2M\epsilon}{\prod_{K \geq i \geq k+1} \mu^{(i)}} r_v^{(k)}.$$

On the other hand, we next show

$$\overline{h}_v^{(k)} \geq \frac{1 - 2M\epsilon}{\prod_{K \geq i \geq k+1} \mu^{(i)}} r_v^{(k)}$$

by induction. The base case

$$\overline{h}_v^{(0)} = r_v^{(0)} \geq \frac{1 - 2M\epsilon}{\prod_{K \geq i \geq 1} \mu^{(i)}} r_v^{(0)}$$

again follows from Eq. 27. Now suppose

$$\overline{h}_v^{(k-1)} \geq \frac{1 - 2M\epsilon}{\prod_{K \geq i \geq k} \mu^{(i)}} r_v^{(k-1)}.$$

Then, using Eq. 26,

$$\overline{h}_v^{(k)} = \mu^{(k)} \min \left\{ \overline{h}_v^{(k-1)} + \nu^{(k)} x_{(u,v)} : u \in \mathcal{N}(v) \right\}$$

$$\geq \mu^{(k)} \min \left\{ (1 - 2M\epsilon) r_v^{(k-1)} \frac{1}{\prod_{K \geq i \geq k} \mu^{(i)}} + \nu^{(k)} x_{(u,v)} : u \in \mathcal{N}(v) \right\}$$

$$\geq \min \left\{ (1 - 2M\epsilon) r_v^{(k-1)} \frac{1}{\prod_{K \geq i \geq k+1} \mu^{(i)}} + \frac{1}{\prod_{K \geq i \geq k+1} \mu^{(i)}} (1 - 2M\epsilon) x_{(u,v)} : u \in \mathcal{N}(v) \right\}$$

$$= \frac{1 - 2M\epsilon}{\prod_{K \geq i \geq k+1} \mu^{(i)}} \min \left\{ r_v^{(k-1)} + x_{(u,v)} : u \in \mathcal{N}(v) \right\}$$

$$= \frac{1 - 2M\epsilon}{\prod_{K \geq i \geq k+1} \mu^{(i)}} r_v^{(k)}.$$

Putting the upper bound and lower bound together, taking $k = K$, and using $x_v(\Gamma^K(G)) = r_v^{(K)}$ yields

$$(1 - 2M\epsilon) x_v(\Gamma^K(G)) \leq \overline{h}_v^{(K)} \leq (1 + 2M\epsilon) x_v(\Gamma^K(G))$$

$\square$

# D    ADDITIONAL EXPERIMENTS

We include additional experiments verifying our theoretical claims with different configurations of the MinAgg GNN including the simple BF-GNN described in Theorem 2.2 as well as several configurations of the complex BF-GNN utilized in Theorem 2.3. All models are trained on 8 NVidia A100 GPUs using the AdamW optimizer with a learning rate of 0.001. We evaluate each model using the metrics described in Section 3.

## D.1    SIMPLE MINAGG GNN

We show in Fig. 7 that, with gradient descent, a simple MinAgg GNN will converge to the parameter configuration described in Theorem 2.2. The simple MinAgg GNN is trained with the specified train set in Theorem 2.2: four single-edge graphs initialized at step 0 and two double-edge graphs initialized at step 1. Recall that an update for the simple MinAgg GNN is defined as

$$h_v^{(1)} = \sigma(w_2 \min_{u \in \mathcal{N}(v)} \{\sigma(W_1(x_u \oplus x_{(v,u)} + b_1))\} + b_2)$$

where $w_2, b_1, b_2 \in \mathbb{R}$ and $W_1 \in \mathbb{R}^{1 \times 2}$. From Theorem 2.2, we know that the simple BF-GNN will implement a single step of Bellman-Ford if $w_2 W_{11} = w_2 w_{12} = 1$ and $w_2 b_1 + b_2 = 0$. Therefore, in Fig. 7 (a), we see that, via gradient descent, the parameter configurations converge to the expected values (indicated by the black dotted lines). We further empirically verify the results in Theorem 2.2 in Fig. 7(b) by showing that the as the parameter configurations converge to the expected values, the test error also converges to zero.

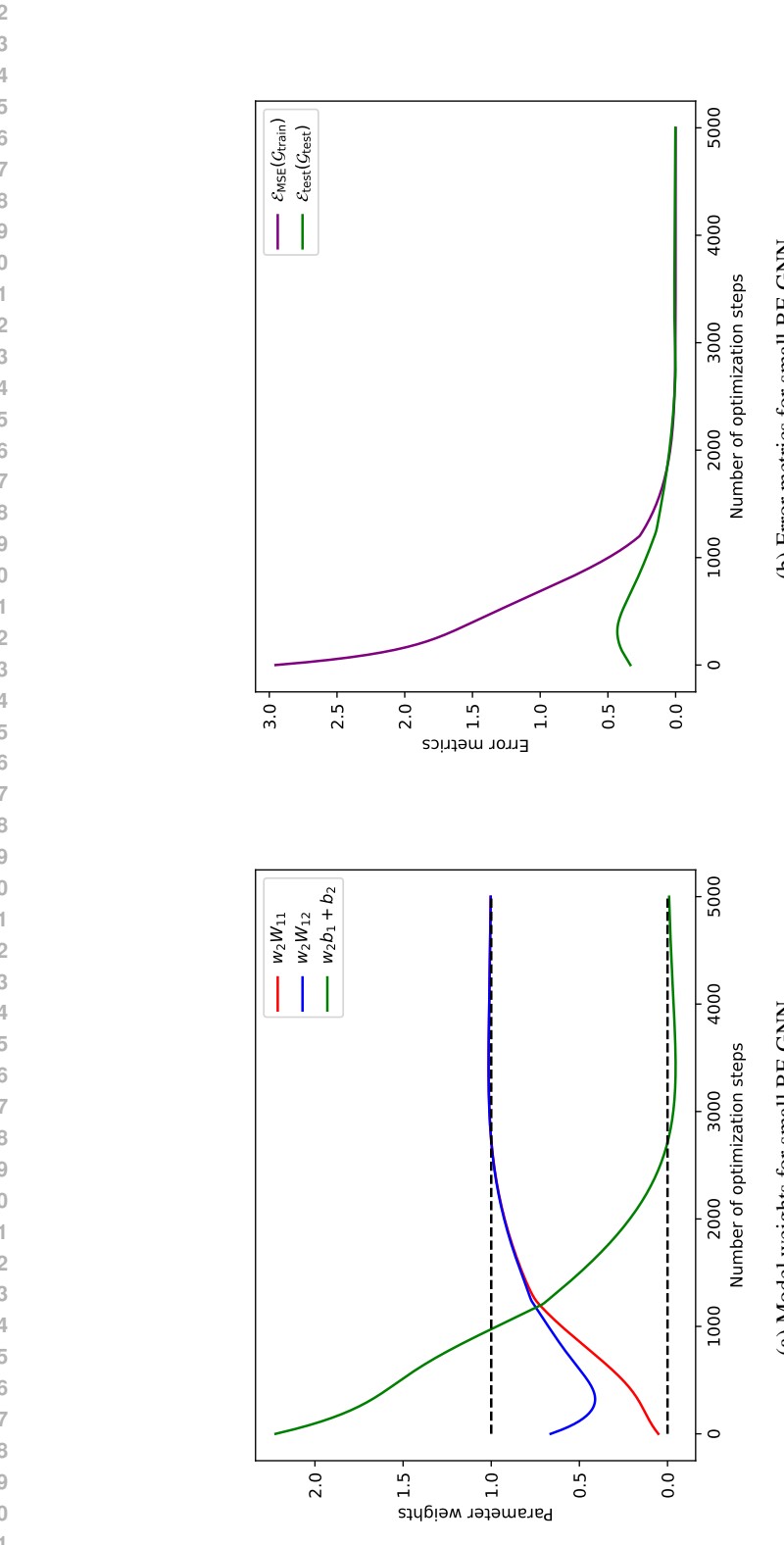

(a) Model weights for small BF-GNN.

(b) Error metrics for small BF-GNN.

Figure 7: Performance and parameter weights for the small BF-GNN instance (trained with MSE). Recall from to Theorem 2.2, we expect that $w_2 b_1 + b_2 = 0$ and $w_2 W_{11} = w_2 W_{12} = 1.0$. We indicate these values by the black dotted lines in (a) and show that the parameters of our small BF-GNN converge to the expected parameter values from Theorem 2.2. Additionally, we verify that convergence to this parameter configuration corresponds to low test error in (b) as we have that $\mathcal{E}_{t,\text{test}}$ converges to 0.0018.

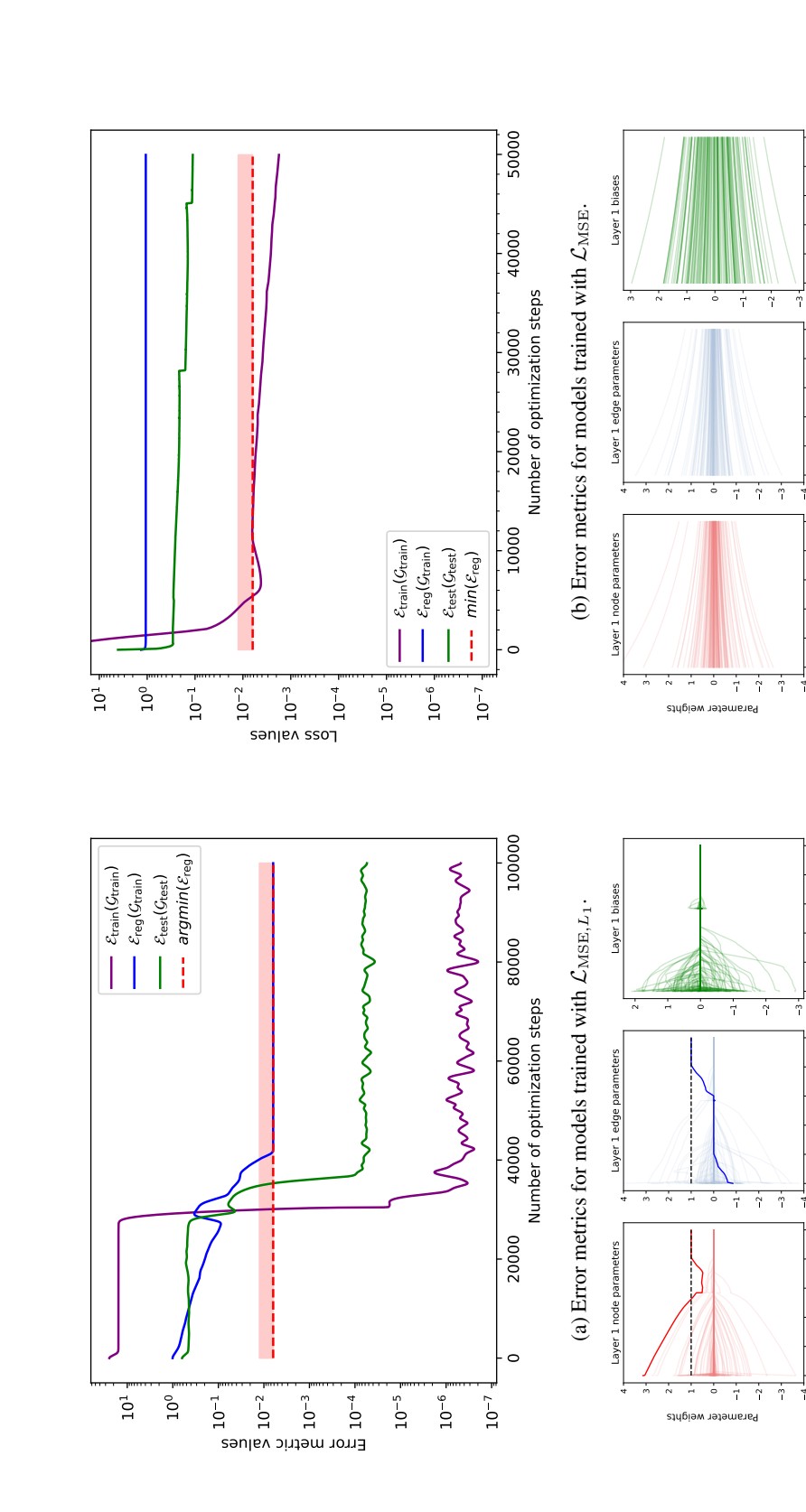

(a) Error metrics for models trained with $\mathcal{L}_{\mathrm{MSE},L_1}$.

(b) Error metrics for models trained with $\mathcal{L}_{\mathrm{MSE}}$.

(c) Model parameters for models trained with $\mathcal{L}_{\mathrm{MSE},L_1}$

(d) Model parameters for models trained with $\mathcal{L}_{\mathrm{MSE}}$

Figure 8: Error metrics and parameter updates for a one-layer MinAgg GNN trained on a single step of the Bellman-Ford algorithm. Note that the dotted line in (a) and (b) is the global minimum of $\mathcal{E}_{\mathrm{reg}}$ and the red highlighted region indicates the error bound described in Theorem 2.3. (a) and (b) show how each error metric changes over each training epoch for the models trained with $\mathcal{L}_{\mathrm{MSE},L_1}$ and $\mathcal{L}_{\mathrm{MSE}}$. Note that $\mathcal{E}_{\mathrm{test}}$ is 0.00008 for the $L_1$ regularized model and $\mathcal{E}_{\mathrm{test}}$ is 0.212 for the un-regularized model. Additionally, (b) and (c) show how the model parameters update from epoch for models trained with $\mathcal{L}_{\mathrm{MSE},L_1}$ and $\mathcal{L}_{\mathrm{MSE}}$, respectively.

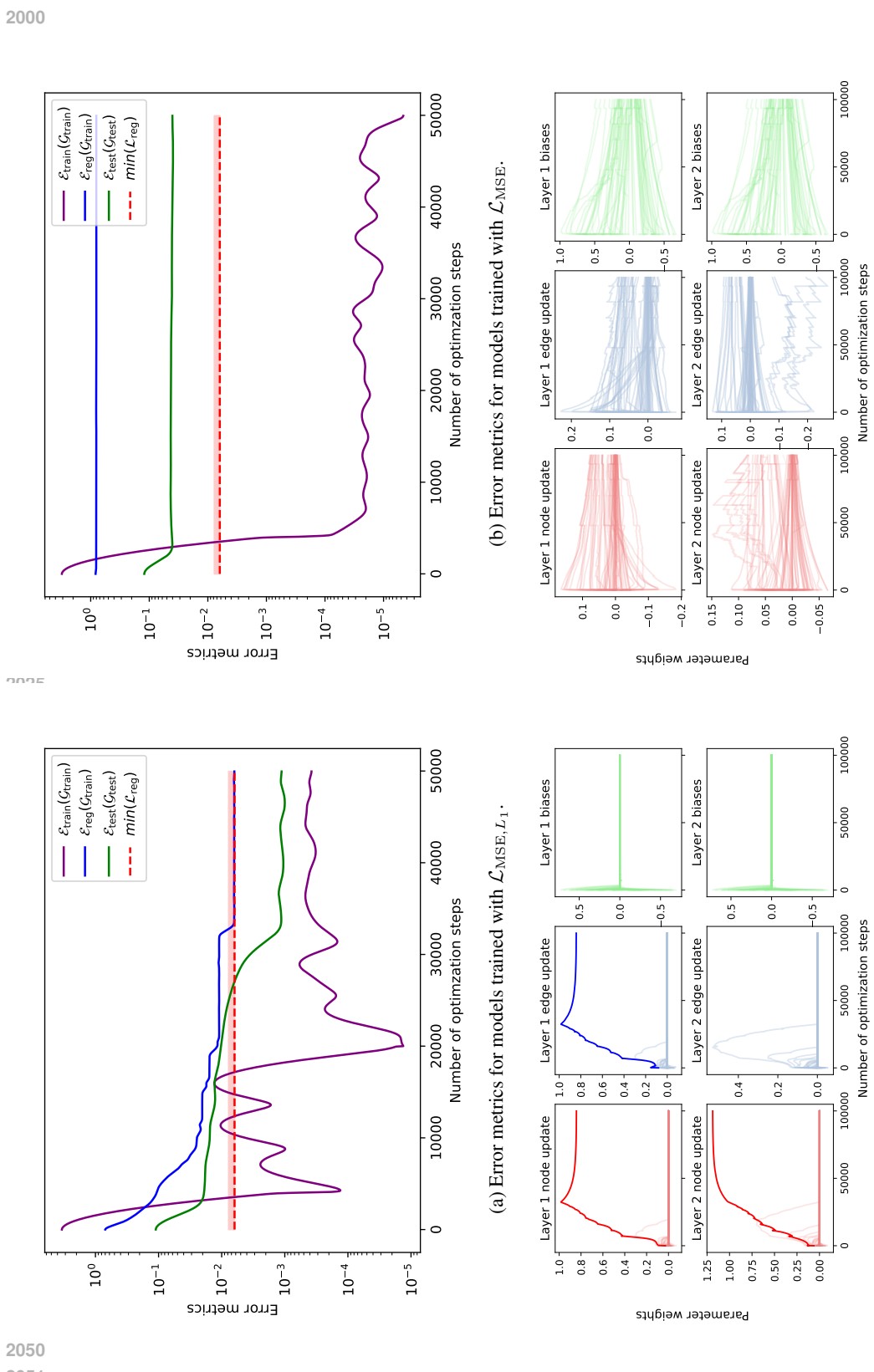

Figure 9: Error metrics and parameter updates for a two-layer MinAgg GNN trained on a single step of the Bellman-Ford algorithm. Note that the dotted line in (a) and (b) is the global minimum of Eq. (8). (a) and (b) show how the train loss and test loss change over each training epoch for the models trained with $\mathcal{L}_{\mathrm{MSE},L_1}$ and $\mathcal{L}_{\mathrm{MSE}}$ as well as the theoretical loss term $\mathcal{L}_{\mathrm{reg}}$ over time. $\mathcal{E}_{\mathrm{test}}$ converges to 0.001 for the $L_1$ regularized model and 0.0312 for the un-regularized model. (b) and (c) show how the model parameters update from epoch to epoch for models trained with $\mathcal{L}_{\mathrm{MSE},L_1}$ and $\mathcal{L}_{\mathrm{MSE}}$, respectively. Each curve has been smoothed with a truncated Gaussian filter with $\sigma = 20$.

## D.2 DEEP MINAGG GNNS

We examine a variety of MinAgg GNN configurations to empirically verify Theorem 2.3. As in Section 3, we compare models trained with $L_1$ regularization against models trained using just $\mathcal{L}_{\text{MSE}}$. Similar to Section 3, for each model, we track $\mathcal{E}_{\text{test}}$, $\mathcal{E}_{\text{train}}$, and $\text{E}_{\text{reg}}$ throughout optimization. Note that $\mathcal{E}_{\text{test}}$ is evaluated on the same set of test graphs described in Section 3 i.e. $\mathcal{G}_{\text{test}}$ is a set of 200 graphs which are a mix of cycles, complete graphs, and Erdös-Renyí graphs with $p = 0.5$. Note that Theorem 2.3 requires $\mathcal{E}_{\text{reg}}$ to fall below a certain $\epsilon$ threshold (indicated by the red region). For each of the complex GNN configurations we analyze below, we observe that models trained with $L_1$ regularization satisfy this bound, aligning with Theorem 2.3. Furthermore, to verify that the model learns the correct parameters which implement Bellman-Ford, we also track a summary of the model parameters per epoch, defined as follows. This is the same as the model parameters from Section 3 in the main text, but here we provide more detail. Recall the definition of MinAgg GNNs in Def. 2.1. Each layer can be precisely expressed as follows:

$$\sigma\Big(W^{\text{up},(\ell)}(\min\{\sigma(W^{\text{agg},(\ell)}(h_u^{(\ell-1)} \oplus x_{(u,v)})) + b^{\text{agg},(\ell)}\} \oplus h_v^{(\ell-1)})\Big) + b^{\text{up},(\ell)}$$

To analyze parameter dynamics, we visualize the following for each optimization step and each layer:

- Layer $\ell$ node parameters: Given a node feature in $\mathbb{R}^d$, the first $d$ columns of $W^{\text{agg},(\ell)}$, $W^{\text{agg},(\ell)}[:, : d]$ scale the incoming neighboring node features. The contribution of incoming neighboring node features to the layer-wise output node feature for $v$ can be summarized as follows:

$$\bigoplus_{j=1}^{d}\Big(\bigoplus_{i=1}^{d_{\text{up}}} W^{\text{up},(\ell)}[i, : d_{\text{agg}}] \odot W^{\text{agg},(\ell)}[:, j]\Big) \oplus \Big(\bigoplus_{d_{\text{up}}}^{d_{\text{agg}}+d_{\text{up}}} W^{\text{up},(\ell)}[i, d_{\text{agg}} : d_{\text{agg}} + d_{\text{up}}]\Big)$$

where $\odot$ is the element-wise product and $\oplus$ denotes concatenation.

- Layer $\ell$ edge parameters: Similar to above, we know that the last column, $d + 1$, of $W^{\text{agg},(\ell)}$ scales the incoming edge features. Therefore, the contribution of neighoring edges to the layer-wise output node feature can be summarized as

$$\bigoplus_{i=1}^{d_{\text{up}}} W^{\text{up},(\ell)}[i : d_{\text{agg}}] \odot W^{\text{agg},(\ell)}[:, d + 1].$$

- Layer $\ell$ biases: We track the bias terms for each layer as $b^{\text{agg},(\ell)} \oplus b^{\text{up}}$. Note that for the sparse implementation of Bellman-Ford that we describe previously, we require that the bias terms all converge to zero.

Finally, note that we summarize each model's size generalization ability in Table 2 (analogous to Table 1 in the main text) in both the setting where we make a single forward pass through the model and when we use the model as a module and make repeated forward passes through the model. In Table 1, we see that each of the models trained with $L_1$ regularization achieve significantly lower test error than those trained without $L_1$ regularization.

### D.2.1 ONE LAYER

We configure the single layer GNN with BF-GNN with $d_{\text{agg}} = 64$ and $d_{\text{up}} = 1$. We evaluate $\mathcal{L}_{\text{MSE}}$ on $\mathcal{G}_{\text{train}}$ which is the same as for the simple BF-GNN (four two-node path graphs and four three-node path graphs starting from step one of Bellman-Ford). Intuitively, the two-node path graphs provide a signal which controls the edge update feature while the three-node path controls the node update feature of Bellman-Ford. The results for a single layer BF-GNN are summarized in Fig. 8.

First, as illustrated in Fig. 8 (a) and (b), the train error $\mathcal{L}\text{MSE}$ alone does not capture the model's generalization ability. Both the $L_1$-regularized and non-regularized models achieve low $\mathcal{L}\text{MSE}$ (0.002 for both the $L_1$ regularized model and the un-regularized model), yet only the regularized model—where $\mathcal{L}_{\text{reg}}$ converges to its minimum value—exhibits low test error.

Furthermore, we verify in Fig. 8 (b) and (c) that the parameters for the single layer MinAgg GNN converge to a configuration which *approximately implements* a single step of Bellman-Ford. Since we

Table 2: Error ($\mathcal{E}_{\text{test}}$) versus size for all model configurations. The first row of the table contains the test error for both the single (indicated by 1L) and two-layer (indicated by 2L) model configurations trained on a single step of Bellman-Ford and the second row contains the test error for all model configurations trained on two steps of Bellman-Ford. We use 'reg' to indicate that the model is trained with $L_1$ regularization and 'un-reg' to indicate a model trained without $L_1$ regularization. Similar to Table 1, we examine the error for a single pass of each model (one step of BF for the first row and two steps of BF for the second row) and for three forward passes of each model (three steps of BF for the first row and six steps of BF for the second row). Each test set consists of Erdös–Rényi graphs generated with the corresponding sizes listed with $p$ such that the expected degree $np = 5$. For both models trained with $L_1$ regularization and without, the error does not change much as the number of nodes in the test graphs increase. However, when each model is used as a module and iterated, we see that the $L_1$ regularized model remains accurate while the error for the un-regularized model increases significantly.

| | # of nodes | Single | | | |
| | | 1L, un-reg. | 1L, reg. | 2L, un-reg. | 2L, reg. |
|---|---|---|---|---|---|
| One step | 100 | $0.0079 \pm 0.0027$ | $0.00006 \pm 0.00003$ | $0.0569 \pm 0.0064$ | $0.0022 \pm 0.0002$ |
| | 500 | $0.0070 \pm 0.0025$ | $0.00006 \pm 0.00004$ | $0.0560 \pm 0.0072$ | $0.0022 \pm 0.0003$ |
| | 1K | $0.0071 \pm 0.0026$ | $0.00006 \pm 0.00004$ | $0.0558 \pm 0.0073$ | $0.0021 \pm 0.0003$ |
| Two steps | 100 | - | - | $0.0296 \pm 0.002$ | $0.0173 \pm 0.001$ |
| | 500 | - | - | $0.0297 \pm 0.002$ | $0.0174 \pm 0.001$ |
| | 1K | - | - | $0.0308 \pm 0.002$ | $0.0180 \pm 0.001$ |
| | # of nodes | Iterated | | | |
| | | 1L, un-reg. | 1L, reg. | 2L, un-reg. | 2L, reg. |
| One step | 100 | $0.0320 \pm 0.0028$ | $0.00012 \pm 0.00004$ | $0.0097 \pm 0.0017$ | $0.00133 \pm 0.0001$ |
| | 500 | $0.0289 \pm 0.0025$ | $0.00012 \pm 0.00006$ | $0.0074 \pm 0.0010$ | $0.00147 \pm 0.0001$ |
| | 1K | $0.0290 \pm 0.0025$ | $0.00011 \pm 0.00001$ | $0.0072 \pm 0.0013$ | $0.00151 \pm 0.0001$ |
| Two steps | 100 | – | – | $0.0596 \pm 0.0131$ | $0.0182 \pm 0.0009$ |
| | 500 | – | – | $0.0391 \pm 0.0040$ | $0.0197 \pm 0.0006$ |
| | 1K | – | – | $0.0367 \pm 0.0025$ | $0.0199 \pm 0.0007$ |

are only considering a single layer MinAgg GNN and the input node feature are the initial distances to source ($d = 1$), the node parameter summary that we consider is

$$W^{\text{up},(1)}[: d_{\text{agg}}] \odot W^{\text{agg},(1)}[:, 1] \oplus W^{\text{up},(1)}[d_{\text{agg}} + d]$$

where $W^{\text{agg},(1)} \in \mathbb{R}^{d_{\text{agg}} \times 2}$ and $W^{\text{up},(1)} \in \mathbb{R}^{d_{\text{agg}} + d}$. Additionally, the edge parameter summary we consider is

$$W^{\text{up},(1)}[: d_{\text{agg}}] \odot W^{\text{agg},(1)}[:, 2].$$

Therefore, for a single layer MinAgg GNN, the model parameters which exactly implement Bellman-Ford has a single $k \in [64]$ such that

$$W^{\text{up},(1)}[k] \cdot W^{\text{agg},(1)}[k, 1] = W^{\text{up},(1)}[k] \cdot W^{\text{agg},(1)}[k, 2] = 1.0.$$

This means there is a single identical positive non-zero value for both the node and edge parameters. Additionally, all biases are zero. In Fig. 8 (c), we see that the trained MinAgg GNN using $L_1$ regularization approximately converges to this configuration of parameters. However, the MinAgg GNN trained without regularization does not achieve this parameter configuration, explaining the higher test error $\mathcal{E}_{\text{test}}$.

### D.2.2 TWO LAYER, SINGLE STEP

We configure a two layer GNN with $d_{\text{agg}} = 64$ and $d_{\text{up}} = 1$ for all layers and train it on a single step of Bellman-Ford. Note that this setup is overparameterized for modeling a single step of Bellman-Ford. The results of training on a single step of Bellman-Ford are summarized in Fig. 9. As with the single layer MinAggGNN configuration, the train set again consists of four two-node path graphs and four three-node path graphs with varying edge weights.

First, while $\mathcal{L}_{\text{MSE}}$ converged to a low error for both models, only the model trained with $\mathcal{L}_{\text{MSE},L_1}$ has $\mathcal{L}_{\text{reg}}$ converging to a low value. This also corresponds to a significantly lower $\mathcal{L}_{\text{test}}$, again

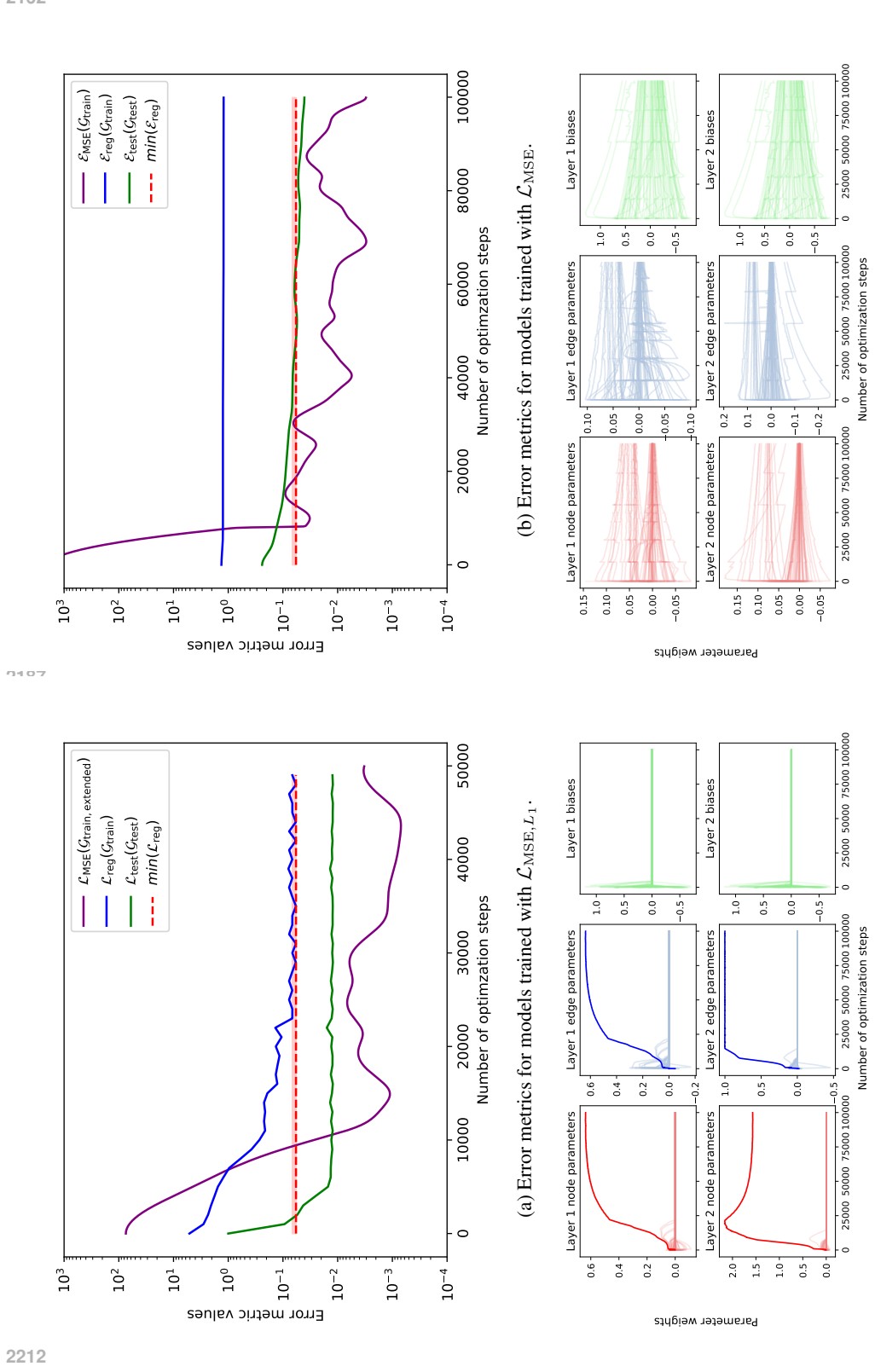

(a) Error metrics for models trained with $\mathcal{L}_{\mathrm{MSE},L_1}$.

(b) Error metrics for models trained with $\mathcal{L}_{\mathrm{MSE}}$.

(c) Model parameters summaries for model trained with $\mathcal{L}_{\mathrm{MSE},L_1}$.

(d) Model parameters summaries for model trained with $\mathcal{L}_{\mathrm{MSE}}$.

Figure 10: Performance metrics and parameter updates for a two-layer MinAgg GNN trained on two steps of the Bellman-Ford algorithm. The dotted line in (a) and (b) is the global minimum of Eq. (8) and the red region represents the $\epsilon$ bounds in Theorem 2.3. In (a) and (b), we track the change in the train loss, test loss, and $\mathcal{L}_{\mathrm{reg}}$ over each optimization step for the models trained with $\mathcal{L}_{\mathrm{MSE},L_1}$ and $\mathcal{L}_{\mathrm{MSE}}$. Note that the final test loss for the model trained with $\mathcal{L}_{\mathrm{MSE},L_1}$ is 0.006 while the final test loss for the model trained with $\mathcal{L}_{\mathrm{MSE}}$ is 0.288. (b) and (c) show how the model parameters change over each optimization step models trained with $\mathcal{L}_{\mathrm{MSE},L_1}$ and $\mathcal{L}_{\mathrm{MSE}}$, respectively. Each curve has been smoothed with a truncated Gaussian filter with $\sigma = 20$.

demonstrating an overparameterized model's ability to learn Bellman-Ford under an $L_1$-regularized training error and generalize to larger graph sizes with lower test error. These results align with our theoretical results, where the convergence of $\mathcal{L}_{\mathrm{reg}}$ to its minimum value provides a certificate for size generalization.

In Fig. 9 (b) and (c), we again further verify the size generalization ability of our models and emphasize the importance of sparsity regularization for size generalization by showing that the model parameter summaries for the model trained with $\mathcal{L}_{\mathrm{MSE},L_1}$ converge to an implementation of a single step of Bellman-Ford. Since $d_{\mathrm{up},(\ell)} = 1$ for both layers, the node parameter summary that we analyze is again

$$W^{\mathrm{up},(\ell)}[: d_{\mathrm{agg}}] \odot W^{\mathrm{agg},(\ell)}[:, 1] \oplus W^{\mathrm{up},(\ell)}[d_{\mathrm{agg}} + 1]$$

and the edge parameter summary is

$$W^{\mathrm{up},(\ell)}[: d_{\mathrm{agg}}] \odot W^{\mathrm{agg},(\ell)}[:, 2].$$

for both layers. Similar to the single layer and single edge case, for the first layer, we expect that in the sparse implementation of a single step of Bellman, there will only a unique identical and positive non-zero value for both the node and edge parameter summaries. Therefore, for this sparse implementation, for any node $v$ in a given input graph, the node feature for $v$ at the first layer is $a(x_{u'} + x_{(u',v)})$ where $a > 0$ and $u' = \mathrm{argmin}_{u \in \mathcal{N}(v)}\{x_u + x_{(u,v)}\}$. In the second layer, $W^{\mathrm{up},(2)}[65] = 1/a$ is the only positive non-zero parameter. Therefore, the final output for $v$ will be $\min_{u \in \mathcal{N}(v)}\{x_u + x_{(u,v)}\}$. In Fig. 9 (b) and (c), we see again that the model trained with $L_1$ regularization (i.e. $\mathcal{L}_{\mathrm{MSE},L_1}$) has its parameters approximately converge to this sparse implementation of Bellman-Ford. In contrast, the model trained without $L_1$ regularization (i.e. only $\mathcal{L}_{\mathrm{MSE}}$) does not appear to converge such a sparse implementation of Bellman-Ford, which accounts for the higher test error of the model.

### D.2.3 TWO LAYER, TWO STEPS

In the main text, we evaluate the ability of a two layer MinAgg GNN to learn two steps of Bellman-Ford. Here, we show the ability of the MinAgg GNN to learn two steps of Bellman-Ford in a somewhat under-parameterized setting as we let $d_{\mathrm{agg}} = 64$ and $d_{\mathrm{up}} = 1$. The results are summarized in Fig. 10. Similar to the other model configurations evaluated (both in the supplement and the main text), we see that in Fig. 10 (a) and (b) that the $L_1$ regularized model achieves much lower $\mathcal{E}_{\mathrm{reg}}$ and correspondingly, much lower $\mathcal{E}_{\mathrm{test}}$. The parameter configurations visualized in Fig. 10 (c) and (d) show that the $L_1$ regularized model with low $\mathcal{E}_{\mathrm{reg}}$ converges to the sparse implementation of two-steps Bellman-Ford, as we have shown theoretically in Theorem 2.3. Additionally, note in Table 2, the gap between the error for the unregularized model and the error for the regularized model is much lower than that of Table 1 in the main text.

### D.3 ADDITIONAL SYNTHETIC AND REAL DATASETS

To provide further evidence of our claims, we provide a comparison of both the unregularized and $L_1$-regularized BF GNN error on several synthetic and real datasets. For the synthetic datasets we use:

- Stochastic block models (SBM): these graphs are generated using 3-7 partitions of 35 vertices each and a randomly generated probability matrix.

- Barabasi-Albert models (BA): these graphs are generated according to the Barabasi-Albert model with preferential attachment parameter $m \in [3, 65]$.

- Random geometric graphs: we randomly position 5-65 nodes in $[0, 1]^2$. Two nodes are joined by an edge if the distance between them is less than 0.1.

For the real datasets, we use a terrain triangulation of patch of land in Norway with 40,000 vertices. Edge weights are determined by the Euclidean distance between points on the terrain. Additionally, we use Airports USA (USAir97) (25; 24), where nodes represent airports and edges represent the existence of commercial flights between them. All edge weights for this graph are randomly generated. In Table 3, we evaluate the performance of the two-step BF GNN model. For the sake of comparison, we also include a comparison to the GAT (both trained with mean squared error and also trained with $L_1$ regularization.

Additionally, we further evaluate our model as an iterative module for solving more than $k$ steps of Bellman–Ford on several synthetic test graphs. We show this for both learning a single step

|  | Synthetic | | | Norway | Airports-USA |
|---|---|---|---|---|---|
|  | SBM | BA | Geometric | | |
| MSE | $0.053 \pm 0.006$ | $0.441 \pm 0.431$ | $0.226 \pm 0.334$ | $0.0141 \pm 0.0652$ | $0.1119 \pm 0.0924$ |
| $L_1$-reg | $0.003 \pm 0.0005$ | $0.005 \pm 0.0011$ | $0.003 \pm 0.0022$ | $0.0003 \pm 0.0008$ | $0.0081 \pm 0.0299$ |
| GAT (MSE) | $0.977 \pm 0.015$ | $0.838 \pm 0.157$ | $0.971 \pm 0.047$ | $1.611 \pm 3.027$ | $5.426 \pm 7.044$ |
| GAT ($L_1$) | $0.722 \pm 0.121$ | $1.237 \pm 0.2137$ | $0.616 \pm 0.297$ | $0.9178 \pm 0.1416$ | $0.998 \pm 0.0081$ |

Table 3: Comparison of test error across all synthetic and real datasets.

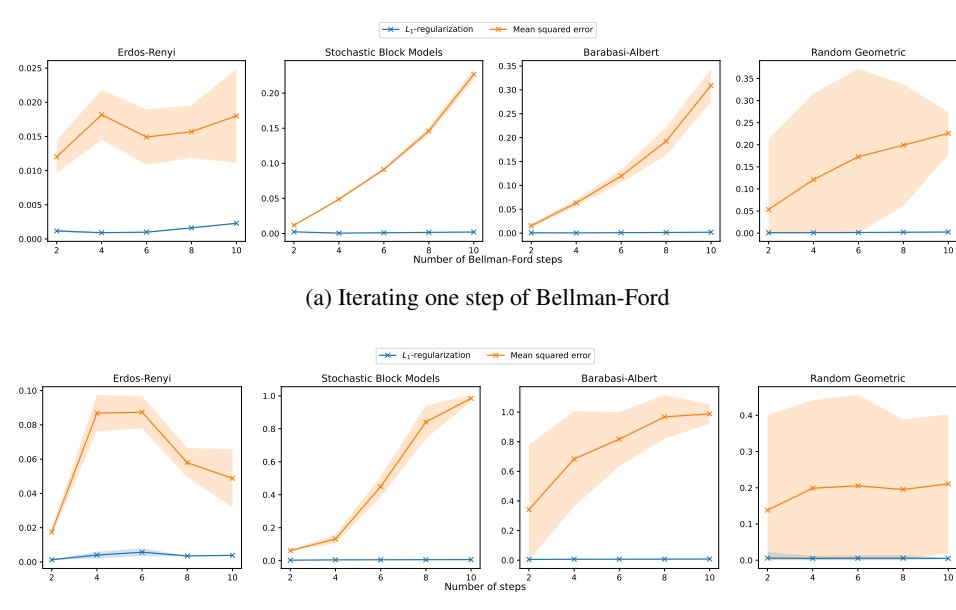

(a) Iterating one step of Bellman-Ford

(b) Iterating two steps of Bellman-Ford.

Figure 11: Extended results for iterative modules.

of Bellman–Ford and learning two steps of Bellman–Ford; see Fig. 11. Across iterations, the $L_1$-regularized models exhibit substantially slower error accumulation (flatter curves) than the unregularized $\mathcal{L}_{\text{MSE}}$ models.

### D.4 HIDDEN EDGE WEIGHTS

We next evaluate a setting where the algorithmic skeleton (shortest paths via Bellman–Ford) is appropriate, but a component of the computation is unknown and must be learned. Concretely, edges expose an observed attribute $x_e$. The effective distance used by the shortest-path objective is an unknown, nonnegative transformation $g(x_e)$. The model must both learn $g$ and learn to execute BF, the latter of which is aided by algorithmic alignment. As a simple instantiation, we set $g(x) = (x-2)^2$ and supervise on shortest-path labels computed with these transformed edge costs (see Table 4).

This scenario mirrors real applications—for example, routing to minimize fuel or energy consumption where per-road cost is a nonlinear function of observable features (e.g., grade, speed limit, and congestion) rather than raw length alone. In this regime, the $L_1$-regularized, BF-aligned MinAgg GNN reliably learns a sparse approximation to $g$ and extrapolates to larger and structurally different graphs, substantially outperforming the unregularized $\mathcal{L}_{\text{MSE}}$ model. This provides a concrete example of how algorithmic alignment remains effective when we have only a prior over the desired computation (e.g., shortest-path computation) and must infer the remaining components from data.

### D.5 INFERENCE TIME

We benchmarked the inference time of a trained 2-step Bellman-Ford neural network model against NetworkX's optimized Bellman-Ford implementation on Erdos-Renyi random graphs with edge probability $p = 0.5$. The experiment compared NetworkX's

|  | ER | SBM | BA | Geometric |
|---|---|---|---|---|
| MSE | $0.383 \pm 0.055$ | $0.541 \pm 0.093$ | $0.276 \pm 0.129$ | $0.517 \pm 0.732$ |
| $L_1$-reg | $0.0054 \pm 0.0006$ | $0.0055 \pm 0.0006$ | $0.0057 \pm 0.0013$ | $0.0291 \pm 0.0334$ |

Table 4: Hidden edge function $f(x) = (x-2)^2$

`single_source_bellman_ford_path_length` function (CPU) against our trained SingleSkipBFModel running on both CPU and GPU (NVIDIA RTX A6000). We tested 100 graphs each for $n \in \{100, 500, 1000\}$ nodes, with all graphs verified to have diameter $\leq 2$ to ensure that 2 steps of Bellman-Ford compute full shortest paths. Results (Table 5) show that the CPU model is fastest for small graphs ($n = 100$, 1.16ms vs 2.56ms for NetworkX), while the GPU model achieves significant speedups for larger graphs ($n = 1000$, 5.73ms vs 350.78ms for NetworkX, a $61\times$ speedup). The improvements in our model over NetworkX are due to PyTorch's optimized operations and GPU acceleration.

| $n$ | NetworkX (CPU) | Model (CPU) | Model (GPU) |
|---|---|---|---|
| 100 | $2.56 \pm 0.39$ | $1.16 \pm 0.45$ | $2.13 \pm 14.75$ |
| 500 | $66.26 \pm 4.14$ | $26.41 \pm 8.89$ | $2.18 \pm 0.51$ |
| 1000 | $350.78 \pm 530.20$ | $239.74 \pm 3.73$ | $5.73 \pm 0.24$ |

Table 5: Mean inference time (ms) with standard deviation for Bellman-Ford computation on Erdos-Renyi graphs.

## E    LIMITATIONS

We now discuss limitations to our current study. First, we do not provide experiments for graph neural networks (GNNs) with many layers, and thus our experimental findings may not generalize to deeper architectures. Second, our approach requires training sets to be explicitly constructed rather than sampled from a distribution, which may limit the applicability of our results to more general or practical scenarios. Finally, we focus exclusively on the properties of the global minimum of the loss and do not discuss optimization dynamics or the process of reaching this minimum, which are important considerations in real-world training.

## F  TABLE OF NOTATION

| Symbol | Definition |
|---|---|
| **General Notations** ||
| $[n]$ | The set $\{1, 2, \ldots, n\}$. |
| $x \oplus y$ | Concatenation of vectors $x$ and $y$. |
| $x_i$ or $[x]_i$ | $i$-th component of vector $x$. |
| $\beta$ | A large constant representing the unreachable node feature value. |
| **Graphs and Attributed Graphs** ||
| $G = (V, E, X_{\mathrm{v}}, X_{\mathrm{e}})$ | Attributed graph with vertices $V$, edges $E$, edge weights $X_{\mathrm{e}}$, and node attributes $X_{\mathrm{v}}$. |
| $X_{\mathrm{e}}, X_{\mathrm{v}}$ | Edge weights $\{x_e : e \in E\}$ and node attributes $\{x_v : v \in V\}$. |
| $\mathrm{d}^{(t)}(s, v)$ | Length of the $t$-step shortest path from node $s$ to $v$; $\beta$ if no such path exists. |
| $G^{(t)}$ | $t$-step Bellman-Ford (BF) instance with node features $\{x_v = \mathrm{d}^{(t)}(s, v) : v \in V\}$. |
| $\Gamma$ | Operator implementing a single step of the BF algorithm. |
| $\mathscr{G}$ | Set of all edge-weight-bounded attributed graphs. |
| $P_k^{(\ell)}(a_1, \ldots, a_k)$ | $k$-edge path graph at step $\ell$ with edge weights $a_1, \ldots, a_k$. |
| $\mathcal{N}(v)$ | Neighborhood of node $v$ in the graph. |
| $V^*(G)$ | Set of reachable nodes in graph $G$, i.e., nodes with $x_v \neq \beta$. |
| **Graph Neural Networks (GNNs)** ||
| $\mathcal{A}_\theta$ | $L$-layer Bellman-Ford Graph Neural Network (MinAgg GNN) parameterized by $\theta$. |
| $h_v^{(\ell)}$ | Hidden feature of node $v$ at layer $\ell$ in the MinAgg GNN. |
| $\mathcal{A}_\theta(G)$ | Output graph of MinAgg GNN $\mathcal{A}_\theta$ after $L$ layers, with updated node features. |
| $d_\ell$ | Dimensionality of hidden features at layer $\ell$. |
| $f^{\mathrm{agg}}, f^{\mathrm{up}}$ | MLPs used for aggregation and update operations in MinAgg GNN layers. |
| $W_j^{\mathrm{agg}}, W_j^{\mathrm{up}}$ | Weight matrices for aggregation and update MLPs in MinAgg GNN. |
| $b_j^{\mathrm{agg}}, b_j^{\mathrm{up}}$ | Bias vectors for aggregation and update MLPs in MinAgg GNN. |
| $K$ | Number of message passing steps in the MinAgg GNN. |
| $m$ | Number of layers in the MLPs used for aggregation and update in MinAgg GNN. |
| $L$ | Total number of layers in the MinAgg GNN. |
| $d$ | Dimensionality of hidden features in MinAgg GNN layers. |
| **Training and Loss Functions** ||
| $\mathcal{H}_{\mathrm{small}}$ | A set of small training graphs used to analyze GNN performance. |
| $\mathcal{G}_{\mathrm{train}}$ | Set of training examples for the MinAgg GNN, consisting of input-output graph pairs. |
| $\mathcal{L}_{\mathrm{reg}}$ | Regularized loss function for MinAgg GNN, combining training loss and parameter sparsity penalty. |
| $\mathcal{L}_{\mathrm{MAE}}$ | Mean absolute error loss over the training set. |
| $\eta$ | Regularization coefficient for sparsity in the MinAgg GNN. |
| $\mathcal{E}_{\mathrm{test}}$ | Multiplicative error over the test set. |
| $\mathcal{E}_{\mathrm{reg}}$ | Model sparsity combined with mean absolute error over the training set. Same as $\mathcal{L}_{\mathrm{reg}}$. |
| $\mathcal{E}_{\mathrm{train}}$ | Mean squared error over the training set. |

Table 6: Notation table summarizing key symbols and terms.

# G  LLM Use

LLMs were used for refining writing, proof-reading, and organizing citations.

# H  Table of Contents

## Contents

