# OpenReview forum: "Graph neural networks extrapolate out-of-distribution for shortest paths"
_ICLR.cc/2026/Conference — Submitted to ICLR 2026_

### Official Review · Reviewer_7xvo · 2025-10-26

**Soundness:** 3
**Presentation:** 4
**Contribution:** 3
**Rating:** 4
**Confidence:** 3

**Summary:**

This work aims to bridge the gap between graph algorithms and GNN models. The paper introduces a message-passing GNN variant that can approximate the steps of the Bellman-Ford algorithm when trained with a loss function that promotes sparsity. It is shown that the model can achieve OOD generalization in computing Bellman-Ford steps across varying graph structures. The theoretical results are supported by empirical evaluations on synthetic datasets.

**Strengths:**

- The problem addressed in this paper is highly relevant. Designing GNNs to mirror the steps of graph algorithms is an interesting direction. So far, the ability of these models to simulate such algorithms has been supported mainly by empirical evidence.

- The main strength of the paper lies in its theoretical contribution. The theoretical results demonstrate that message-passing GNNs that use the min aggregator can learn to simulate the Bellman–Ford algorithm. Once the models are properly trained, they can provably achieve OOD generalization, which is a particularly interesting result.

- Overall, the paper is well-written and easy to read. It is also well-structured, thus enhancing the reader's understanding.

**Weaknesses:**

- While the experimental results support the theoretical findings, the evaluation is restricted to simplified settings where only at most two steps of the Bellman–Ford algorithm are simulated in a single forward pass of the model. Therefore, it does not capture real-world scenarios in which nodes are distant from each other and more than two steps are required.

- No empirical results are provided to demonstrate that the proposed GNN model can achieve high performance on tasks where shortest path distances between nodes are important. For instance, the authors could construct a synthetic dataset where node interactions depend on the nodes' shortest path distances, and show that the proposed model can outperform other standard GNN architectures.

- Theorem 2.2 appears to have limited significance, since it is restricted to a single message-passing iteration and one iteration of the Bellman–Ford algorithm. As a result, its practical relevance is somewhat limited.

- It is not clear how much the loss $L_{reg}(G_{train}, A_\theta)$ deviates from its global minimum across different $\mathscr{G}_K$ instances. If this deviation is large, the bound can be quite loose. In the current experiments, only a single training set $\mathscr{G}_K$ is considered. A more comprehensive analysis is required.

- In my understanding, the $\Gamma$ map defined in l.214 produces node features that are all equal to 0 since $x_u=0$ for all $u \in V$, $x(u,v) \geq 0$ for all $u \in V$ and $x(v, v) = 0$. Therefore, $x_u + x(v,v) = 0$. I suppose that the node itself should not be included into its neighborhood in the definition of the $\Gamma$ map.

**Questions:**

- What is the purpose of Theorem 2.2 given that Theorem 2.3 is more general? The result of Theorem 2.2 holds only for a single message-passing iteration which has limited practical significance.

---

> ### Author Response · Authors · 2025-11-25
>
> **Weaknesses:**
>
> 1. **No empirical results are provided to demonstrate that the proposed GNN model can achieve high performance on tasks where shortest path distances between nodes are important. For instance, the authors could construct a synthetic dataset where node interactions depend on the nodes' shortest path distances, and show that the proposed model can outperform other standard GNN architectures.**
>
> Thank you for the suggestion. We have performed additional experiments beyond shortest path computations in Section D.4. Here we perform an experiment on a synthetic task where shortest-path computation is a component of a larger computation, which requires learning some hidden function of the edge weights. These experiments demonstrate that the regularized model achieves superior extrapolation for these tasks.
>
> 2. **It is not clear how much the loss deviates from its global minimum across different instances. If this deviation is large, the bound can be quite loose. In the current experiments, only a single training set is considered. A more comprehensive analysis is required.**
>
> Thank you for raising this.Theorem 2.3 yields a bound that scales like $1/M$, where $M$ is the total number of nodes across the training graphs. For the canonical training set $\mathscr{G}_K$ (smallest $M$), the admissible interval for the global minimum is $[0.00742, 0.00862]$, and our achieved minimum $0.00747$ lies within it. We agree that using different training sets would be interesting for future work.
>
> 3. **Concern about self-aggregation and initialization: including a node in its own neighborhood might seem to force all node features to 0 unless initialization is handled carefully.**
>
> We do in fact include a node's feature in its own aggregation. However, since only the source has initial node feature zero, this does not cause all node features to be zero. We define this node feature initialization around l.184.
>
> **Questions:**
>
>
> 1. **What is the purpose of Theorem 2.2 given that Theorem 2.3 is more general? The result of Theorem 2.2 holds only for a single message-passing iteration which has limited practical significance.**
>    While Theorem 2.2 is more limited in scope, it demonstrates in a simple setting that networks with few parameters can achieve guaranteed extrapolation. Specifically, when a GNN has sufficiently few parameters, low training loss on a carefully chosen training set forces the network to implement the correct algorithmic structure, enabling extrapolation to arbitrary graphs. This observation motivates sparsity regularization in Theorem 2.3: by constraining the number of non-zero parameters, we reduce the analysis of overparameterized networks to the simpler limited-parameter case. The proof techniques developed for Theorem 2.2, particularly how training constraints force specific parameter configurations, are refined and extended in Theorem 2.3, making the simpler result a stepping stone to the more general guarantee.

---

### Official Review · Reviewer_ycWC · 2025-10-30

**Soundness:** 3
**Presentation:** 2
**Contribution:** 2
**Rating:** 4
**Confidence:** 5

**Summary:**

This paper shows that Graph Neural Networks trained with a sparsity-regularized loss can learn to implement the Bellman-Ford shortest-path algorithm, thereby provably extrapolating to graphs far larger and structurally different from those seen in training.

**Strengths:**

1. This works provides strong theoretical contribution, a formal extrapolation guarantee for message-passing GNNs on shortest path, going beyond prior intuitive “algorithmic alignment” discussions.
2. The setup  formalization is clear and rigorous
3. The theoretical results are backed up by empirical validations.

**Weaknesses:**

1. The scope of this work is very narrow , the results are confined to shortest-path problems and to a Bellman-Ford–aligned architecture, and extension to other algorithms or even other graph problems remains untested.
2. IMHO the main weakness  of this work is that the practical contribution is unclear. While it is fair that theory is done in a simplified way, it should be clear how the community benefits from these theoretical insights. E.g., whether the conclusions are carried to real-tasks and real-setting, or if we can derive practical insights or conclusions to how to improve GNN generalization in such settings.
Specifically, there are no experiments on real-world dataset, e.g. with the sparse-regularized training.
3. Shortest path is not a task that is important of its own, as it has an efficient algorithm to solve without ML. Therefore, it has to be clear why is it interesting to know that GNNs can implement it in some setting other than for just knowing? While this is cool, IMHO it does not suffice without some clear insight based on this discovery.

**Questions:**

Please provide any practical insight or evaluation for the theoretical results, e.g. evaluate whether Sparse-regularized training is also beneficial for other OOD size generalization?

---

> ### Author Response · Authors · 2025-11-25
>
> **Weaknesses:**
>
> 1. **The scope of this work is very narrow , the results are confined to shortest-path problems and to a Bellman-Ford–aligned architecture, and extension to other algorithms or even other graph problems remains untested.**
>
>  We have expanded our evaluation beyond the initial setting: Appendix D.3 adds diverse synthetic and real-world graphs, and Appendix D.4 studies tasks where shortest-path computation is only one component of a more complex computation. Across these, the regularized, aligned model shows superior extrapolation.
>
> 2. **IMHO the main weakness of this work is that the practical contribution is unclear. While it is fair that theory is done in a simplified way, it should be clear how the community benefits from these theoretical insights. E.g., whether the conclusions are carried to real-tasks and real-setting, or if we can derive practical insights or conclusions to how to improve GNN generalization in such settings. Specifically, there are no experiments on real-world dataset, e.g. with the sparse-regularized training.**
>
> Thank you for raising this. We have added experiments on real-world datasets (Norway terrain triangulation with ~40k vertices; Airports USA network) and more complex synthetic graphs (Appendix D.3), as well as tasks where shortest-path computation is a component of a larger pipeline with hidden edge-weight transformations (Appendix D.4). In all cases, sparsity-regularized, algorithm-aligned models exhibit superior extrapolation, supporting practical relevance. We emphasize that our goal is not to provide a practically competitive shortest-path solver; rather, shortest paths serve as a controlled setting to study how alignment and regularization enable out-of-distribution extrapolation, consistent with our scope clarification above.
>
> 3. **Shortest path is not a task that is important of its own, as it has an efficient algorithm to solve without ML. Therefore, it has to be clear why is it interesting to know that GNNs can implement it in some setting other than for just knowing? While this is cool, IMHO it does not suffice without some clear insight based on this discovery.**
>    Out-of-distribution generalization capabilities of GNNs are not well-understood. Previous work has suggested that aligning an architecture to algorithmic computational paths helps with OOD generalization, but this has remained largely intuitive. Our work provides the first concrete demonstration that algorithmic alignment, combined with appropriate regularization, can provably enable OOD extrapolation. A key insight emerging from our analysis is that alignment can yield provable benefits to OOD generalization, and that regularization makes the analysis of these benefits tractable. Our analysis reveals that achieving provably guaranteed OOD generalization relies on the interplay between training samples, regularization (loss), model architecture, and task structure. In particular, suitable combinations of these elements can indeed lead to provably guaranteed OOD generalization. Importantly, our experiments on hidden edge weight transformations (see Appendix D.4) show that this framework remains effective even when the full desired computation is not known, extending the value of algorithmic alignment beyond simply implementing known algorithms.
>
> **Questions:**
>
> 1. **Please provide any practical insight or evaluation for the theoretical results, e.g. evaluate whether Sparse-regularized training is also beneficial for other OOD size generalization?**
>    We believe that sparsity regularization could be an important practical regularization technique to help achieve OOD generalization more broadly. Our work demonstrates that the interplay between training samples, regularization (loss), model architecture, and task structure is crucial for achieving provably guaranteed OOD generalization. While our theoretical analysis is specific to shortest paths and algorithm-aligned GNNs, the principle that appropriate regularization can make extrapolation analysis tractable may extend to other settings. It would be interesting to test sparsity-regularized training on larger-scale problems and other types of OOD size generalization tasks to evaluate its broader applicability. More generally, we hope this work encourages the exploration of other forms of regularization that promote extrapolation in different settings, as the specific regularization that enables tractable analysis may vary depending on the problem structure. We have conducted additional experiments (Appendix D) showing that $L_1$-regularized, algorithm-aligned models extrapolate better across diverse synthetic and real datasets and in problems where Bellman-Ford is one component (hidden edge weight transformations).

---

### Official Review · Reviewer_MFFo · 2025-10-31

**Soundness:** 3
**Presentation:** 2
**Contribution:** 3
**Rating:** 6
**Confidence:** 3

**Summary:**

This paper investigates out-of-distribution (OOD) size generalization for shortest-path computation with message-passing GNNs. The authors propose a Min-Aggregation GNN (MinAgg) designed to align with Bellman–Ford (BF) updates and train it using a sparsity-regularized objective. The key idea is that learning on a small, curated toy set can certify extrapolation to arbitrary-size graphs. Two theoretical results support the claim: (i) a $1$-layer theorem showing that near-zero loss forces the network to implement approximately one BF step; and (ii) a general theorem for $L$-layer MinAgg establishing that, under near-minimal $L_0$-regularized loss, the network correctly implements $K$ steps of BF. Empirical results corroborate the theory, demonstrating stronger size-OOD performance than unregularized baselines.

**Strengths:**

1. The paper targets a canonical algorithmic task (shortest paths) where alignment to BF is intuitive and verifiable, making the claims clear.

2. Treating a small, carefully designed training set and a sparsity term as an OOD certificate is insightful and potentially portable to other algorithmic operators.

3. The two theorems are precise, with explicit error control.

**Weaknesses:**

1. The guarantees are proved for $L_0$ regularization, whereas training uses $L_1$, which leaves a theory–practice gap.

2. The theory guarantees only $K$ BF steps and requires $L \ge K$. Thus, arbitrary size extrapolation is limited to $\le$ K hops, and scaling to larger effective diameters demands larger K (hence deeper networks) or iterative reuse of a trained K-step block.

3. Graphs are restricted to undirected graphs with non-negative weights and zero self-loops, while BF handles directed graphs and negative edges (without negative cycles). It is unclear whether the method and analysis can be extended to those settings.

**Questions:**

1. How large can $K$ be before $G_K$ becomes impractical? Please evaluate and report for a larger K.

2. Can the proof technique and MinAgg architecture be adapted to directed graphs and negative weights without negative cycles? If not, please clarify the limitations.

3. Could you report inference cost compared to running classical BF, and any compression benefits from learned sparsity?

---

> ### Author Response · Authors · 2025-11-25
>
> **The guarantees are proved for $L_0$ regularization, whereas training uses $L_1$, which leaves a theory–practice gap.**
>    Our theoretical guarantees are stated for an $L_0$ penalty because this makes the extrapolation analysis tractable. In our experiments, we instead train with an $L_1$ penalty. For the models we report, it turns out that our learned parameters (via $L_1$ penalty) in fact also minimize the corresponding $L_0$-regularized objective, in the sense that they induce the same sparsity pattern and thus achieve low $L_0$-regularized loss value. For these trained models, the extrapolation guarantee therefore applies directly. What we do not provide is a general result showing that minimising the $L_1$-regularised loss will always recover an $L_0$ minimiser in our setting. The sparsity-inducing behaviour of $L_1$ is well studied, however, and formalising this link for our architecture is a natural direction for future work.
>
> 1. **How large can $K$ be before $\mathscr{G}_K$ becomes impractical? Please evaluate and report for a larger $K$.**
>    The training set $\mathscr{G}_K$ contains $5K + 4K^2$ graphs, which scales quadratically in $K$. This means the training set remains tractable even for moderately large $K$. In our experiments, we successfully train models with $K=2$, though convergence becomes more challenging for $K=3$ with extended training. We note that graph neural networks tend to be shallow in practice due to oversmoothing problems, with most networks having fewer than 4 layers. We will include the explicit characterization of the training set size $|\mathscr{G}_K| = 5K + 4K^2$ in the paper.
>
> 2. **Can the proof technique and MinAgg architecture be adapted to directed graphs and negative weights without negative cycles? If not, please clarify the limitations.**
>    Our approach is not directly applicable to graphs with negative edge weights. Representing Bellman-Ford for negative weights requires non-zero bias terms, as otherwise the ReLU activation zeros out negative contributions. We conjecture that a similar analysis could be performed for negative edge weights, but the sparse representation would need to be different. However, our techniques easily generalize to directed graphs: the same sparsity structure applies, requiring only a change from bidirectional to directed message passing. We chose not to include this analysis in the paper, as it does not provide substantial additional insight and GNNs for directed graphs are less common in practice.
>
> 3. **Could you report the inference cost compared to running classical BF, and any compression benefits from learned sparsity?**
>     We added an inference-time benchmark comparing our trained 2-step Bellman–Ford model against NetworkX on Erdős–Rényi graphs. The CPU model is competitive for small graphs, while the GPU model attains speedups on larger graphs. Please see Appendix D.5 of the main paper for full setup and results. We believe the speedup is due to pytorch optimization.
>
>     We reiterate that our work does not aim to provide practical benefits over classical algorithms. Rather, we use shortest paths as a controlled setting to study how algorithmic alignment and regularization enable out-of-distribution extrapolation. The learned sparsity does not yield compression benefits, as it simply recovers the structure inherent to the Bellman-Ford algorithm. Our contribution is the theoretical framework demonstrating when and how such extrapolation can be guaranteed, not practical performance improvements. However, our experiments on hidden edge weights (see Appendix D.4) demonstrate that our techniques can be beneficial even when the full computation is not known a priori. In such scenarios, there may not be a fast previously known algorithm available, making the neural approach a viable alternative for discovering and executing the desired computation.

---

### Official Review · Reviewer_11s3 · 2025-11-01

**Soundness:** 3
**Presentation:** 3
**Contribution:** 3
**Rating:** 4
**Confidence:** 4

**Summary:**

In the submitted manuscript, the authors study the generalisation of Graph Neural Networks (GNNs) across graphs of different sizes for the shortest path problem. To do so, they study a GNN with minimum aggregation and prove strong theoretical results on how the output of the Bellman Ford algorithm both upper and lower bounds the predictions of their minimum aggregation GNN. This theoretical progress is complemented by strong empirical results of a regularised version of their GNN variant on several synthetic graph generators.

**Strengths:**

- The paper is well-written and clear.

- The motivation of the work is strong and the results are interesting. This seems like a very promising research direction to me.

- The analysis of the learned parameters and the observation of how closely they align with the parameters of the Bellman Ford algorithm is very nice.

**Weaknesses:**

- It seems to me that the experiments are on a very limited set of rather regular, synthetic graphs. Results on real-world data or at least more challenging synthetic graph generators would add to the empirical credibility of the work (see my questions for suggestions in this direction).

- I believe that some of the results are not quite correctly interpreted.

**Questions:**

1] I am very happy to see that you use the minimum as the aggregation function in your GNN. It is a well-motivated use of this function and makes for a suitable GNN. But in practice, I am not sure how the gradient flow of your model works then. The derivative of the min function is not continuous, so how are you able to differentiate through it?

2] In your proposed solution, you rely on the insight that the underlying algorithmic solution, i.e., the Bellman Ford (BF) algorithm, is sparse in the parameters. Your general formulation is not able to discover the BF algorithm and you need to regularise the loss to obtain it. Could you comment on the degree to which this approach generalises; are most algorithms on graphs sparsely parameterised and therefore the regularisation of GNNs is a promising direction in general to solve algorithmic problems or is the current approach unique to the shortest path problem?

3] It seems to me that the experiments are insufficient to demonstrate that your proposed approach is of practical relevance to the shortest path problem.

3.1] The results you display in Table 1 demonstrate results on growing graphs with a constant expected degree. This may be a rather easy generalisation scenario since the local 'topology' of each graph may stay constant if the expected degree of each node on all graphs is equal. Would the same results hold if you randomly sample $p$ from a distribution for the test set?

3.2] The test set up you describe in Lines 402-10 states that the test set contains 3-cycles, 4-cycles, complete graphs and ER graphs with $p=0.5$. May this collection of test graphs not be too simple? On complete graphs all shortest paths are 1 and cycles also have a relatively regular shortest path structure. Would your results still hold if you include more complex graph generators, like for example stochastic block models or Barabási–Albert graphs?

3.3] Generally, it seems to me that it would be good to train on subgraphs of real world data and observe whether you generalise to the whole graph or whether you are able to partition graph level datasets into smaller and larger graphs and to observe whether training on the smaller graphs allows you to generalise to the whole graph.

4] In Line 425 you state that the results in Figure 4 (a) and (b) validate your result in Theorem 2.3. I am not sure how direct this validation is. Is it not just two disjoint pieces of evidence pointing to the same conclusion? Would you not need to analytically calculate the bound that you derive in Theorem 2.3 and show that your predictions fall within the bound to speak of real validation of the theorem (I apologise if the bound value can be deduced from Figure 4, I was not able to do so based on my understanding).

5] In Lines 436-7 you state that the test error of the $L_1$-regularised model does not accumulate. I could not find this in the results. For the regularised model we observe an increase in the error both as the graph size increases and if we compare row entries of columns 2 and 4 in Table 1. Am I misreading the table or should this claim be softened?

6] Minor Comments:

6.1] In Line 181 you write that the "initial conditions and final answer are therefore contained in the node embeddings". I did not fully understand this. Which final answer are you putting in the node embeddings? The example that you provide for the shortest path problem, in which you distinguish the source node from all other nodes in the node embeddings does not seem to include an embedding component that reveals final answers.

6.2] Figure 1 is not discussed in the text and I personally feel that this general visualisation of GNNs does not contribute much to the understanding of your work. I think it could be moved to the appendix at almost no cost to your paper.

6.3] Your Figures in Appendix D don't seem to have captions and there is some characters in the pdf "[t]" suggesting that there may be an error in the tex associated with these figures.

---

> ### Author Response · Authors · 2025-11-25
>
> **Questions:**
>
> 1.
>    Torch-geometric, the package we use for our GNN implementation, differentiates through min-aggregation using a subgradient of the minimum function. Concretely, if $y = \min\{x_j : j \in [n]\}$, backpropagation sets $\partial y/\partial x_i = 1/m$ for each of the $m$ entries that attain the minimum and 0 for all others. At points where several entries tie for the minimum, any convex combination of these indicator gradients is a valid subgradient; the implementation corresponds to choosing the uniform combination over minimizers. This is the standard subgradient treatment of piecewise-linear, almost-everywhere differentiable functions, and it is well studied in the optimization literature.
>
> 2.
>    Our analysis is specific to shortest paths and to the particular sparse, algorithm-aligned GNN we study. It is not clear if other graph algorithms automatically admit the same treatment. Rather, our results isolate two ingredients: alignment with a known algorithmic implementation, and a regularization that makes extrapolation analysis tractable. These ideas could, in principle, be instantiated more broadly. For algorithms that admit local, iterated-update, sparsely parameterized realizations, our framework suggests that regularizing a similarly sparse, aligned GNN is a natural direction to explore. Rigorously extending our proofs to such settings remains an open problem. More generally, even when such sparse realizations are not available, we hope this work encourages the design of other forms of regularization that promote extrapolation.
>
> 3.
>    We do not primarily aim to provide practical relevance to the shortest path problem, although we do believe some practical insights can be made. Our primary goal is to use shortest paths computations, and their alignment with GNNs, as a clean and controlled test bed for understanding how algorithmic alignment aids out-of-distribution extrapolation [2,3]. The main contribution is therefore the conceptual and theoretical framework. In particular, the technique of introducing regularization to make extrapolation analysis tractable may be useful beyond this specific problem setting. Experiments on hidden edge weight transformations (see Appendix D.4) demonstrate practical relevance when the full computation is not known. In these settings, the model must learn unknown components while executing the algorithmic computation, showing that shortest path computation can be implicit and the model must discover what that computation is. Additionally, our results demonstrate that it is important to consider the interaction between training data, regularization, and architecture design in order to obtain generalizable models. This approach suggests that sparsity regularization will be useful for shortest path or similar graph problems.
>    1.
>       It is true that most generalization in GNNs relies on train and test graphs having similar local structure. Indeed, the power of our result is precisely that it goes beyond this setting; our extrapolation guarantee applies to all test graphs, regardless of size or topology. To make this more clear we have included a number of additional experiments (see Appendix D.3) on a variety of different test graphs. In all cases, our regularized and aligned model shows superior generalization.
>    2.
>       We agree that our test graphs are overly limited in complexity. We have therefore added a number of experiments on more complex graph generators, including stochastic block models, Barabási–Albert graphs, and random geometric graphs, as well as real-world datasets (see Appendix D.3). Again we reiterate that our theoretical results do indeed apply to all graphs, regardless of topology.
>    3.
>       This is an interesting idea, and it aligns with previous work on generalization in GNNs. However, our focus is on guaranteed extrapolation, and our theoretical results on this require a carefully chosen training set. Thus, our results do not directly apply to the setting you suggest, where training occurs on subgraphs of real-world data, as this would not satisfy the conditions needed for our proof. A valuable direction for future work would be investigating what properties of training graphs are required for guaranteed generalization, and if these properties occur in random graphs or in real world graphs. This would also be an interesting direction to explore empirically.

---

> > ### Author Response · Authors · 2025-11-25
> >
> > 4.
> >    We thank the reviewer for raising this, and we agree more explicit clarification is needed. Theorem 2.3 yields a bound that scales like $1/M$, where $M$ is the total number of nodes across the training graphs. Applying the theorem only to the canonical training set $\mathscr{G}_K$ (i.e., the smallest $M$) gives the largest admissible $\varepsilon$ and hence the widest interval. The predicted global minimum is $\frac{6}{2M(mL + mK + K)}$ and the theorem applies to any $\varepsilon \le \frac{1}{2M(mL + mK + K)}$, i.e., $[\frac{6}{2M(\cdot)}, \frac{6}{2M(\cdot)} + \frac{1}{2M(\cdot)}]$. In our setting, this is $[0.00742, 0.00862]$, and our experimentally observed minimum $0.00747$ lies within this range, so the theorem applies directly. If we instead include all training graphs used in the experiments, $M$ increases and the interval tightens. The resulting bound nearly contains the empirical minimum, as shown by the red region in Figure 4(a). We will include this brief analysis and give change the figure reporting the loss computed over just the $\mathscr{G}_K$ training set to make the connection explicit.
> >
> > 5.
> >    We thank the reviewer for pointing this out, and we agree the statement should be amended. A single forward pass of the network will return the correct $K$-step Bellman-Ford distance to source along with some small multiplicative error. However, this error accumulates as we iterate the network. On the other hand, one can see clearly that the error accumulates much more slowly than the non-aligned network over several different types of graphs (see Appendix D.3 for detailed analysis of error accumulation when iterating the GNN module). We will clarify this relative accumulation in the text and figures.
> >
> > 6.
> >    1.
> >       The phrase “final answer” refers to the output of the GNN: the shortest path distances stored in the final node embeddings $h_v^{(L)}$ after $L$ layers. The initial node embeddings $X_{\mathrm{v}}$ encode the problem input (source node $x_v = 0$, unreachable nodes $x_v = \beta$), while the final embeddings $h_v^{(L)}$ contain the computed solution. We will revise the text to make this distinction explicit.
> >    2.
> >       We will add a reference to Figure 1 in the text where we introduce the GNN architecture to clarify its role.
> >    3.
> >       We thank the reviewer for catching this. We have fixed the LaTeX errors in the figures in Appendix D.

---

### Author Response · Authors · 2025-11-25

We would like to thank all reviewers for their thoughtful feedback. We address two recurring concerns here before responding to individual points.

- **Clarifying our scope and contribution.** Several reviewers raised questions about the practical relevance of studying shortest paths, given that efficient classical algorithms already exist. We emphasize that our primary goal is not to provide a practically competitive method for shortest path computation or graph problems more broadly. Instead, we use shortest paths and an aligned GNN as a controlled test bed to understand when and how alignment and regularization can support out-of-distribution extrapolation. In this sense, our work is closer in spirit to mechanistic interpretability [2,3], where relatively simple models and tasks are studied in depth to illuminate general principles, rather than to improve state-of-the-art performance. The impact of algorithmic alignment on OOD generalization has been studied empirically in benchmarks such as CLRS [1], but theoretical understanding has remained limited. Our main contribution is the first provable demonstration that algorithmic alignment, combined with appropriate regularization, can enable guaranteed out-of-distribution extrapolation, a capability that is not well-understood for GNNs more generally. Our work suggests that achieving provably guaranteed OOD generalization relies on the interplay between training samples, regularization (loss), model architecture, and task structure. In particular, suitable combinations of these elements can indeed lead to provably guaranteed OOD generalization. Practical relevance is also addressed by our additional experiments (discussed further below) which show regularization and algorithmic alignment help models learn hidden algorithmic structures even when the full computation is not known.

- **Expanding experimental evaluation.** Several reviewers noted that our initial experimental evaluation could be expanded, particularly in the diversity of test graphs. To address this, we have conducted extensive additional experiments, detailed in Appendix D (D.3, D.4, and D.5) of the main paper (a new version has been uploaded). These include: (1) evaluation on more complex synthetic graph generators (stochastic block models, Barabási–Albert graphs, random geometric graphs); (2) evaluation on real-world datasets (Norway terrain triangulation with 40,000 vertices, Airports USA network); (3) comparison with GAT baselines; (4) analysis of error accumulation when iterating the GNN module; and (5) experiments on problems where Bellman-Ford is only one component of a larger computation. (5) is particularly important as it demonstrates that algorithmic alignment remains effective when we have only a prior over the desired computation (e.g., shortest-path computation) and must infer the remaining components from data. This scenario mirrors real applications—e.g., routing to minimize fuel or energy consumption where per-road cost is a nonlinear function of observable features (e.g., grade, speed limit, and congestion) rather than raw length alone. Across all these diverse test settings, our L1-regularized, algorithm-aligned models consistently achieve substantially lower extrapolation error than unregularized models, demonstrating that the benefits of our approach extend well beyond the simple test graphs in our initial evaluation. Importantly, our theoretical guarantees apply to all graphs regardless of topology, and these experiments provide empirical validation of this claim.

## References

1. Petar Veličković, Adria Puigdomènech Badia, David Budden, Razvan Pascanu, Andrea Banino, Misha Dashevskiy, Michal Valko, and Charles Blundell. The CLRS Algorithmic Reasoning Benchmark. In International Conference on Machine Learning (ICML), 2022.
2. Ziqian Zhong, Ziming Liu, Max Tegmark, and Jacob Andreas. The Clock and the Pizza: Two Stories in Mechanistic Explanation of Neural Networks. arXiv preprint arXiv:2306.17844, 2023.
3. Neel Nanda, Lawrence Chan, Tom Lieberum, Jess Smith, and Jacob Steinhardt. Progress Measures for Grokking via Mechanistic Interpretability. In International Conference on Learning Representations (ICLR), 2023.

---

### Meta-Review · Area_Chair_vX2e · 2025-12-30

**Summary:**

In this paper, the authors study graph out-of-distribution (OOD) generalization from algorithmic learning paradigms, particularly the canonical shortest path problem. The authors show that GNNs are guaranteed to extrapolate under certain loss functions and provide experimental support for the theory.

The reviewers raised various concerns ranging from the significance of the theoretical findings (and even the studied problem), insufficient experimental support (e.g., limited settings), and other technical issues. Though the authors provide a rebuttal, including extended discussions on the setting and more experiments, none of the reviewers responded. Considering the rebuttal and the discussions that could have taken place, the paper makes a borderline case. Though the theoretical contributions are potential contributions, the scope, experiments, and whether the results are significant for the general audience are somewhat limited, as reflected by none of the reviewers particularly championing the paper. Considering the highly competitive nature of ICLR, I think the paper does not meet the bar for acceptance yet, but encourage the authors to revise the paper for another round. Currently, I vote for rejection.

**Reviewer Concerns:**

Reviewer 11s3:
W1-Experiments are limited: not addressed.
W2-Some of the results are not quite correctly interpreted: partially addressed.

Reviewer MFFo:
W1/W2-Technical details: partially addressed.
W3-Whether the method and analysis can be extended: not addressed.

Reviewer ycWC:
W1-The scope of this work is very narrow: though authors provide extended discussions, I am not sure the reviewer could be convinced.
W2-Practical contribution is unclear: partially addressed.
W3-Why is it interesting: I am not sure the reviewer could be convinced.

Reviewer 7xvo:
W1-the evaluation is restricted to simplified settings: not addressed.
W2-No empirical results are provided… : partially addressed.
W3-Theorem 2.2 appears to have limited significance: not addressed.
W4-Details regarding the proposed method: likely partially addressed.

**Reviewer Scores:**

For Reviewer 11s3, the initial rating is 4, and it is likely to stay at 4 or increase to 6.

For Reviewer MFFo, the initial rating is 6, and it is likely to stay at 6.

For Reviewer ycWC, the initial rating is 4, and it is likely to stay at 4.

For Reviewer 7xvo, the initial rating is 4, and it is likely to stay at 4.

---

### Decision · Program_Chairs · 2026-01-26

Reject